# Stochastic Batch Acquisition:
# A Simple Baseline for Deep Active Learning

**Andreas Kirsch**[*]  *andreas.kirsch@cs.ox.ac.uk*
*OATML, Department of Computer Science, University of Oxford*[†]

**Sebastian Farquhar**[*]  *sebastian.farquhar@cs.ox.ac.uk*
*OATML, Department of Computer Science, University of Oxford*[†]

**Parmida Atighehchian**  *parmi.atg@gmail.com*
*ServiceNow*[†]

**Andrew Jesson**  *andrew.jesson@cs.ox.ac.uk*
*OATML, Department of Computer Science, University of Oxford*[†]

**Frédéric Branchaud-Charron**  *frederic.branchaud.charron@gmail.com*
*ServiceNow*[†]

**Yarin Gal**  *yarin.gal@cs.ox.ac.uk*
*OATML, Department of Computer Science, University of Oxford*

**Reviewed on OpenReview:** *https://openreview.net/forum?id=vcHwQyNBjW*

## Abstract

We examine a simple stochastic strategy for adapting well-known single-point acquisition functions to allow batch active learning. Unlike acquiring the top-$K$ points from the pool set, score- or rank-based sampling takes into account that acquisition scores change as new data are acquired. This simple strategy for adapting standard single-sample acquisition strategies can even perform just as well as compute-intensive state-of-the-art batch acquisition functions, like BatchBALD or BADGE while using orders of magnitude less compute. In addition to providing a practical option for machine learning practitioners, the surprising success of the proposed method in a wide range of experimental settings raises a difficult question for the field: when are these expensive batch acquisition methods pulling their weight?

## 1 Introduction

Active learning is a widely used strategy for efficient learning in settings where unlabelled data are plentiful, but labels are expensive (Atlas et al., 1989; Settles, 2010). For example, labels for medical image data may require highly trained annotators, and when labels are the results of scientific experiments, each one can require months of work. Active learning uses information about unlabelled data and the current state of the model to acquire labels for those samples that are most likely to be informative.

While many acquisition schemes are designed to acquire labels one at a time (Houlsby et al., 2011; Gal et al., 2017), recent work has highlighted the importance of *batch acquisition* (Kirsch et al., 2019; Ash et al., 2020). Acquiring in a batch lets us parallelise labelling. For example, we could hire hundreds of annotators to work in parallel or run more than one experiment at once. Batch acquisition also saves compute as single-point selection also incurs the cost of retraining the model for every new data point.

---

[*]Joint first authors
[†]Work done while there.

Table 1: *Acquisition runtime (in seconds, 5 trials, $\pm$ s.d.).* The stochastic acquisition methods are as fast as top-$K$, here with BALD scores, and **orders of magnitude** faster than BADGE or BatchBALD. Synthetic pool set with $M = 10,000$ pool points with 10 classes. BatchBALD & BALD: 20 parameter samples. In [superscript], mean accuracy over acquisition steps from Figure 2 on Repeated-MNIST with 4 repetitions (5 trials).

| $K$ | Top-$K$ (BALD) [80%] | Stochastic (PowerBALD) [90%] | BatchBALD [90%] | BADGE [86%] |
|---|---|---|---|---|
| 10 | $0.2 \pm 0.0$ | $0.2 \pm 0.0$ | $566.0 \pm 17.4$ | $9.2 \pm 0.3$ |
| 100 | $0.2 \pm 0.0$ | $0.2 \pm 0.0$ | $5,363.6 \pm 95.4$ | $82.1 \pm 2.5$ |
| 500 | $0.2 \pm 0.0$ | $0.2 \pm 0.0$ | $29,984.1 \pm 598.7$ | $409.3 \pm 3.7$ |

Unfortunately, existing performant batch acquisition schemes are computationally expensive (Table 1). Intuitively, this is because batch acquisition schemes face combinatorial complexity when accounting for the interactions between possible acquisition points. Recent works (Pinsler et al., 2019; Ash et al., 2020; 2021) trade off a principled motivation with various approximations to remain tractable. A commonly used, though extreme, heuristic is to take the top-$K$ highest scoring points from an acquisition scheme designed to select a single point ("*top-K acquisition*").

This paper examines a simple baseline for batch active learning that is competitive with methods that cost orders of magnitude more across a wide range of experimental contexts. We notice that single-acquisition score methods such as BALD (Houlsby et al., 2011) do not take correlations between the samples into account and thus only act as a noisy proxy for future acquisition scores as we motivate in Figure 1. This leads us to stochastically acquire points following a distribution determined by the single-acquisition scores instead of acquiring the top-$K$ samples. In sequential decision making, stochastic multi-armed bandits and reinforcement learning, variants of this are also known a Boltzmann exploration (Cesa-Bianchi et al., 2017). We examine other related methods in §4. In deep active learning this simple approach remains under-explored, yet it can match the performance of earlier more complex methods for batch acquisition (e.g., BatchBALD, Kirsch et al. (2019); see in §5) despite being very simple. Indeed, this acquisition scheme has a time complexity of only $\mathcal{O}(M \log K)$ in the pool size $M$ and acquisition size $K$, just like top-$K$ acquisition.

We show empirically that the presented stochastic strategy performs as well as or better than top-$K$ acquisition with almost identical computational cost on several commonly used acquisition scores, empirically making it a strictly-better batch strategy. Strikingly, the empirical comparisons between this stochastic strategy and the evaluated more complex methods cast doubt on whether they function as well as claimed. Concretely, in this paper we:

- examine a family of three computationally cheap stochastic batch acquisition strategies (softmax, power and soft-rank)—the latter two which have not been explored and compared to in detail before;

- demonstrate that these strategies are preferable to the commonly used top-$K$ acquisition heuristic; and

- identify the failure of two existing batch acquisition strategies to outperform this vastly cheaper and more heuristic strategy.

**Outline.** In §2, we present active learning notation and commonly used acquisition functions. We propose stochastic extensions in §3, relate them to previous work in §4, and validate them empirically in §5 on various datasets, showing that these extensions are competitive with some much more complex active learning approaches despite being orders of magnitude computationally cheaper. Finally, we further validate some of the underlying theoretical motivation in §6 and discuss limitations in §7.

## 2 Background & Problem Setting

The stochastic approach we examine applies to batch acquisition for active learning in a pool-based setting (Settles, 2010) where we have access to a large unlabelled *pool* set, but we can only label a small subset of the points. The challenge of active learning is to use what we already know to pick which points to label in the most efficient way. Generally, we want to avoid labelling points similar to those already labelled. In the pool-based active learning setting, we are given a large pool of unlabeled data points and can request labels for only a small subset, the acquisition batch. To formalize this, we first introduce some notation.

**Notation.** Following Farquhar et al. (2021), we formulate active learning over *indices* instead over data points. This simplifies the notation. The large, initially fully unlabelled, pool set containing $M$ input points is

$$\mathcal{D}^{\text{pool}} = \{x_i\}_{i \in \mathcal{I}^{\text{pool}}}, \tag{1}$$

where $\mathcal{I}^{\text{pool}} = \{1, \ldots, M\}$ is the initial full index set. We initialise a training dataset with $N_0$ randomly selected points from $\mathcal{D}^{\text{pool}}$ by acquiring their labels, $y_i$,

$$\mathcal{D}^{\text{train}} = \{(x_i, y_i)\}_{i \in \mathcal{I}^{\text{train}}}, \tag{2}$$

where $\mathcal{I}^{\text{train}}$ is the index set of $\mathcal{D}^{\text{train}}$, *initially* containing $N_0$ indices between 1 and $M$. A model of the predictive distribution, $p(y \mid x)$, can then be trained on $\mathcal{D}^{\text{train}}$.

### 2.1 Active Learning

At each acquisition step, we select additional points for which to acquire labels. Although many methods acquire one point at a time (Houlsby et al., 2011; Gal et al., 2017), one can alternatively acquire a whole batch of $K$ examples. An acquisition function $a$ takes $\mathcal{I}^{\text{train}}$ and $\mathcal{I}^{\text{pool}}$ and returns $K$ indices from $\mathcal{I}^{\text{pool}}$ to be added to $\mathcal{I}^{\text{train}}$. We then label those $K$ datapoints and add them to $\mathcal{I}^{\text{train}}$ while making them unavailable from the pool set. That is,

$$\mathcal{I}^{\text{train}} \leftarrow \mathcal{I}^{\text{train}} \cup a(\mathcal{I}^{\text{train}}, \mathcal{I}^{\text{pool}}), \tag{3}$$

$$\mathcal{I}^{\text{pool}} \leftarrow \mathcal{I}^{\text{pool}} \setminus \mathcal{I}^{\text{train}}. \tag{4}$$

A common way to construct the acquisition function is to define some scoring function, $s$, and then select the point(s) that score the highest.

**Probabilistic Model.** We assume classification with inputs $X$, labels $Y$, and a discriminative classifier $p(y \mid x)$. In the case of Bayesian models, we further assume a subjective probability distribution over the parameters, $p(\omega)$, and we have $p(y \mid x) = \mathbb{E}_{p(\omega)}[p(y \mid x, \omega)]$.

Two commonly used acquisition scoring functions are Bayesian Active Learning by Disagreement (BALD) (Houlsby et al., 2011) and predictive entropy (Gal et al., 2017):

**BALD.** *Bayesian Active Learning by Disagreement* (Houlsby et al., 2011) computes the expected information gain between the predictive distribution and the parameter distribution $p(\omega \mid \mathcal{D}^{\text{train}})$ for a Bayesian model. For each candidate pool index, $i$, with mutual information, I, and entropy, H, the score is

$$
\begin{aligned}
s_{\text{BALD}}(i; \mathcal{I}^{\text{train}}) &\coloneqq \mathrm{I}[Y; \Omega \mid X = x_i, \mathcal{D}^{\text{train}}] \\
&= \mathrm{H}[Y \mid X = x_i, \mathcal{D}^{\text{train}}] - \mathbb{E}_{p(\omega \mid \mathcal{D}^{\text{train}})}[\mathrm{H}[Y \mid X = x_i, \omega]].
\end{aligned} \tag{5}
$$

**Entropy.** The *(predictive) entropy* (Gal et al., 2017) does not require Bayesian models, unlike BALD, and performs worse for data with high observation noise Mukhoti et al. (2021). It is identical to the first term of the BALD score

$$s_{\text{entropy}}(i; \mathcal{I}^{\text{train}}) \coloneqq \mathrm{H}[Y \mid X = x_i, \mathcal{D}^{\text{train}}]. \tag{6}$$

## 2.2 Acquisition Functions

These scoring functions were introduced for single-point acquisition:

$$a_s(\mathcal{I}^{\text{train}}) \coloneqq \underset{i \in \mathcal{I}^{\text{pool}}}{\arg\max}\, s(i; \mathcal{I}^{\text{train}}). \tag{7}$$

For deep learning in particular, single-point acquisition is computationally expensive due to retraining the model for every acquired sample. Moreover, it also means that labelling can only happen sequentially instead of in bulk. Thus, single-point acquisition functions were expanded to multi-point acquisition via acquisition batches in batch active learning. The most naive batch acquisition function selects the highest $K$ scoring points

$$a_s^{\text{batch}}(\mathcal{I}^{\text{train}}; K) \coloneqq \underset{I \subseteq \mathcal{I}^{\text{pool}}, |I|=K}{\arg\max} \sum_{i \in I} s(i; \mathcal{I}^{\text{train}}). \tag{8}$$

Maximizing this sum is equivalent to taking the top-k scoring points, which cannot account for the interactions between points in an acquisition batch because individual points are scored independently. For example, if the most informative point is duplicated in the pool set, all instances will be acquired, which is likely wasteful when we assume no label noise (see also Figure 1 in Kirsch et al. (2019)).

## 2.3 Batch Acquisition Functions

Some acquisition functions are explicitly designed for batch acquisition (Kirsch et al., 2019; Pinsler et al., 2019; Ash et al., 2020). They try to account for the interaction between points, which can improve performance relative to simply selecting the top-$K$ scoring points. However, existing methods can be computationally expensive. For example, BatchBALD rarely scales to acquisition sizes of more than 5–10 points due to its long runtime (Kirsch et al., 2019), as we evidence in Table 1.

**ACS-FW.** Pinsler et al. (2019) propose *"Active Bayesian CoreSets with Frank-Wolfe optimization"* which successively minimizes the distance between the (expected) loss on the pool set and the loss on a potential training set using a greedy Frank-Wolfe optimization in a Hilbert space. Either a posterior-weighted Fisher inner-product or a posterior-weighted loss inner-product between two samples are used. For non-linear models, random projections of the gradients speed up the Fisher inner-product.

**BADGE.** Ash et al. (2020) propose *"Batch Active learning by Diverse Gradient Embeddings"*: it motivates its batch selection approach using a k-Determinantal Point Process (Kulesza & Taskar, 2011) based on the (inner product) similarity matrix of the scores (gradients of the log loss) using hard pseudo-labels (the highest probability class according to the model's prediction) for each pool sample. See also Kirsch & Gal (2022) for a more detailed analysis. In practice, they use the intialization step of k-MEANS++ with Euclidian distances between the scores to select an acquisition batch. BADGE is also computationally expensive as we elaborate in Section 4.

**BatchBALD.** Kirsch et al. (2019) extend BALD to batch acquisition using the mutual information between the parameter distribution and the *joint* distribution of the predictions of multiple point in an acquistion batch: this mutual information is the expected information gain for a full acquistion batch:

$$s_{\text{BatchBALD}}(i_1, ..., i_K; \mathcal{I}^{\text{train}}) \coloneqq \mathrm{I}[Y_1, ..., Y_K; \Omega \mid X_1 = x_{i_1}, ..., X_K = x_{i_K}, \mathcal{D}^{\text{train}}]. \tag{9}$$

Kirsch et al. (2019) greedily construct an acquisition batch by iteratively selecting the next unlabelled pool point that maximizes the joint score with the already selected points. This is $1 - 1/e$-optimal as the expected information gain is submodular (Krause & Golovin, 2014). They note that their approach is computationally expensive, and they only consider acquisition batches of up to size 10.

# 3 Method

Selecting the top-$K$ points at acquisition step $t$ amounts to the assumption that the informativeness of these points is independent of each other. This leads to the pathology that if the most informative pool point is duplicated in the pool set, each instance would be selected (up to the acquisition size). This is clearly wrong.

Table 2: *Summary of stochastic acquisition variants.* Perturbing the scores $s_i$ themselves with $\epsilon_i \sim \text{Gumbel}(0; \beta^{-1})$ i.i.d. yields a softmax distribution. Log-scores result in a power distribution, with assumptions that are reasonable for active learning. Using the score-ranking, $r_i$ finally is a robustifying assumption. $\beta$ is included for completeness; we use $\beta := 1$ in our experiments—except for the ablation in §6.1.

| Perturbation | Distribution | Probability mass |
|---|---|---|
| $s_i + \epsilon_i$ | Softmax | $\propto \exp \beta s_i$ |
| $\log s_i + \epsilon_i$ | Power | $\propto s_i^{\beta}$ |
| $-\log r_i + \epsilon_i$ | Soft-rank | $\propto r_i^{-\beta}$ |

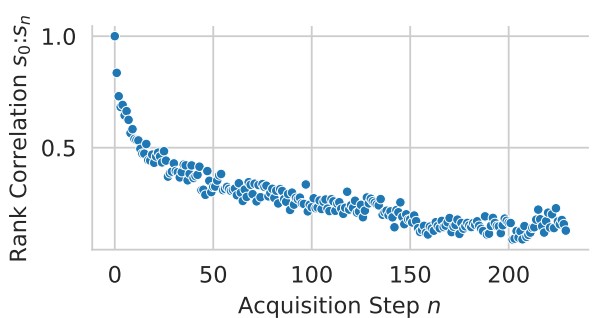

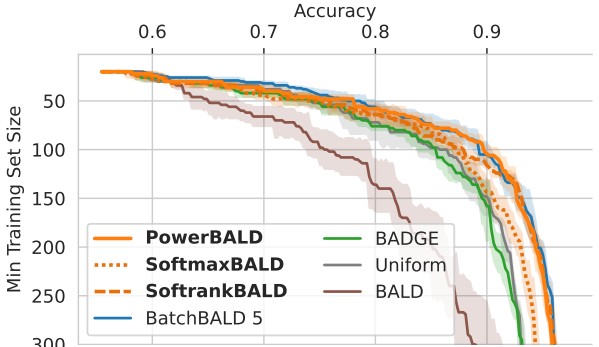

Figure 1: *Acquisition scores at individual acquisition step $t$ are only a loose proxy for later scores at $t + n$ (here: $t = 0$).* Specifically, the Spearman rank-correlation between acquisition scores on the zerot-th and $n$'th time-step falls with $n$. While top-$K$ acquisition incorrectly implicitly assumes the rank-correlation remains 1, stochastic acquisitions do not. Using Monte-Carlo Dropout BNN trained on MNIST at initial 20 points and 73% initial accuracy; score ranks computed over test set.

Figure 2: *Performance on Repeated-MNIST with 4 repetitions (5 trials).* **Up and to the right is better (↗).** PowerBALD outperforms (top-$K$) BALD and BADGE and is on par with BatchBALD. This is despite being orders of magnitude faster. Acquisition size: 10, except for BatchBALD (5). See Figure 10 in the appendix for an ablation study of BADGE's acquisition size, and Figure 11 for a comparison of all BALD variants at acquisition size 5.

**Top-$K$ Acquisition Pathologies.** Another way to see that this is wrong is to think step by step, that is to split the batch acquisition into multiple steps of size 1: We select the top pool sample by acquisition score and retrain the model once for each possible class label for this point. We then compute the averaged acquisition scores on the pool set given each of these models weighted by the original model's probability of each class label. We select the top pool sample using this new (averaged) score, and repeat the process, exponentially branching out as necessary. This is equivalent to the joint acquisition batch selection in BatchBALD (Kirsch et al., 2019):

$$\mathrm{I}[Y_1, ..., Y_K; \Omega \mid X_1 = x_{i_1}, ..., X_K = x_{i_K}, \mathcal{D}^{\text{train}}] = \tag{10}$$

$$= \sum_{j=1}^{K} \mathbb{E}_{\mathrm{P}(y_{i_1}, ..., y_{i_{j-1}} \mid x_{i_1}, ..., x_{i_{j-1}}, \mathcal{D}^{\text{train}})} \mathrm{I}[Y_j; \Omega \mid X_j = x_{i_j}, X_1 = x_{i_1}, Y_1 = y_{i_1}, ..., X_{i_{j-1}} = x_{i_{j-1}}, Y_1 = y_{i_{j-1}}, \mathcal{D}^{\text{train}}]$$

$$\neq \sum_{j=1}^{K} \mathrm{I}[Y_j; \Omega \mid X_j = x_{i_j}, \mathcal{D}^{\text{train}}] = \sum_{j=1}^{K} s_{\text{BALD}}(i_j; \mathcal{I}^{\text{train}}) \tag{11}$$

using the chain rule of the mutual information. We see that the informativeness of the samples will usually not be independent of each other. Of course, the acquisition scores for models trained with these additional points will be quite different from the first set of scores. After all, the purpose of active learning is to add the *most informative* points—those that will update the model the most. In contrast, selecting a top-$K$ batch using the same scores implicitly assumes that the score ranking will not change due to other points.

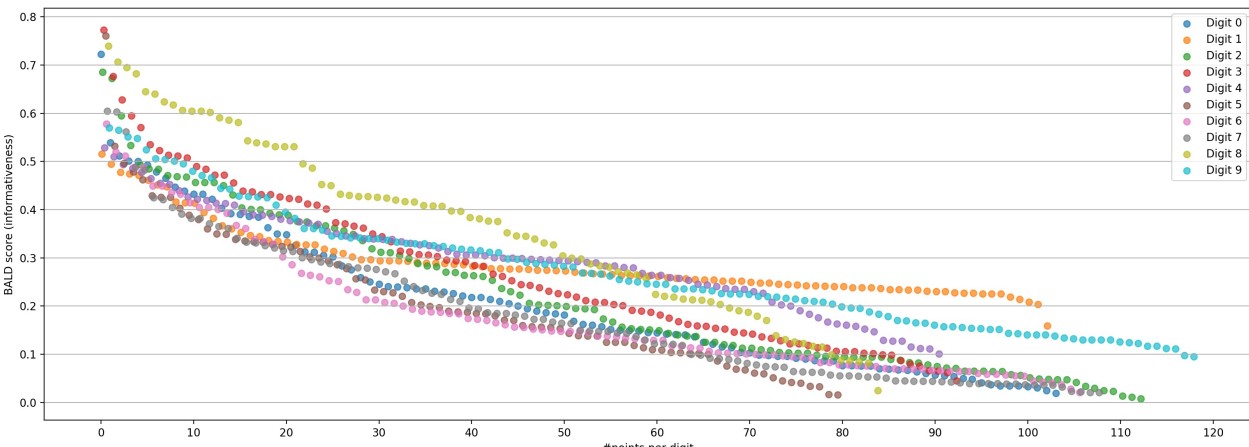

Figure 3: *BALD scores for 1000 randomly-chosen points from the MNIST dataset (handwritten digits).* The points are colour-coded by digit label and sorted by score. The model used for scoring has been trained to 90% accuracy first. If we were to pick the top scoring points (e.g. scores above 0.6), most of them would be 8s, even though we can assume that after acquiring the first couple of them, the model would consider them less informative than other available data. Points are slightly jittered on the x-axis by digit label to avoid overlaps.

We provide empirical confirmation in Figure 1 that, in fact, the ranking of acquisition scores at step $t$ and $t + K$ is decreasingly correlated as $K$ grows when we retrain the model for each acquired point. Figure 3 also illustrates this on MNIST. Moreover, as we will see in §6, this effect is the strongest for the most informative points.

Instead, we investigate the use of stochastic sampling as an alternative to top-$K$ acquisition, which implicitly acknowledges the uncertainty within the batch acquisition step using a simple noise process model governing how scores change. We motivate and investigate the theory behind this in §6, but given how simple the examined methods are, this theory only obstructs their simplicity. Specifically, we examine three simple stochastic extensions of single-sample scoring functions $s(i; \mathcal{I}^{\text{train}})$ that make slightly different assumptions. These methods are compatible with conventional active learning frameworks that typically take the top-$K$ highest scoring samples. For example, it is straightforward to adapt entropy, BALD, and other scoring functions for use with these extensions.

**Gumbel Noise.** These stochastic acquisition extensions assume that future scores differ from the current score by a perturbation. We model the noise distribution of this perturbation as the addition of Gumbel-distributed noise $\epsilon_i \sim \text{Gumbel}(0; 1)$, which is frequently used to model the distribution of extrema. At the same time, the choice of a Gumbel distribution for the noise is also one of mathematical convenience, in the spirit of a straightforward baseline: the maximum of sets of many other standard distributions, such as the Gaussian distribution, is not analytically tractable. On the other hand, taking the highest-scoring points from a distribution perturbed with Gumbel noise is equivalent to sampling from a softmax distribution[1] without replacement. We investigate the choice of Gumbel noise further in §6.

This follows from the *Gumbel-Max* trick (Gumbel, 1954; Maddison et al., 2014) and, more specifically, the Gumbel-Top-$K$ trick (Kool et al., 2019). We provide a short proof in appendix §B.2. Expanding on Maddison et al. (2014):

**Proposition 3.1.** *For scores $s_i$, $i \in \{1, \ldots, n\}$, and $k \leq n$ and $\beta > 0$, if we draw $\epsilon_i \sim \text{Gumbel}(0; \beta^{-1})$ independently, then $\arg\text{top}_k\{s_i + \epsilon_i\}_i$ is an (ordered) sample without replacement from the categorical distribution* $\text{Categorical}(\exp(\beta\,s_i)/\sum_j \exp(\beta\,s_j), i \in \{1, \ldots, n\})$.

$\beta \geq 0$ is a 'coldness' parameter. In the spirit of providing a simple and surprisingly effective baseline without hyperparameters, we fix $\beta := 1$ for our main experiments. However, to understand its effect, we examine

---

[1]Also known as Boltzmann/Gibbs distribution.

ablations of $\beta$ in §6.1. Overall, for $\beta \to \infty$, this distribution will converge towards top-$K$ acquisition; whereas for $\beta \to 0$, it will converge towards uniform acquisition.

We apply the perturbation to three quantities in the three sampling schemes: the scores themselves, the log scores, and the rank of the scores. Perturbing the log scores assumes that scores are non-negative and uninformative points should be avoided. Perturbing the ranks can be seen as a robustifying assumption that requires the relative scores to be reliable but allows the absolute scores to be unreliable. Table 2 summarizes the three stochastic acquisition variants with their associated sampling distributions, which make slightly different assumptions about the acquisition scores:

**Soft-Rank Acquisition.** This first variant only relies on the rank order of the scores and makes no assumptions on whether the acquisition scores are meaningful beyond that. It thus uses the *least* amount of information from the acquisition scores. It only requires the *relative score order* to be useful and ignores the *absolute score values.* If the absolute scores provide useful information, we would expect this method to perform worse than the variants below, which make use of the score values. As we will see, this is indeed sometimes the case.

Ranking the scores $s(i; \mathcal{I}^{\mathrm{train}})$ with descending ranks $\{r_i\}_{i \in \mathcal{I}^{\mathrm{pool}}}$ such that $s(r_i; \mathcal{I}^{\mathrm{train}}) \geq s(r_j; \mathcal{I}^{\mathrm{train}})$ for $r_i \leq r_j$ and smallest rank being 1, we sample index $i$ with probability $\mathrm{p}_{\mathrm{softrank}}(i) \propto r_i^{-\beta}$ with coldness $\beta$. This is invariant to the actual scores. We can draw $\epsilon_i \sim \mathrm{Gumbel}(0; \beta^{-1})$ and create a perturbed 'rank'

$$s^{\mathrm{softrank}}(i; \mathcal{I}^{\mathrm{train}}) := -\log r_i + \epsilon_i. \tag{12}$$

Following Proposition 3.1, taking the top-$K$ points from $s^{\mathrm{softrank}}$ is equivalent to sampling without replacement from the rank distribution $\mathrm{p}_{\mathrm{softrank}}(i)$.

**Softmax Acquisition.** The next simplest variant uses the actual scores instead of the ranks. Again, it perturbs the scores by a Gumbel-distributed random variable $\epsilon_i \sim \mathrm{Gumbel}(0; \beta^{-1})$

$$s^{\mathrm{softmax}}(i; \mathcal{I}^{\mathrm{train}}) := s(i; \mathcal{I}^{\mathrm{train}}) + \epsilon_i. \tag{13}$$

However, this makes no assumptions about the semantics of the absolute values of the scores: the softmax function is invariant to constants shifts. Hence, the sampling distribution will only depend on the *relative scores* and not their absolute value.

**Power Acquisition.** For many scoring functions, the scores are non-negative, and a score close to zero means that the sample is not informative in the sense that we do not expect it will improve the model—we do not want to sample it. This is the case with commonly used score functions such as BALD and entropy. BALD measures the expected information gain. When it is zero for a sample, we do not expect anything to be gained from acquiring a label for that sample. Similarly, entropy is upper-bounding BALD, and the same consideration applies. This assumption also holds for other scoring functions such as the standard deviation and variation ratios; see Appendix B.1. To take this into account, the last variant models the future log scores as perturbations of the current log score with Gumbel-distributed noise

$$s^{\mathrm{power}}(i; \mathcal{I}^{\mathrm{train}}) := \log s(i; \mathcal{I}^{\mathrm{train}}) + \epsilon_i. \tag{14}$$

By Proposition 3.1, this is equivalent to sampling from a power distribution

$$\mathrm{p}_{power}(i) \propto \left( \frac{1}{s(i; \mathcal{I}^{\mathrm{train}})} \right)^{-\beta}. \tag{15}$$

This may be seen by noting that $\exp(\beta \log s(i; \mathcal{I}^{\mathrm{train}})) = s(i; \mathcal{I}^{\mathrm{train}})^\beta$. Importantly, as scores $\to 0$, the (perturbed) log scores $\to -\infty$ and will have probability mass $\to 0$ assigned. This variant takes the absolute scores into account and avoids data points with score 0.

Given the above considerations, when using BALD, entropy, and other appropriate scoring functions, power acquisition is the most sensible. Thus, we expect it to work best. Indeed, we find this to be the case in the toy experiment on Repeated-MNIST (Kirsch et al., 2019) depicted in Figure 2. However, even soft-rank acquisition often works well in practice compared to top-$K$ acquisition; see also appendix §D for a more

in-depth comparison. In the rest of the main paper, we mostly focus on power acquisition. We include results for all methods in §C. In Appendix G, we present the simple implementation of the stochastic acquisition variants we use in our experiments.

## 4    Related Work

Researchers in active learning (Atlas et al., 1989; Settles, 2010) have identified the importance of *batch* acquisition as well as the failures of top-$K$ acquisition using straightforward extensions of single-sample methods in a range of settings including support vector machines (Campbell et al., 2000; Schohn & Cohn, 2000; Brinker, 2003; Hoi et al., 2006; Guo & Schuurmans, 2007), GMMs (Azimi et al., 2012), and neural networks (Sener & Savarese, 2018; Kirsch et al., 2019; Ash et al., 2020; Baykal et al., 2021).

Many of these methods aim to introduce structured diversity to batch acquisition that accounts for the *interaction* of the points acquired in the learning process. In most cases, the computational complexity scales poorly with the acquisition size ($K$) or pool size ($M$), for example because of the estimation of joint mutual information (Kirsch et al., 2019); the $\mathcal{O}(KM)$ complexity of using a k-means++ initialisation scheme (Ash et al., 2020), which approximates k-DPP-based batch active learning (Bıyık et al., 2019), or Frank-Wolfe optimization (Pinsler et al., 2019); or the $\mathcal{O}(M^2 \log M)$ complexity of methods based on $K$-centre coresets (Sener & Savarese, 2018) (although heuristics and continuous relaxations can improve this somewhat). In contrast, we examine simple and efficient stochastic strategies for adapting well-known single-sample acquisition functions to the batch setting. The proposed stochastic strategies are based on observing that acquisition scores would change as new points are added to the acquisition batch and modelling this difference for additional batch samples in the most naive way, using Gumbel noise. The presented stochastic extensions have the same complexity $\mathcal{O}(M \log K)$ as naive top-$K$ batch acquisition, yet outperform it, and they can perform on par with above more complex methods.

For multi-armed bandits, Soft Acquisition is also known as Boltzmann exploration (Cesa-Bianchi et al., 2017). It has also been shown that adding noise to the scores, specifically via Thompson sampling, is effective for choosing informative batches (Kalkanli & Özgür, 2021). Similarly, in reinforcement learning, stochastic prioritisation has been employed as *prioritized replay* (Schaul et al., 2016) which may be effective for reasons analogous to those motivating the approach examined in this work.

While stochastic sampling has not been extensively explored for acquisition in deep active learning, most recently it has been used as an auxiliary step in diversity-based active learning methods that rely on clustering as main mechanism (Ash et al., 2020; Citovsky et al., 2021). Kirsch et al. (2019) empirically find that additional noise in the acquisition scores seems to benefit batch acquisition but do not investigate further. Fredlund et al. (2010) suggest modeling single-point acquisition as sampling from a "*query density*" modulated by the (unknown) sample density p($x$) and analyze a binary classification toy problem. Farquhar et al. (2021) propose stochastic acquisition as part of de-biasing actively learned estimators.

Most relevant to this work, and building on Fredlund et al. (2010) and Farquhar et al. (2021), Zhan et al. (2022) propose a stochastic acquisition scheme that is asymptotically optimal. They normalize the acquisition scores via the softmax function to obtain a query density function for unlabeled samples and draw an acquisition batch from it, similar to SoftmaxEntropy. Their method aims to achieve asymptotic optimality for active learning processes by mitigating the impact of bias. In contrast, in this work, we examine multiple stochastic acquisition strategies based on score-based or rank-based distributions and apply these strategies to several single-sample acquisition functions, such as BALD and entropy (and standard deviation, variation ratios, see Figure 12); and we focus on active learning in a (Bayesian) deep learning setting. As such our empirical results and additional proposed strategies can be seen as complementary to their work.

Thus, while stochastic sampling is generally well-known within acquisition functions, to our knowledge, this work is the first[2] to investigate simple stochastic sampling methods entirely as alternatives to naive top-$K$ acquisition in (Bayesian) deep active learning and to compare them to more complex approaches in various settings.

---

[2]A workshop version was presented at ICML 2021, and the first submission of this work was concurrent to Zhan et al. (2022).

# 5 Experiments

In this section, we empirically verify that the presented stochastic acquisition methods (a) outperform top-$K$ acquisition and (b) are competitive with specially designed batch acquisition schemes like BADGE (Ash et al., 2020) and BatchBALD (Kirsch et al., 2019); and are vastly cheaper than these more complicated methods.

To demonstrate the seriousness of the possible weakness of recent batch acquisition methods, we use a range of datasets. These experiments show that the performance of the stochastic extensions is not dependent on the specific characteristics of any particular dataset. Our experiments include computer vision, natural language processing (NLP), and causal inference (in §6.1). We show that stochastic acquisition helps avoid selecting redundant samples on Repeated-MNIST (Kirsch et al., 2019), examine performance in active learning for computer vision on EMNIST (Cohen et al., 2017), MIO-TCD (Luo et al., 2018), Synbols (Lacoste et al., 2020), and CLINC-150 (Larson et al., 2019) for intent classification in NLP. MIO-TCD is especially close to real-world datasets in size and quality. In appendix §C.5, we further investigate edges cases using the Synbols dataset under different types of biases and noise, and in Appendix C.7, we also separately examine stochastic batch acquisition using last-layer MFVI models on CIFAR-10 (Krizhevsky et al., 2009), SVHN (Netzer et al., 2011), Repeated-MNIST, Fashion-MNIST (Xiao et al., 2017) and compare to ACS-FW (Pinsler et al., 2019).

In the main paper, we consider BALD as scoring function. We examine predictive entropy on many of these datasets in the appendix. Additionally, other scoring functions are examined on Repeated-MNIST in appendix §C.2.1. Overall, we observe similar results as for BALD.

For the sake of legible figures, we mainly focus on power acquisition in this section, as it fits BALD and entropy best: the scores are non-negative, and zero scores imply uninformative samples. We show that all three methods (power, softmax, softrank) can perform similarly in appendix §D.

We are not always able to compare to BADGE and BatchBALD because of computational limitations of those methods. BatchBALD is computationally infeasible for large acquisition sizes ($\geq 10$) because of time constraints, cf. Table 1. When possible, we use BatchBALD with acquisition size 5 as baseline. Note that this gives BatchBALD an advantage as it is known to perform better with smaller acquisition sizes (Kirsch et al., 2019). Similarly, BADGE ran out of memory for large dataset sizes, such as EMNIST 'ByMerge' with 814,255 examples, independently of the acquisition size.

Figures interpolate linearly between available points, and we show 95% confidence intervals.

**Experimental Setup & Compute.** We document the experimental setup and model architectures in detail in appendix §C.1. Our experiments used about 25,000 compute hours on Titan RTX GPUs.

**Runtime Measurements.** We emphasize that the stochastic acquisition strategies are much more computationally efficient compared to specialised batch-acquisition approaches like BADGE and BatchBALD. Runtimes, shown in Table 1, are essentially identical for top-$K$ and the stochastic versions. Both are orders of magnitude faster than BADGE and BatchBALD even for small batches. Unlike those methods, stochastic acquisition scales *linearly* in pool size and *logarithmically* in acquisition size. Runtime numbers do not include the cost of retraining models (identical in each case). The runtimes for top-$K$ and stochastic acquisition appear constant over $K$ because the execution time is dominated by fixed-cost memory operations. The synthetic dataset used for benchmarking has 4,096 features, 10 classes, and 10,000 pool points.

**Repeated-MNIST.** Repeated-MNIST (Kirsch et al., 2019) duplicates MNIST a specified number of times and adds Gaussian noise to prevent perfect duplicates. Redundant data are incredibly common in industrial applications but are usually removed from standard benchmark datasets. The controlled redundancies in the dataset allow us to showcase pathologies in batch acquisition methods. We use an acquisition size of 10 and 4 dataset repetitions.

Figure 2 shows that PowerBALD outperforms top-$K$ BALD. While much cheaper computationally, cf. Table 1, PowerBALD also outperforms BADGE and even performs on par with BatchBALD. We use an acquisition size of 10 for all methods, except for BatchBALD, for which we use an acquisition size of 5, as explained above. Note that BatchBALD performs better for smaller acquisition sizes while BADGE (counterintuitively) can perform better for larger ones; see Figure 10 in the appendix for an ablation. BatchBALD, BALD, and

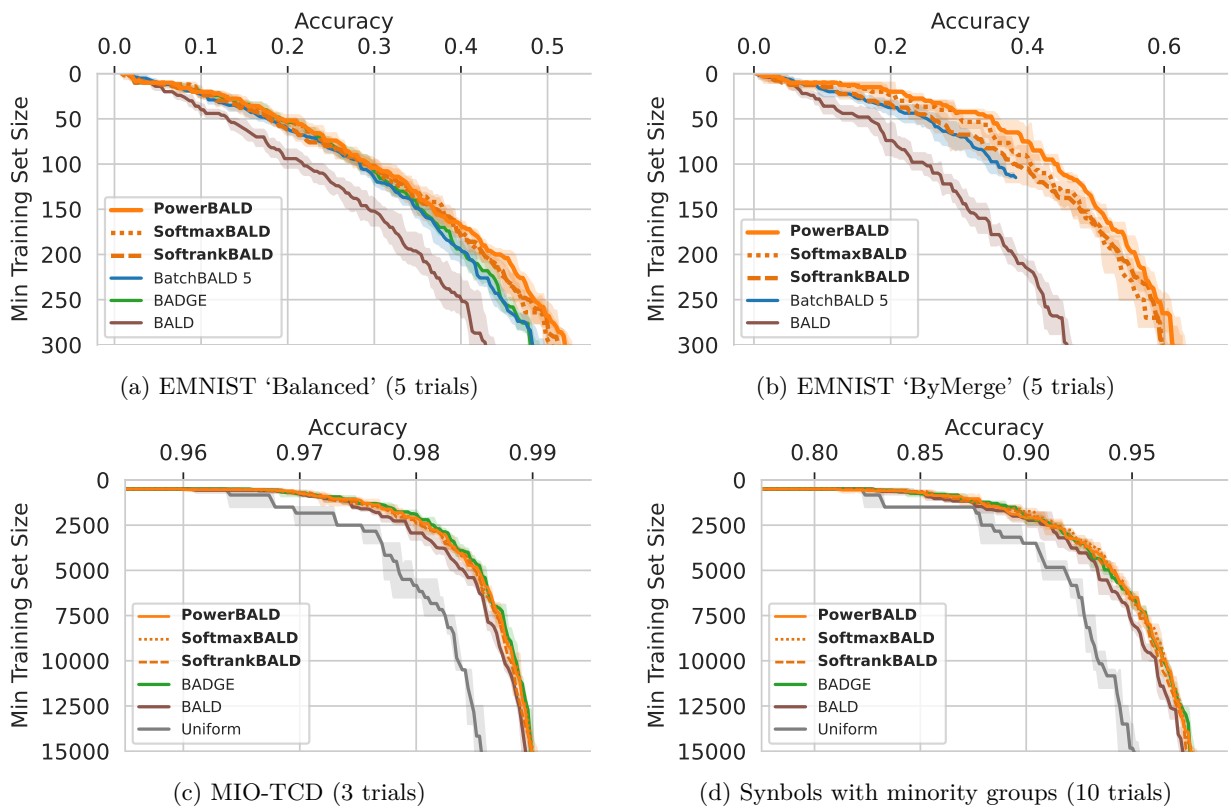

Figure 4: *Performance on various datasets.* BatchBALD took infeasibly long on these datasets & acquisition sizes. **(a)** *EMNIST 'Balanced':* On 132k samples, PowerBALD (acq. size 10) outperforms BatchBALD (acq. size 5) and BADGE (acq. size 10). **(b)** *EMNIST 'ByMerge':* On 814k samples, PowerBALD (acq. size 10) outperforms BatchBALD (acq. size 5). BADGE (not shown) OOM'ed, and BatchBALD took > 12 days for 115 acquisitions. **(c)** *MIO-TCD:* PowerBALD performs better than BALD and on par with BADGE (all acq. size 100). **(d)** *Synbols with minority groups:* PowerBALD performs on par with BADGE (all acq. size 100).

the stochastic variants all become equivalent for acquisition size 1 when points are acquired individually, which performs best Kirsch et al. (2019).

**Computer Vision: EMNIST.** EMNIST (Cohen et al., 2017) contains handwritten digits and letters and comes with several splits: we examine the 'Balanced' split with 131,600 samples in Figure 4a[3] and the 'ByMerge' split with 814,255 samples in Figure 4b. Both have 47 classes. We use an acquisition size of 5 for BatchBALD, and 10 otherwise.

We see that the stochastic methods outperform BatchBALD on it and both BADGE and BatchBALD on 'Balanced' (Figure 4a). They do not have any issues with the huge pool set in 'ByMerge' (Figure 4b). In the appendix, Figures 27 and 28 show results for all three stochastic extensions, and Figure 17 shows an ablation of different acquisition sizes for BADGE. For 'ByMerge', BADGE ran out of memory on our machines, and BatchBALD took more than 12 days for 115 acquisitions when we halted execution.

**Computer Vision: MIO-TCD.** The Miovision Traffic Camera Dataset (MIO-TCD) (Luo et al., 2018) is a vehicle classification and localisation dataset with 648,959 images designed to exhibit realistic data characteristics like class imbalance, duplicate data, compression artefacts, varying resolution (between 100 and 2,000 pixels), and uninformative examples; see Figure 9 in the appendix. As depicted in Figure 4c, PowerBALD performs better than BALD and essentially matches BADGE despite being much cheaper to compute. We use an acquisition size of 100 for all methods.

---

[3]This result exactly reproduces BatchBALD's trajectory in Figure 7 from Kirsch et al. (2019).

**Computer Vision: Synbols.** Synbols (Lacoste et al., 2020) is a character dataset generator which can demonstrate the behaviour of batch active learning under various edge cases (Lacoste et al., 2020; Branchaud-Charron et al., 2021). In Figure 4d, we evaluate PowerBALD on a dataset with minority character types and colours. PowerBALD outperforms BALD and matches BADGE. Further details as well as an examination of the 'spurious correlation' and 'missing synbols' edge cases (Lacoste et al., 2020; Branchaud-Charron et al., 2021) can be found in appendix §C.5.

**Natural Language Processing: CLINC-150.** We perform intent classification on CLINC-150 (Larson et al., 2019), which contains 150 intent classes plus an out-of-scope class. This setting captures data seen in production for chatbots. We fine-tune a pretrained DistilBERT model from HuggingFace (Wolf et al., 2020) on CLINC-150 for 5 epochs with Adam as optimiser. In appendix §C.6, we see that PowerEntropy shows strong performance: it performs better than Entropy and almost on par with BADGE. This demonstrates that our technique is domain independent and can be easily reused for other tasks.

**MFVI Last-Layer Comparison with ACS-FW**. In Appendix C.7, we provide a comparison of the proposed stochastic acquisition functions with BALD and ACS-FW (Pinsler et al., 2019) in a Bayesian last-layer setting using variational inference with a mean-field Gaussian approximation (instead of Monte-Carlo dropout). We find that for smaller acquisition sizes, the stochastic acquisition functions also outperform BALD on Fashion-MNIST and Repeated-MNIST. For larger acquisition sizes on CIFAR-10 and SVHN, this is not the case, however: no single method seems to perform better than the others. However, we find that ACW-FW overall takes much longer to run than stochastic and top-$K$ acquisition (Figure 25). We leave a more thorough investigation of this setting to future work.

**In Summary.** We have verified that stochastic acquisition functions outperform top-$K$ batch acquisition in several settings and perform on par with more complex methods such as BADGE or BatchBALD. Moreover, we refer the reader to Jesson et al. (2021), Murray et al. (2021), Tigas et al. (2022), Holmes et al. (2022), Malik et al. (2023), and Rubashevskii et al. (2023) for additional works that use the proposed stochastic acquisition functions in this paper and provide further empirical validation.

## 6  Further Investigations

In this section, we examine and validate assumptions about the underlying score dynamics by examining the scores across acquisitions. We further hypothesise about when top-$K$ acquisition is the most detrimental to active learning.

**Why Gumbel Noise?** Intuitively, to select the $k$-th point in the acquisition batch, we want to take into account how much additional information (increase in acquisition scores) the still-to-be-selected additional $K - k$ points will provide. As such we want to model the maximum over all possible additional candidate points that are still to be selected to complete the acquisition batch. Empirically, acquisition scores are similar to a truncated exponential distribution ('80/20' rule) as visualized in Figures 3 and 7c. Note that this is a rough approximation—we do not claim that the distribution of acquisition scores really truncated exponential. A sum of i.i.d. exponential variables follows the Erlang distribution (which is a special case of the Gamma distribution) and has an exponential tail. We know that the maximum of a set of i.i.d. random variables that follow an exponential distribution or a distribution which has an exponential tail is known to be well approximated by a Gumbel distribution in the sample limit (Gumbel, 1954)[4], and thus the maximum over sums of such random variables also follows a Gumbel distribution. Concretely, we can use the following result from Garg et al. (2023):

**Theorem 6.1** (Extreme Value Theorem (EVT) (Mood, 1950; Fisher & Tippett, 1928)). *For i.i.d. random variables $X_1, \ldots, X_n \sim f_X$, with exponential tails, $\lim_{n \to \infty} \max_i (X_i)$ follows the Gumbel (GEV -1) distribution. Furthermore, $\mathcal{G}$ is max-stable, i.e. if $X_i \sim \mathcal{G}$, then $\max_i (X_i) \sim \mathcal{G}$ holds.*

However, it is unlikely that the increase in acquisition scores can be modeled well as i.i.d exponential random variables with the *same* rate, but the Gumbel approximation also seems to hold empirically for the hypoexponential distribution which is a sum of exponential distributions with *different* rates. We do not have a proof for this, but we present a numerical simulation in appendix §H.

---

[4]See also the following Math StackExchange thread.

Overall, this motivates us to use a Gumbel distribution as a simple model for the increase in acquisition scores the still-to-be-selected additional $K - k$ points will provide under the modelling assumption that the increase in acquisition scores at each step is exponentially distributed with *different* rates at each step. Zhan et al. (2022) provide a different analysis for the use of Gumbel noise in the context of active learning as we relate in §4.

**Acquisition Asymptotics of Bayesian Models.** For well-specified and well-defined Bayesian parametric models, the posterior distribution of the model parameters converges to the true parameters as the number of data points increases (Van der Vaart, 2000).

For such models and assuming that the predictions are independent given the model parameters, the total correlation between the predictions decreases as the number of training points increases, as the posterior distribution of the model parameters becomes more concentrated around the true parameters:

$$\mathrm{TC}[Y_1, \ldots, Y_K \mid x_1, \ldots, x_K, \mathcal{D}^{\mathrm{train}}] \to 0 \quad \text{as} \quad |\mathcal{D}^{\mathrm{train}}| \to \infty. \tag{16}$$

This can be proved by noting that in the finite data limit, the posterior parameter distribution converges to the true model parameters, and the marginal distribution then factorizes. This means that the predictions become more independent as the number of training points increases and fully independent in the infinite data limit.

The total correlation is defined as:

$$\mathrm{TC}[Y_1, \ldots, Y_K \mid x_1, \ldots, x_K, \mathcal{D}^{\mathrm{train}}] := \underbrace{\sum_i \mathrm{H}[Y_i \mid x_i, \mathcal{D}^{\mathrm{train}}]}_{\text{top-}K \text{ Entropy}} - \underbrace{\mathrm{H}[Y_1, \ldots, Y_K \mid x_1, \ldots, x_K, \mathcal{D}^{\mathrm{train}}]}_{\text{`Batch Entropy'}}., \tag{17}$$

We can also write the total correlation as difference between top-$K$ BALD and BatchBALD:

$$\mathrm{TC}[Y_1, \ldots, Y_K \mid x_1, \ldots, x_K, \mathcal{D}^{\mathrm{train}}] = \underbrace{\sum_i \mathrm{I}[Y_i; \Omega \mid x_i, \mathcal{D}^{\mathrm{train}}]}_{\text{top-}K \text{ BALD}} - \underbrace{\mathrm{I}[Y_1, \ldots, Y_K; \Omega \mid x_1, \ldots, x_K, \mathcal{D}^{\mathrm{train}}]}_{\text{BatchBALD}}. \tag{18}$$

As the total correlation converges to 0, the top-$K$ BALD term (first term) becomes equal to the BatchBALD term (the second term on the right side), and the same happens for top-$K$ entropy and 'BatchEntropy', which we could similarly define.

Thus, for well-specified and well-defined Bayesian parametric models, the top-$K$ acquisition functions will eventually become equivalent to the BatchBALD and 'BatchEntropy' acquisition functions as the number of training points increases. This tells us that top-$K$ acquisition is the most detrimental to active learning in the earlier stages of learning, when the total correlation between the predictions is still high. This is consistent with our empirical results below ('Increasing Top-$K$ Analysis').

At the same time, as the number of training points increases and the model parameters concentrate, the expected information gain (BALD) also decreases. The mutual information with a deterministic variable is always 0, and thus:

$$\mathrm{I}[Y; \Omega \mid x, \mathcal{D}^{\mathrm{train}}] \to 0 \quad \text{as} \quad |\mathcal{D}^{\mathrm{train}}| \to \infty. \tag{19}$$

This asymptotic behavior is a trivial but important result, as it tells us that the expected information gain (BALD) will eventually become uninformative as the number of training points increases and no better than random acquisition, and the important question is: when? Given that we only have noisy estimators, this determines until when active learning is of use compared to random acquisition.

Many different active learning methods that are considered non-Bayesian nevertheless approximate the expected information gain or the expected predictive information (Kirsch & Gal, 2022; Smith et al., 2023), which is an expected total correlation. Hence, the considerations apply to those methods, too.

Finally, we observe that estimators such as BatchBALD, which utilize Monte-Carlo samples for parameter approximations, are inherently limited by the logarithm of the total number $M$ of these samples, $\log M$.

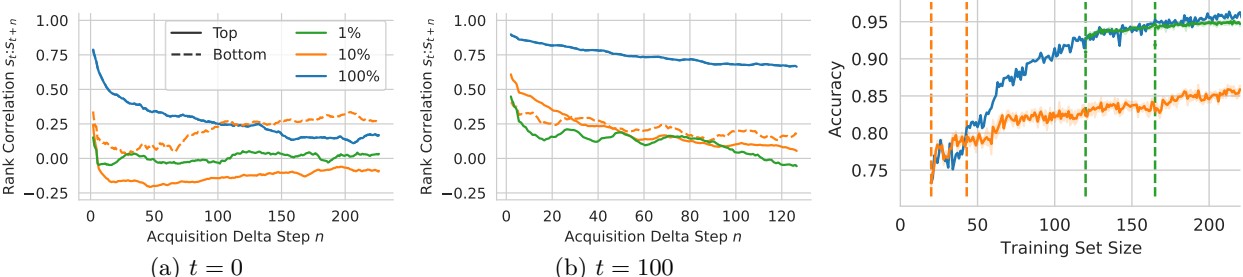

(a) $t = 0$        (b) $t = 100$

Figure 5: *Rank correlations for BALD scores on MNIST between the* Figure 6: *Top-K acquisition hurts initial scores and later scores of the top- or bottom-scoring 1%, 10% and less later in training (BALD on 100% of test points (smoothed with a size-10 Parzen window).* Rank- *MNIST).* At $t \in \{20, 100\}$ (blue), *orders decorrelate faster for the most informative samples and in the* we keep acquiring samples using the *early stages of training. The top scorers' ranks anti-correlate after* BALD scores from those two steps. *roughly 40 (100) acquisitions unlike the bottom ones. Later in training,* At $t = 20$ (orange), the model per-*the acquisition scores stay more strongly correlated. This suggests the* forms well for $\approx 20$ acquisitions; at *acquisition size could be increased later in training.* $t = 120$ (green), for $\approx 50$; see §6.

This constraint implies that their informativeness can diminish rapidly. Concretely, consider an empirical estimator $\hat{I}[\cdot; \Omega]$ built using Monte-Carlo samples $\omega_1, \ldots, \omega_k$. This is analogous to computing the exact mutual information $I[\cdot; \hat{\Omega}]$ with the 'empirical' random variable $\hat{\Omega}$, which uniformly samples from $\omega_1, \ldots, \omega_k$. Given that the discrete mutual information is restricted by the entropy of its terms, we have:

$$\hat{I}[\cdot; \Omega] = I[\cdot; \hat{\Omega}] \leq H[\hat{\Omega}] = \log M.$$

For instance, BatchBALD employs a greedy approach to select the $t$-th acquisition sample in its batch. It does this by maximizing the empirical $\hat{I}[Y; \Omega \mid x, Y_{t-1}, x_{t-1}, \ldots, Y_1, x_1 \mathcal{D}^{\text{train}}]$ for the subsequent candidate samples denoted by $x$. We can represent this relationship as:

$$\log M \geq \hat{I}[Y_1, \ldots, Y_K; \Omega \mid x_1, \ldots, x_K, \mathcal{D}^{\text{train}}] = \sum_{i=1}^{K} \hat{I}[Y; \Omega \mid x, Y_y, x_y, \ldots, Y_1, x_1 \mathcal{D}^{\text{train}}].$$

From the above equation, as $K$ grows, the estimated $\hat{I}[Y_K; \Omega \mid x_K, Y_{K-1}, x_{K-1}, \ldots, Y_1, x_1 \mathcal{D}^{\text{train}}]$ approaches zero since it is restricted by $\log M$. For a scenario with $M = 100$ parameter samples (leading to $\log_{10} M = 2$), BatchBALD might rapidly lose its informativeness after just two acquisitions in a classification scenario involving 10 categories. This situation arises if the pool set contains a minimum of two highly diverse, that is uncorrelated, data points with maximum disagreement.

**Rank Correlations Across Acquisitions.** In Section 3, we made the following assumptions: (1) the acquisition scores $s_t$ at step $t$ are a proxy for scores $s_{t'}$ at step $t' > t$; (2) the larger $t' - t$ is, the worse a proxy $s_t$ is for $s'_t$; (3) this effect is the largest for the most informative points.

We demonstrate these empirically by examining the Spearman rank correlation between scores during acquisition. Specifically, we train a model for $n$ steps using BALD as single-point acquisition function. We compare the rank order at each step to the starting rank order at step $t$. To denoise the rankings across $n$, we smooth the rank correlations with a Parzen window of size 10 and to reduce the effect of noise to the rank order, we round all scores to 2 decimal places. This especially removes unimportant rank changes for points with low scores around 0.

Figure 1 shows that acquisition scores become less correlated as more points are acquired. Figure 5a shows this in more detail for the top and bottom 1%, 10% or 100% of scorers of the test set across acquisitions starting at step $t = 0$ for a model initialised with 20 points. The top-10% scoring points (solid green) quickly become uncorrelated across acquisitions and even become *anti-correlated*. In contrast, the points overall (solid blue) correlate well over time (although they have a much weaker training signal on average). This result supports all three of our hypotheses.

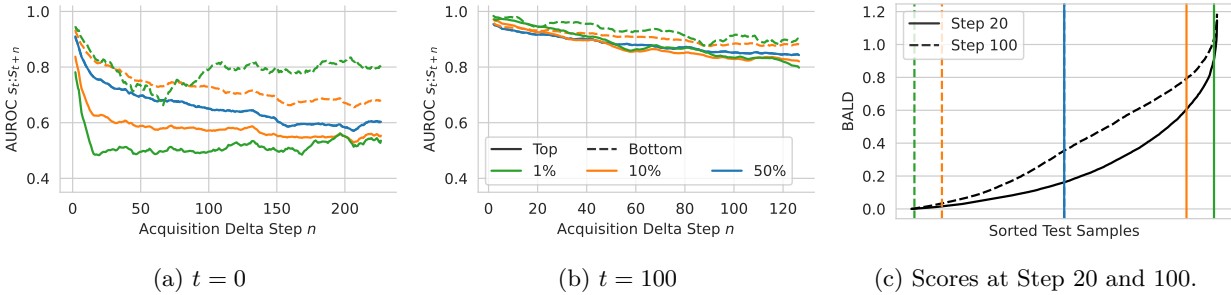

(a) $t = 0$       (b) $t = 100$       (c) Scores at Step 20 and 100.

Figure 7: *AUROCs for BALD scores on MNIST between the initial scores and later scores of the top- or bottom-scoring 1%, 10% and 50% of test points (smoothed with a size-10 Parzen window).* AUROC between original points as 'ground truth' and later scores as predictors. This is equivalent to the probability that the acquisition score at $n$ for a point in $t = 0$'s top or bottom 1%, etc. is larger than points outside. This tells us how likely other points outside the batch have higher acquisition scores. This ignores the ranking of points otherwise. **(a, b)** Points in the top quantiles are superseded by other points in the top quantiles in the later acquisitions to a large degree. This is much more pronounced early in the training than later. The bottom quantiles are more stable. **(c)** The overall score distributions at steps $t = 0, 100$ are visualized and the relevant top and bottom quantiles are marked.

At the same time, we see that as training progresses, and we converge towards the best model, the order of scores becomes more stable across acquisitions. In Figure 5b the model begins with 120 points ($t = 100$), rather than 20 ($t = 0$). Here, the most informative points are less likely to change their rank—even the top-1% ranks do not become *anti-correlated*, only decorrelated. Thus, we hypothesise that further in training, we might be able to choose larger $K$.

We do not display the correlations for bottom 1% of scorers as their scores are close to 0 throughout training and thus noisy and uninformative, see also Figure 7c for this.

Overall, this analysis can be confounded by noisy samples and swaps in rank order between samples with similar scores that are not meaningful in the sense that they would not influence batch acquisition.

Thus, to provide a different analysis, we also consider the more direct question in Figure 7 of how likely other samples have higher acquisition scores at $t + n$ than the top samples from $t$ for different quantiles (1%, 10%, 50%) of the test set. As a sanity check, we also examine the bottom quantiles. This is equivalent to computing the *AUROC* between the original points as 'ground truth' and later scores as predictors: for acquisition step n, the AUROC is $p(S_n^{t,\text{top/bottom } p\%} \lessgtr S_n^{t,\text{bottom/top } 1-p\%})$ with $S_n^{t,\text{top/bottom } p\%} := \{s_{t+n,i} : s_{t,i} \in \text{top/bottom } p\% \text{ of } \{s_t, j\}\}$. Specifically, we set up a binary classification with the top or bottom 1%, 10% or 50% of the test set as positive and the rest as negative. This again helps us quantify how much the scores meaningfully change across acquisition steps. These results match the previous ones and provide another validation for the mentioned assumptions.

**Increasing Top-$K$ Analysis.** Another way to investigate the effect of top-$K$ selection is to freeze the acquisition scores during training and then continue single-point 'active learning' as if those were the correct scores. Comparing this to the performance of regular active learning with updated single-point scores allows us to examine how well earlier scores perform as proxies for later scores. We perform this toy experiment on MNIST, showing that freezing scores early on greatly harms performance while doing it later has only a small effect (Figure 6). For frozen scores at a training set size of 20 (73% accuracy, $t = 0$), the accuracy matches single-acquisition BALD up to a training set size of roughly 40 (dashed orange lines) before diverging to a lower level. But when freezing the scores of a more accurate model, at a training set size of 120 labels (93% accuracy, $t = 100$), selecting the next fifty points according to those frozen scores performs indistinguishably from step-by-step acquisition (dashed green lines). This result shows that top-$K$ acquisition hurts less later in training but can negatively affect performance at the beginning of training.

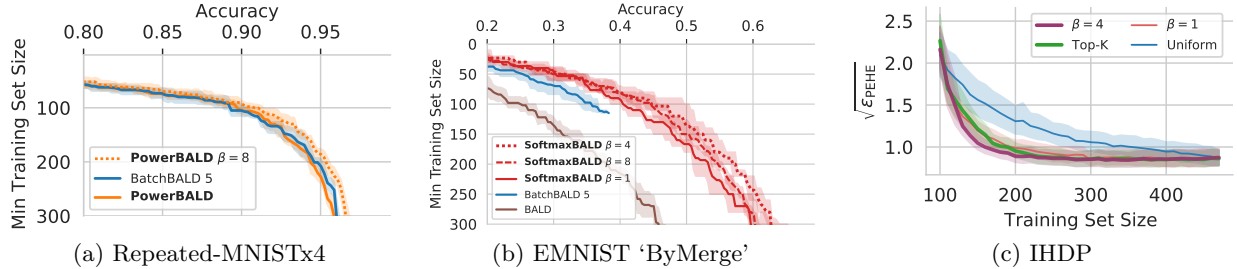

(a) Repeated-MNISTx4    (b) EMNIST 'ByMerge'    (c) IHDP

Figure 8: *Effect of changing $\beta$.* (**a**) *Repeated-MNISTx4 (5 trials):* PowerBALD outperforms BatchBALD for $\beta = 8$. (**b**) *EMNIST 'ByMerge' (5 trials):* SoftmaxBALD for $\beta = 4$ performs best. (**c**) *IHDP (400 trials):* At high temperature ($\beta = 0.1$), CausalBALD with power acquisition is like random acquisition. As the temperature decreases, the performance improves (lower $\sqrt{\epsilon_{\mathrm{PEHE}}}$), surpassing top-$K$ acquisition. Both experiments use an acquisition size of 10.

These observations lead us to ask whether we could dynamically change the acquisition size: with smaller acquisition batches at the beginning and larger ones towards the end of active learning. We leave the exploration of this for future work.

## 6.1 Ablation: Changing $\beta$

So far, we have set $\beta = 1$ in the spirit of examining a simple baseline without additional hyperparameters. The results above show that this already works well and matches the performance of much more expensive methods, raising questions about their value. In addition, however, tuning $\beta$ may be able to further improve performance. In the following, we show that other values of $\beta$ can yield even higher performance on Repeated-MNIST, EMNIST, and when estimating causal treatment effects; we provide many additional results in Appendix F.

**Repeated-MNIST & EMNIST.** In Figure 8a, we see that for PowerBALD the best-performing value, $\beta = 8$,even outperforms BatchBALD. Similarly, tuning $\beta$ can improve the performance of other strategies as well as we see for SoftmaxBALD on EMNIST, where $\beta = 4$ performs better than $\beta = 8$ and $\beta = 1$, showing that the best $\beta$ depends on the dataset.

**Causal Treatment Effects: Infant Health Development Programme.** Active learning for Conditional Average Treatment Effect (CATE) estimation Heckman et al. (1997; 1998); Hahn (1998); Abrevaya et al. (2015) on data from the Infant Health and Development Program (IHDP) estimates the causal effect of treatments on an infant's health from observational data. Statistical estimands of the CATE are obtainable from observational data under certain assumptions. Jesson et al. (2021) show how to use active learning to acquire data for label-efficient estimation. Among other subtleties, this prioritises the data for which matched treated/untreated pairs are available.

We follow the experiments of Jesson et al. (2021) on both synthetic data and the semi-synthetic IHDP dataset (Hill, 2011), a commonly used benchmark for causal effects estimation. In Figure 8c we show that power acquisition performs significantly better than both top-$K$ and uniform acquisition, using an acquisition size of 10 in all cases with further. We provide additional results on semi-synthetic data in appendix §F.3. Note that methods such as BADGE and BatchBALD are not well-defined for causal-effect estimation, while stochastic batch acquisition remains applicable and is effective when fine-tuning $\beta$.

Performance on these tasks is measured using the expected *Precision in Estimation of Heterogeneous Effect (PEHE)* (Hill, 2011) such that $\sqrt{\epsilon_{\mathrm{PEHE}}} = \sqrt{\mathbb{E}[(\widetilde{\tau}(\mathbf{X}) - \tau(\mathbf{X}))^2]}$ (Shalit et al., 2017) where $\widetilde{\tau}$ is the estimated CATE and $\tau$ is CATE (i.e. a form of RMSE).

**Limitations.** Although we highlight the possibility for future work to adapt $\beta$ to specific datasets or score functions, our aim is not to offer a practical recipe for this to practitioners. Our focus is on showing how even the simplest form of stochastic acquisition already raises questions for some recent more complex methods.

## 7 Discussion & Conclusion

Our experiments demonstrate that the stochastic sampling approach we have examined is orders of magnitude faster than sophisticated batch-acquisition strategies like BADGE and BatchBALD while retaining comparable performance in settings across computer vision, NLP, and causal inference. Compared to the flawed top-$K$ batch acquisition heuristic, it is never worse: we see no reason to continue using top-$K$ acquisition.

Importantly, our work raises serious questions about these current methods. If they fail to outperform such a simple baseline in a wide range of settings, do they model the interaction between points sufficiently well? If so, are the scores themselves unreliable? We call on future work in batch active learning to at least demonstrate that it can outperform simple stochastic batch acquisition strategies.

At the same time, this opens doors for improved methods. Although we only put forward a naive model due its computational and mathematical simplicity, future work can explore more sophisticated modelling of the predicted score changes that take the current model and dataset into account. In its simplest form, this might mean choosing the $\beta$ hyperparameter of the acquisition distribution based on the dataset and adapting it online. Our experiments also highlight that the acquisition size could be adapted dynamically, with larger batch sizes acceptable in later acquisition steps.

## Acknowledgements

The authors would like to thank their anonymous TMLR reviewers for their kind, constructive and helpful feedback during the review process, which has significantly improved this work. We would also like to thank Freddie Bickford Smith and Tom Rainforth as well as the members of OATML in general for their feedback at various stages of the project. SF is supported by the EPSRC via the Centre for Doctoral Training in Cybersecurity at the University of Oxford as well as Christ Church, University of Oxford. AK is supported by the UK EPSRC CDT in Autonomous Intelligent Machines and Systems (grant reference EP/L015897/1).

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

## A  Ethical impact

We do not foresee any ethical risks related to this work. Insofar as the sampling methods we have examined reduce computational costs, applications might benefit from reduced resource consumption. The methods appear to be as good or better than alternatives on evaluations examining the ability to learn from data with under-represented groups and on evaluations that measure the difference between performance for the most- and least-represented groups, which may aid algorithmic fairness (see C.5).

## B  Method

### B.1  Other scoring functions

Following Gal et al. (2017), we also examine using variation ratios (least confidence) and standard deviation as scoring functions.

**Variation Ratio.** Also known as *least confidence*, the variation-ratios is the complement of the least-confident class prediction:

$$s_{\text{variation-ratios}}(i; \mathcal{I}^{\text{train}}) := 1 - \max_y \mathrm{p}(y \mid X = x_i). \tag{20}$$

This scoring function is non-negative and a score of 0 means that the sample is uninformative: a score of 0 means that the respective prediction is one-hot, which means that the expected information gain is also 0 as can be easily verified. Thus, variation ratios matches the intuitions behind power acquisition.

**Standard Deviation.** The standard deviation score function measures the sum of the class probability deviations and is closely related to the BALD scores:

$$s_{\text{std-dev}}(i; \mathcal{I}^{\text{train}}) := \sum_y \sqrt{\mathrm{Var}_{\mathrm{p}(\omega)}[\mathrm{p}(y \mid X = x_i, \omega)]}. \tag{21}$$

This scoring function is also non-negative, and no variance for the predictions implies a zero expected information gain and thus an uninformative sample. Thus, the standard deviation should also perform well with power acquisition.

### B.2  Proof of Proposition 3.1

First, we remind the reader that a random variable $G$ is Gumbel distributed $G \sim \text{Gumbel}(\mu; \beta)$ when its cumulative distribution function follows $\mathrm{p}(G \le g) = \exp(-\exp(-\frac{g-\mu}{\beta}))$.

Furthermore, the Gumbel distribution is closed under translation and positive scaling:

**Lemma B.1.** *Let $G \sim Gumbel(\mu; \beta)$ be a Gumbel distributed random variable, then:*

$$\alpha G + d \sim Gumbel(d + \alpha\mu; \alpha\beta). \tag{22}$$

*Proof.* We have $\mathrm{p}(\alpha G + d \le x) = \mathrm{p}(G \le \frac{x-d}{\alpha})$. Thus, we have:

$$\mathrm{p}(\alpha G + d \le x) = \exp(-\exp(-\frac{\frac{x-d}{\alpha} - \mu}{\beta})) \tag{23}$$

$$= \exp(-\exp(-\frac{x - (d + \alpha\mu)}{\alpha\beta})) \tag{24}$$

$$\Leftrightarrow \alpha G + d \sim \text{Gumbel}(d + \alpha\mu; \alpha\beta). \tag{25}$$

$\square$

We can then easily prove Proposition 3.1 using Theorem 1 from Kool et al. (2019), which we present it here slightly reformulated to fit our notation:

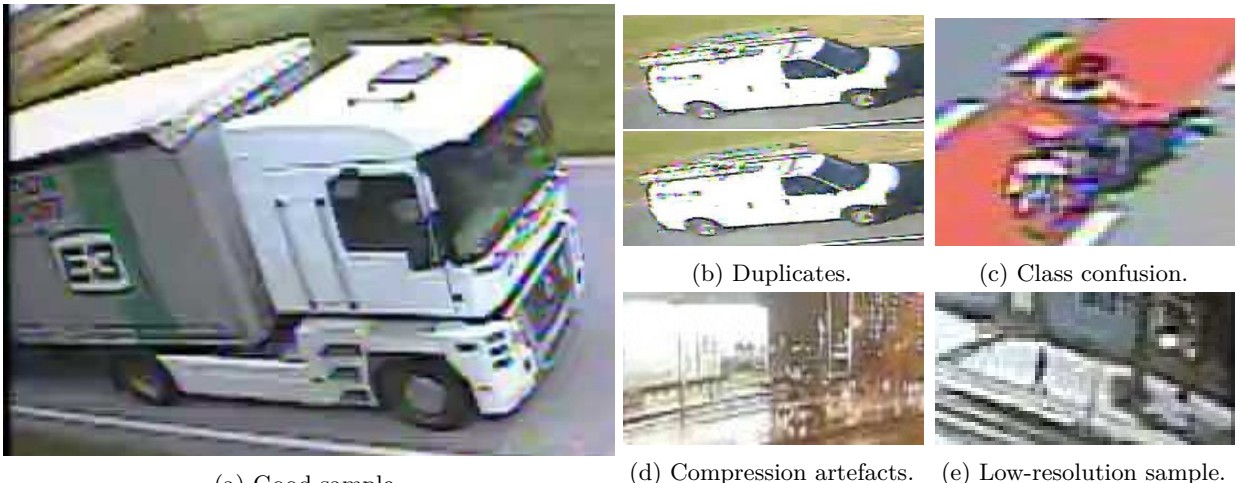

(a) Good sample.

(b) Duplicates.

(c) Class confusion.

(d) Compression artefacts.

(e) Low-resolution sample.

Figure 9: *MIO-TCD Dataset* is designed to include common artifacts from production data. The size and quality of the images vary greatly between crops; from high-quality cameras on sunny days to low-quality cameras at night. (a) shows an example of clean samples that can be clearly assigned to a class. (b)(c)(d) and (e) show the different categories of noise. (b) shows an example of duplicates that exist in the dataset. (c) is a good example where the assigned class is subject to interpretation: motorcycle or bicycle? (d) is a sample with heavy compression artefacts and (e) is an example of samples with low resolution which again is considered a hard example to learn for the model.

**Lemma B.2.** *For $k \leq n$, let $I_1^*, \ldots, I_k^* = \arg \operatorname{top}_k \{s_i + \epsilon_i\}_i$ with $\epsilon_i \sim \text{Gumbel}(0; 1)$, i.i.d.. Then $I_1^*, \ldots, I_k^*$ is an (ordered) sample without replacement from the Categorical$\left( \frac{\exp s_i}{\sum_{j \in n} \exp s_j}, i \in \{1, \ldots, n\} \right)$ distribution, e.g. for a realization $i_1^*, \ldots, i_k^*$ it holds that*

$$P\left(I_1^* = i_1^*, \ldots, I_k^* = i_k^*\right) = \prod_{j=1}^{k} \frac{\exp s_{i_j^*}}{\sum_{\ell \in N_j^*} \exp s_\ell}$$

*where $N_j^* = N \backslash \{i_1^*, \ldots, i_{j-1}^*\}$ is the domain (without replacement) for the $j$-th sampled element.*

Now, it is easy to prove the proposition:

**Proposition 3.1.** *For scores $s_i$, $i \in \{1, \ldots, n\}$, and $k \leq n$ and $\beta > 0$, if we draw $\epsilon_i \sim \text{Gumbel}(0; \beta^{-1})$ independently, then $\arg \operatorname{top}_k \{s_i + \epsilon_i\}_i$ is an (ordered) sample without replacement from the categorical distribution $\text{Categorical}(\exp(\beta s_i)/\sum_j \exp(\beta s_j), i \in \{1, \ldots, n\})$.*

*Proof.* As $\epsilon_i \sim \text{Gumbel}(0; \beta^{-1})$, define $\epsilon_i' \coloneqq \beta \epsilon_i \sim \text{Gumbel}(0; 1)$. Further, let $s_i' \coloneqq \beta s_i$. Applying Lemma B.2 on $s_i'$ and $\epsilon_i'$, $\arg \operatorname{top}_k \{s_i' + \epsilon_i'\}_i$ yields (ordered) samples without replacement from the categorical distribution $\text{Categorical}(\frac{\exp(\beta s_i)}{\sum_j \exp(\beta s_j)}, i \in \{1, \ldots, n\})$. However, multiplication by $\beta$ does not change the resulting indices of $\arg \operatorname{top}_k$:

$$\arg \operatorname{top}_k \{s_i' + \epsilon_i'\}_i = \arg \operatorname{top}_k \{s_i + \epsilon_i\}_i, \tag{26}$$

concluding the proof. □

## C  Experiments

### C.1  Experimental setup & compute

**Frameworks.** We use PyTorch. Repeated-MNIST and EMNIST experiments use PyTorch Ignite. Synbols and MIO-TCD experiments use the BaaL library: https://github.com/baal-org/baal (Atighehchian et al., 2020). Predictive parity is calculated using FairLearn (Bird et al., 2020). The CausalBALD experiments use https://github.com/anndvision/causal-bald (Jesson et al., 2021). The experiments comparing to ACS-FW (Pinsler et al., 2019) use the original authors' implementation with added support for stochastic batch acquisitions: https://github.com/BlackHC/active-bayesian-coresets/releases/tag/stoch_batch_acq_paper. The Repeated-MNIST experiments were run using https://github.com/BlackHC/active_learning_redux/releases/tag/stoch_batch_acq, and the results are also available on WandB (https://wandb.ai/oatml-andreas-kirsch/oatml-snow-stoch-acq).

**Compute.** Results shown in Table 1 were run inside Docker containers with 8 CPUs (2.2Ghz) and 32 Gb of RAM. Other experiments were run on similar machines with Titan RTX GPUs. The Repeated-MNIST and EMNIST experiments take about 5000 GPU hours. The MIO, Synbols and CLINC-150 experiments take about 19000 GPU hours. The CausalBALD experiments take about 1000 GPU hours.

**Dataset Licenses.** Repeated-MNIST is based on MNIST which is made available under the terms of the Creative Commons Attribution-Share Alike 3.0 license. The EMNIST dataset is made available as CC0 1.0 Universal Public Domain Dedication. Synbols is a dataset generator. MIO-TCD is made available under the terms of the Creative Commons Attribution-NonCommercial-ShareAlike 4.0 International License. CLINC-150 is made available under the terms of Creative Commons Attribution 3.0 Unported License.

#### C.1.1  Runtime measurements

The synthetic dataset used for benchmarking has 4,096 features, 10 classes, and 10,000 pool points. VGG-16 models (Simonyan & Zisserman, 2015) were used to sample predictions and latent embeddings.

#### C.1.2  Repeated-MNIST

The Repeated-MNIST dataset is also constructed following Kirsch et al. (2019) with duplicated examples from MNIST with isotropic Gaussian noise added to the input images (standard deviation 0.1).

We use the same setup as Kirsch et al. (2019): a LeNet-5-like architecture with ReLU activations instead of tanh and added dropout. The model obtains 99% test accuracy when trained on the full MNIST dataset. Specifically, the model is made up of two blocks of a convolution, dropout, max-pooling, ReLU with 32 and 64 channels and 5x5 kernel size, respectively. As classifier head, a two-layer MLP with 128 hidden units (and 10 output units) is used that includes dropout between the layers. We use a dropout probability of 0.5 everywhere. The model is trained with early stopping using the Adam optimiser and a learning rate of 0.001. We sample predictions using 100 MC-Dropout samples for BALD. Weights are reinitialized after each acquisition step.

#### C.1.3  EMNIST

We follow the setup from (Kirsch et al., 2019) with 20 MC dropout samples. We use a similar model as for Repeated-MNIST but with three blocks instead of two. Specifically, we use 32, 64, and 128 channels and 3x3 kernel size. This is followed by a 2x2 max pooling layer before the classifier head. The classifier head is a two-layer MLP but with 512 hidden units instead of 128. Again, we use dropout probability 0.5 everywhere.

#### C.1.4  Synbols & MIO-TCD

The full list of hyperparameters for the Synbols and MIO-TCD experiments is presented in Table 3. Our experiments are built using the BaaL library (Atighehchian et al., 2020). We compute the predictive parity using FairLearn (Bird et al., 2020). We use VGG-16 model (Simonyan & Zisserman, 2015) trained for 10 epochs using Monte Carlo dropout for acquisition (Gal et al., 2017) with 20 dropout samples.

Table 3: Hyper-parameters used in Section 5 and C.5

| Hyperparameter | Value |
|---|---|
| Learning rate | 0.001 |
| Optimiser | SGD |
| Weight decay | 0 |
| Momentum | 0.9 |
| Loss function | Crossentropy |
| Training duration | 10 |
| Batch size | 32 |
| Dropout $p$ | 0.5 |
| MC iterations | 20 |
| Query size | 100 |
| Initial set | 500 |

In Figure 9, we show a set of images with common problems that can be found in MIO-TCD.

### C.1.5 CLINC-150

We fine-tune a pretrained DistilBERT model from HuggingFace (Wolf et al., 2020) on CLINC-150 for 5 epochs with Adam as optimiser. Estimating epistemic uncertainty in transformer models is an open research question, and hence, we do not report results using BALD and focus on entropy instead.

### C.1.6 CausalBALD

Using the Neyman-Rubin framework (Neyman, 1923; Rubin, 1974; Sekhon, 2008), the CATE is formulated in terms of the potential outcomes, $Y_t$, of treatment levels $t \in \{0, 1\}$. Given observable covariates, $\mathbf{X}$, the CATE is defined as the expected difference between the potential outcomes at the measured value $\mathbf{X} = \mathbf{x}$: $\tau(\mathbf{x}) = \mathbb{E}[Y_1 - Y_0 \mid \mathbf{X} = \mathbf{x}]$. This causal quantity is fundamentally unidentifiable from observational data without further assumptions because it is not possible to observe both $Y_1$ and $Y_0$ for a given unit. However, under the assumptions of consistency, non-interference, ignoreability, and positivity, the CATE is identifiable as the statistical quantity $\widetilde{\tau}(\mathbf{x}) = \mathbb{E}[Y \mid T = 1, \mathbf{X} = \mathbf{x}] - \mathbb{E}[Y \mid T = 0, \mathbf{X} = \mathbf{x}]$ (Rubin, 1980).

Jesson et al. (2021) define BALD acquisition functions for active learning CATE functions from observational data when the cost of acquiring an outcome, y, for a given covariate and treatment pair, $(\mathbf{x}, t)$, is high. Because we do not have labels for $Y_1$ and $Y_0$ for each $(\mathbf{x}, t)$ pair in the dataset, their acquisition function focusses on acquiring data points $(\mathbf{x}, t)$ for which it is likely that a matched pair $(\mathbf{x}, 1 - t)$ exists in the pool data or has already been acquired at a previous step. We follow their experiments on their synthetic dataset with limited positivity and the semi-synthetic IHDP dataset (Hill, 2011). Details of the experimental setup are given in (Jesson et al., 2021), we use their provided code, and implement the power acquisition function.

The settings for causal inference experiments are identical to those used in Jesson et al. (2021), using the IHDP dataset (Hill, 2011). Like them, we use a Deterministic Uncertainty Estimation Model (van Amersfoort et al., 2021), which is initialised with 100 data points and acquire 10 data points per acquisition batch for 38 steps. The dataset has 471 pool points and a 201 point validation set.

### C.1.7 ACS-FW

We compare to the experiments in the paper 'Bayesian Batch Active Learning as Sparse Subset Approximation' (Pinsler et al., 2019) which introduces the ACS-FW algorithm for batch acquisition. For its deep learning experiments, the paper uses a ResNet feature extractor, followed by a Bayesian multi-class classification model on the final layer. Since exact inference is intractable, variational inference is used with a factorized Gaussian posterior approximation. As prior a zero-mean Gaussian is used, and Monte-Carlo samples are drawn to approximate the predictive distribution.

## C.2 Repeated-MNIST

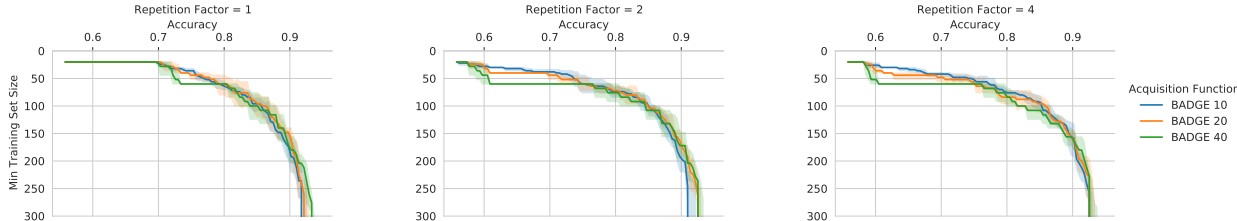

Figure 10: *Repeated-MNIST x4 (5 trials): acquisition size ablation for BADGE.* Acquisition size 40 performs best out of {10, 20, 40} for repetition factor 1, acquisition 20 performs best for repetition factor 2, and all three perform similarly for repetition factor 4.

**BADGE Ablation.** In Figure 10, we see that BADGE performs best with acquisition size 20 on Repeated-MNISTx4 overall. BADGE 40 and BADGE 20 have the highest final accuracy, cf. BADGE 10 while BADGE 20 performs better than BADGE 40 for small training set sizes.

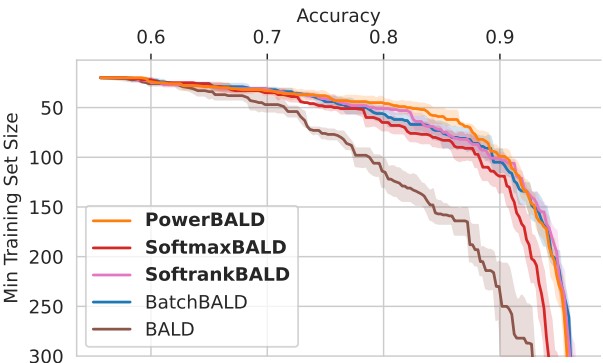

Figure 11: *Repeated-MNIST x4 (5 trials): acquisition size 5.* To compare BatchBALD, BALD and the stochastic acquisition strategies more fairly, we also show the performance of acquisition size 5. This is in line with Figure 2.

**Acquisition Size 5.** In Figure 11, we see that the stochastic acquisition strategies perform better than BALD and as well as BatchBALD for acquisition size 5 as well. Thus, we see no qualitative difference to the results in Figure 2.

### C.2.1  Other scoring functions

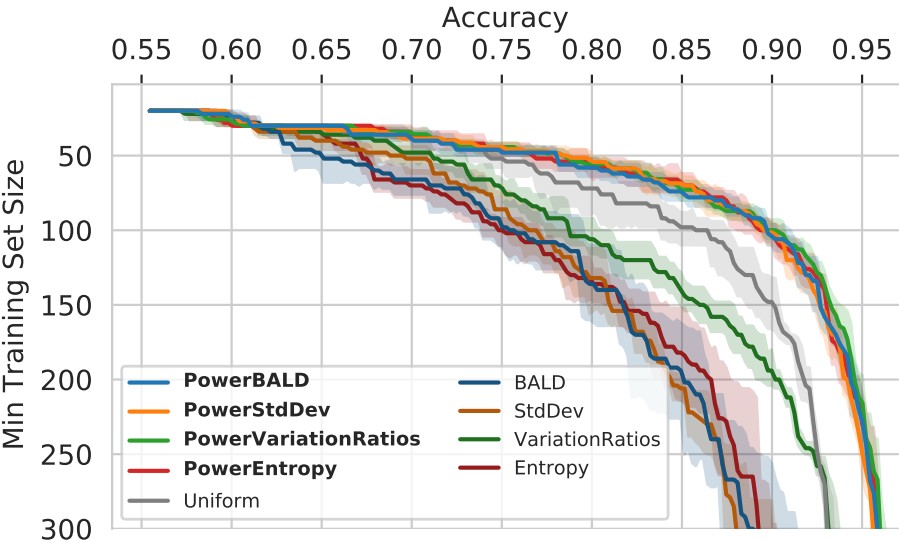

Figure 12: *Repeated-MNIST x4 (5 trials): Performance for other scoring functions.* Entropy, std dev, variation ratios behave like BALD when applying a stochastic sampling scheme (Power from §3).

In Figure 12 shows the performance of other scoring functions than BALD on RepeatedMNIST x4.

### C.2.2  Redundancy ablation

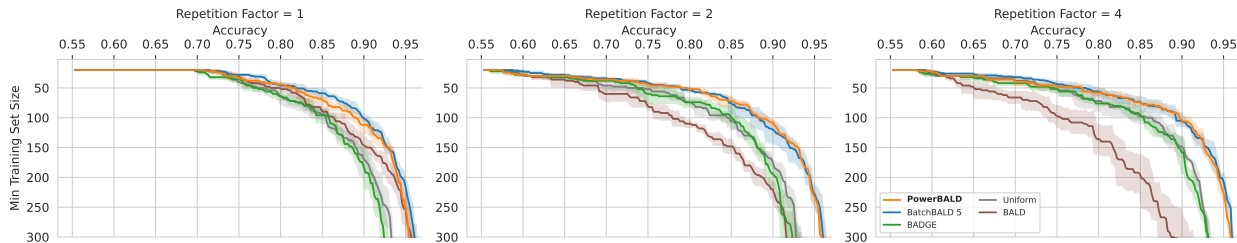

Figure 13: *Repeated-MNIST (5 trials): Performance ablation for different repetition counts.*

In Figure 13, we see the same behaviour in an ablation for different repetition sizes of Repeated-MNIST.

## C.3 MIO-TCD

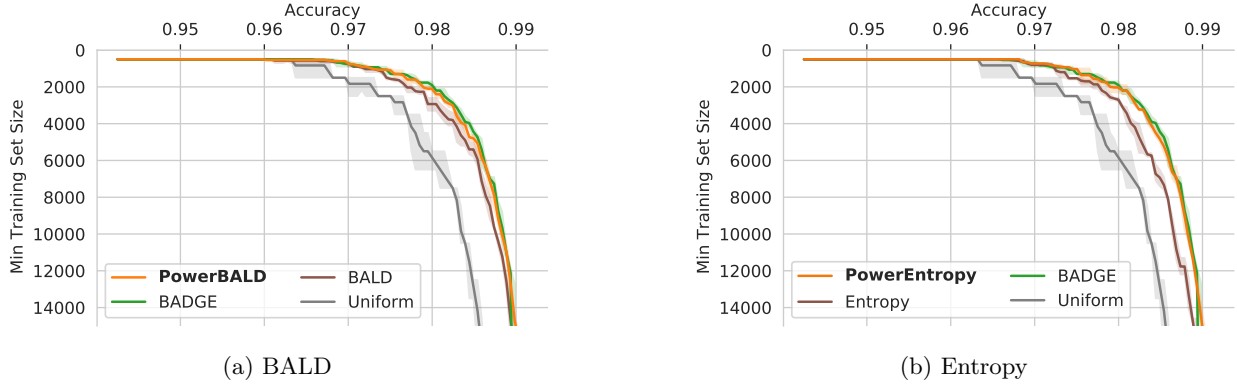

(a) BALD

(b) Entropy

Figure 14: *MIO-TCD (5 trials).*

In Figure 14, we see that power acquisition performs on par with BADGE with both BALD and entropy as underlying score functions.

## C.4 EMNIST

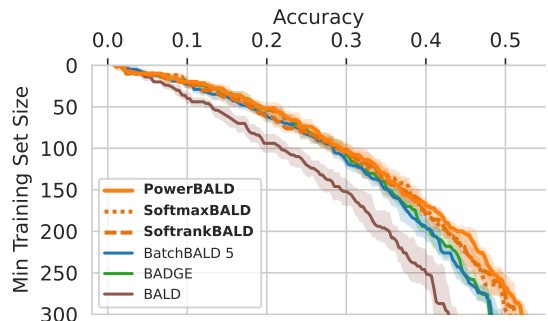

Figure 15: *EMNIST (Balanced) (5 trials): Performance with BALD.*

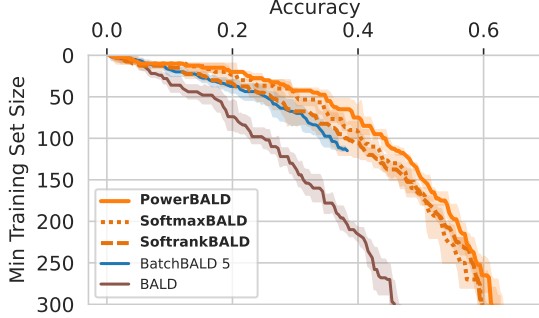

Figure 16: *EMNIST (ByMerge) (5 trials): Performance with BALD.*

In Figure 15 and 16, we see that PowerBALD outperforms BALD, BatchBALD, and BADGE.

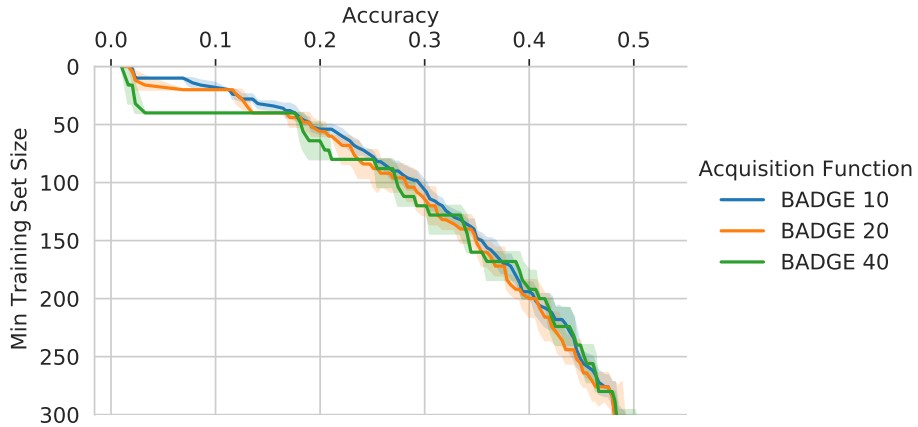

Figure 17: *EMNIST (Balanced) (5 trials): acquisition size ablation for BADGE.*

**BADGE Ablation.** In Figure 17, we see that BADGE performs similarly with all three acquisition sizes. Acquisition size 10 is the smoothest.

### C.5 Edge cases in Synbols

We use Synbols (Lacoste et al., 2020) to demonstrate the behaviour of batch active learning in artificially constructed edge cases. Synbols is a character dataset generator for classification where a user can specify the type and proportion of bias and insert artefacts, backgrounds, masking shapes, and so on. We selected three datasets with strong biases supplied by Lacoste et al. (2020); Branchaud-Charron et al. (2021) to evaluate stochastic batch acquisition methods. We use an acquisition size of 100. The experimental settings are described in Appendix C.1.

For these tasks, performance evaluation includes 'predictive parity', also known as 'accuracy difference', which is the maximum difference in accuracy between subgroups—which are, in this case, different coloured characters. This measure is used most widely in domain adaptation and ethics (Verma & Rubin, 2018). We want to maximise the accuracy while minimising the predictive parity.

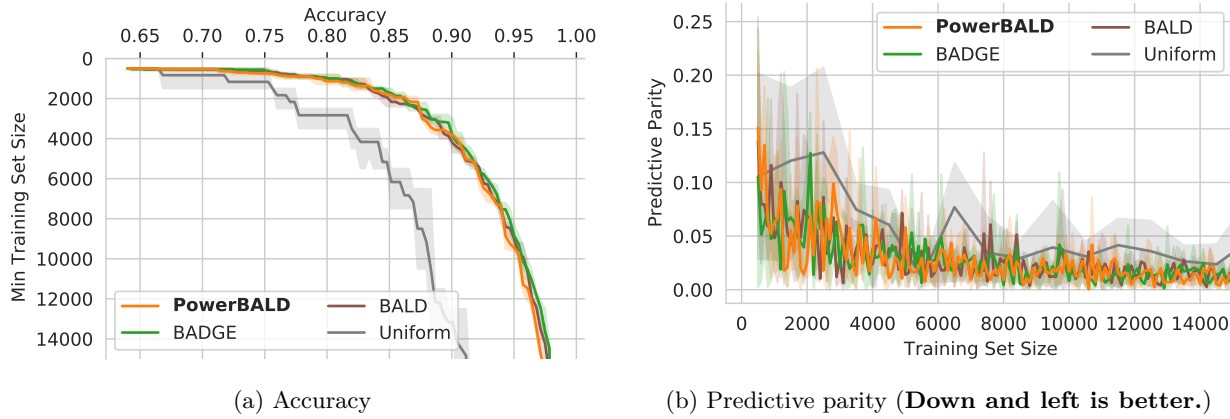

(a) Accuracy

(b) Predictive parity (**Down and left is better.**)

Figure 18: *Performance on Synbols Spurious Correlations (3 trials) with BALD.* Stochastic acquisition matches BADGE and BALD's predictive parity and performance, which is reassuring as stochastic acquisition functions might be affected by spurious correlations.

**Spurious Correlations.** This dataset includes spurious correlations between character colour and class. As shown in Branchaud-Charron et al. (2021), active learning is especially strong here as characters that do not follow the correlation will be informative and thus selected.

We compare the predictive parity between methods in Fig. 18b. We do not see any significant difference between the stochastic batch acquisition method and BADGE or BALD. This is encouraging, as stochastic approaches might select more examples following the spurious correlation and thus have higher predictive parity, but this is not the case.

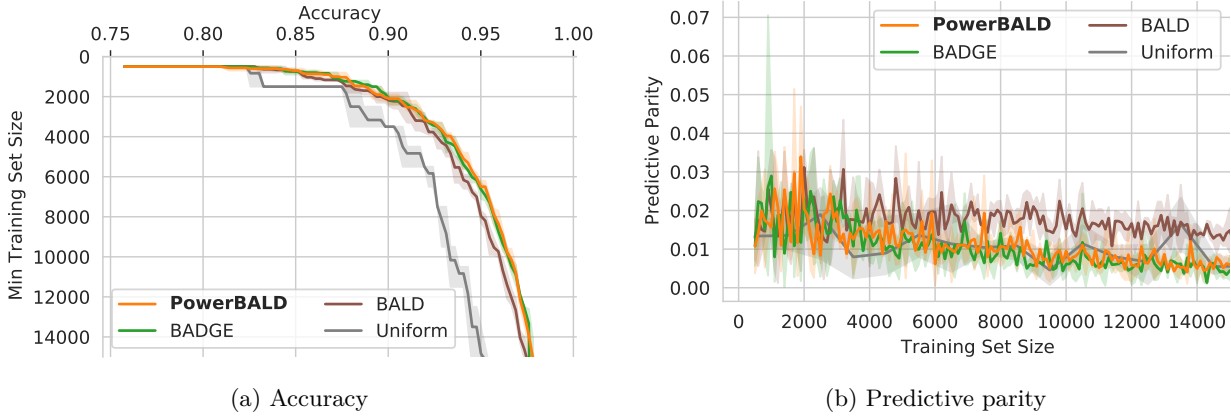

(a) Accuracy
(b) Predictive parity

Figure 19: *Synbols Minority Groups (3 trials): Performance on BALD.* PowerBALD outperforms BALD and matches BADGE for both accuracy and predictive parity.

**Minority Groups.** This dataset includes a subgroup of the data that is under-represented; specifically, most characters are red while few are blue. As Branchaud-Charron et al. (2021) shows, active learning can improve the accuracy for these groups.

Our stochastic approach lets batch acquisition better capture under-represented subgroups. In Figure 19a, PowerBALD has an accuracy almost identical to that of BADGE, despite being much cheaper, and outperforms BALD. At the same time, we see in Figure 19b that PowerBALD has a lower predictive parity than BALD, demonstrating a fairer predictive distribution given the unbalanced dataset.

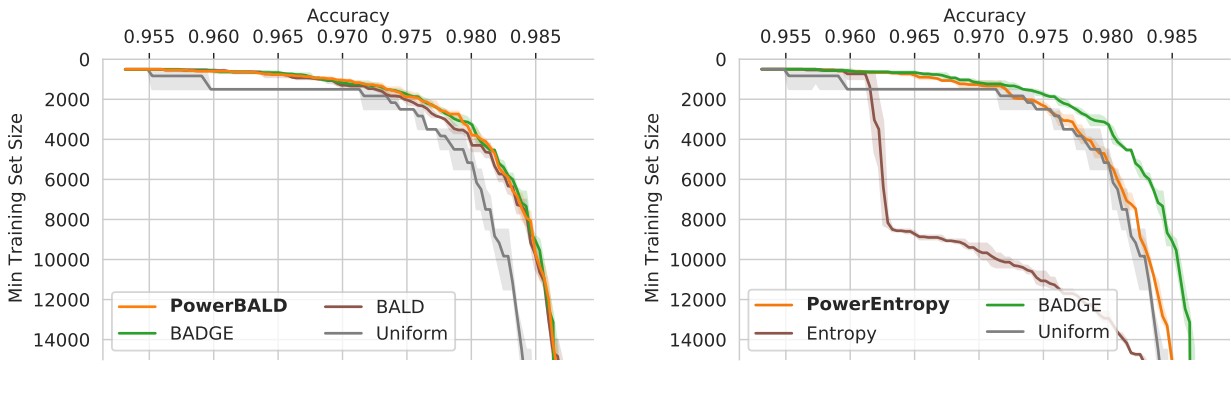

Figure 20: BALD
Figure 21: Entropy

Figure 22: *Performance on Synbols Missing Characters (3 trials).* In this dataset with high aleatoric uncertainty, PowerBALD matches BADGE and BALD performance. PowerEntropy significantly outperforms Entropy which confounds aleatoric and epistemic uncertainty.

**Missing Synbols.** This dataset has high aleatoric uncertainty (input noise). Some images are missing information required to make high-probability predictions—these images have shapes randomly occluding the character—so even a perfect model would remain uncertain. Lacoste et al. (2020) demonstrated that entropy is ineffective on this data as it cannot distinguish between aleatoric and epistemic uncertainty (input noise and model uncertainty), while BALD can do so. As a consequence, entropy will unfortunately prefer samples with occluded characters, resulting in degraded active learning performance. For predictive entropy, stochastic acquisition largely corrects the failure of entropy acquisition to account for missing data (Figure 22) although PowerEntropy still underperforms BADGE here. For BALD, we show in Figure 20 in the appendix that, as before, the stochastic batch acquisition method performs on par with BADGE and marginally better than BALD.

## C.6 CLINC-150

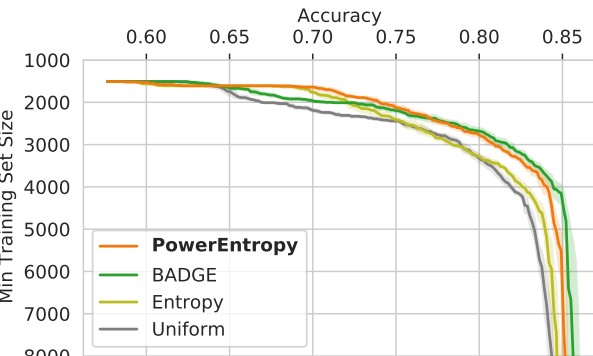

Figure 23: *Performance on CLINC-150 (10 trials).* PowerEntropy performs much better than entropy, which only performs marginally better than uniform, and almost on par with BADGE.

In Figure 23, we see that PowerEntropy performs much better than entropy which only performs marginally better than the uniform baseline. PowerEntropy also performs better than BADGE at low training set sizes, but BADGE performs better in the second half. Between $\approx 2300$ and 4000 samples, BADGE and PowerEntropy perform the same. We use an acquisition size of 100, starting from an initial training set of 1510 points (10 points per intent class).

## C.7 ACS-FW

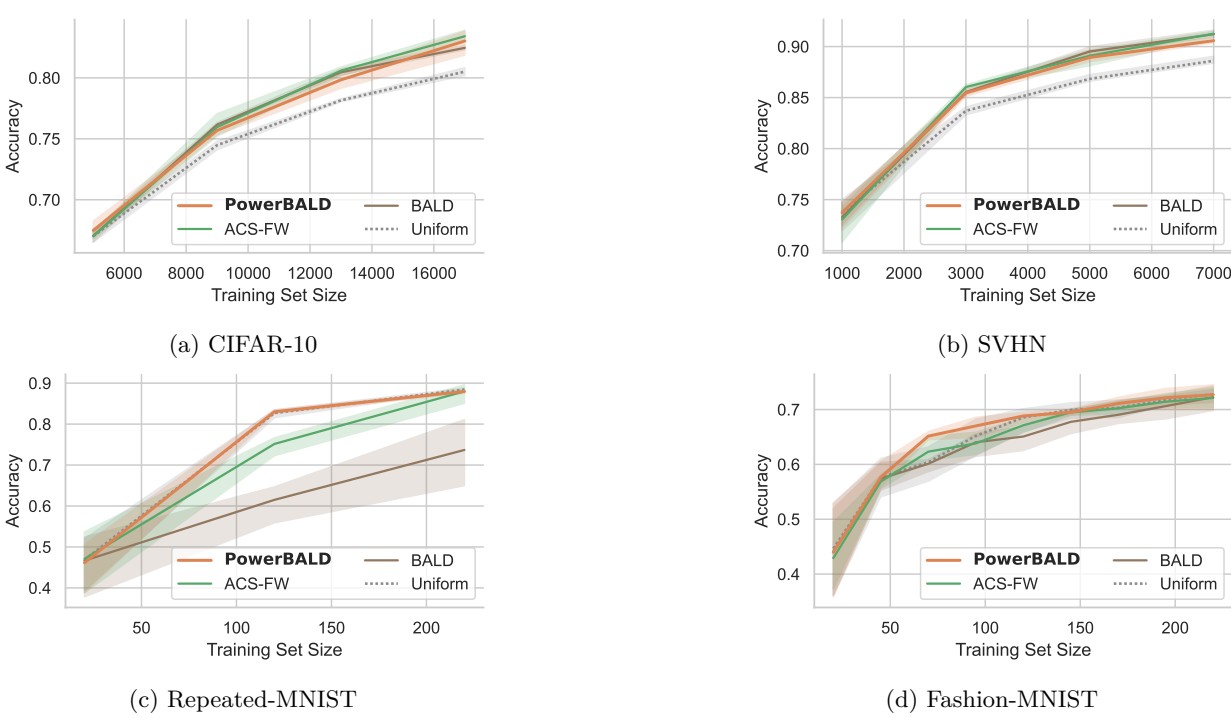

(a) CIFAR-10

(b) SVHN

(c) Repeated-MNIST

(d) Fashion-MNIST

Figure 24: *MFVI Last-Layer Performance (3 trials).*

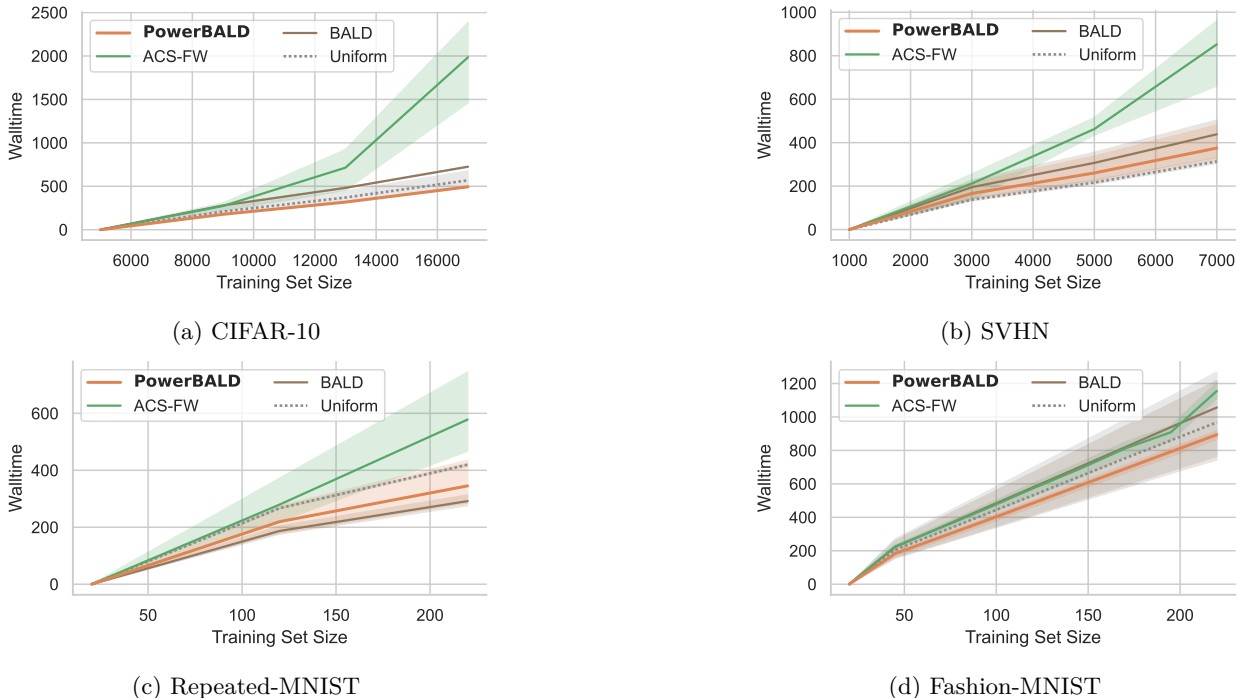

(a) CIFAR-10

(b) SVHN

(c) Repeated-MNIST

(d) Fashion-MNIST

Figure 25: *MFVI Last-Layer Walltime (3 trials).*

We run comparisons for several datasets using last-layer mean-field variational inference: CIFAR-10 (Krizhevsky et al., 2009), SVHN (Netzer et al., 2011), Repeated-MNIST (Kirsch et al., 2019), and Fashion-MNIST (Xiao et al., 2017). We use 5000 initial training samples with an acquisition size of 4000 for CIFAR-10, 1000 initial training samples with an acquisition size of 2000 for SVHN, 20 initial training samples with an acquisition size of 100 for Repeated-MNIST, and 20 initial training samples with an acquisition size of 25 for Fashion-MNIST. The same hyperparameters as in Pinsler et al. (2019) are used.

In Figure 24, we see that PowerBALD performs best on Fashion-MNIST and Repeated-MNIST with smaller acquisition sizes, but ACS-FW and BALD perform similarly on CIFAR-10 and SVHN at larger acquisition sizes. This is in line with Section 6, where we have seen in Figures 5 and 7 that the top 1% of the samples are more sensitive than the top 10%: larger acquisition sizes can be less sensitive to the issues of top-$K$. The performance of ACS-FW contrasts to the results in Pinsler et al. (2019) where ACS-FW performs better than BALD on CIFAR-10 and SVHN. In Appendix E.3, we provide ablations with different acquisition sizes and initial training set sizes for these four datasets. Overall, beyond what we have noted above, these results are not conclusive, however, and the performance curves are very close to each other. On the other hand, the wall time curves in Figure 25 show that ACS-FW is significantly slower than PowerBALD. This is congruent with the motivation that stochastic sampling methods are fast yet competitive and don't underperform compared to top-$K$ acquisition, thus being a good substitute for top-$K$ acquisition.

# D Comparing Power, Softmax and Soft-Rank

## D.1 Empirical Evidence

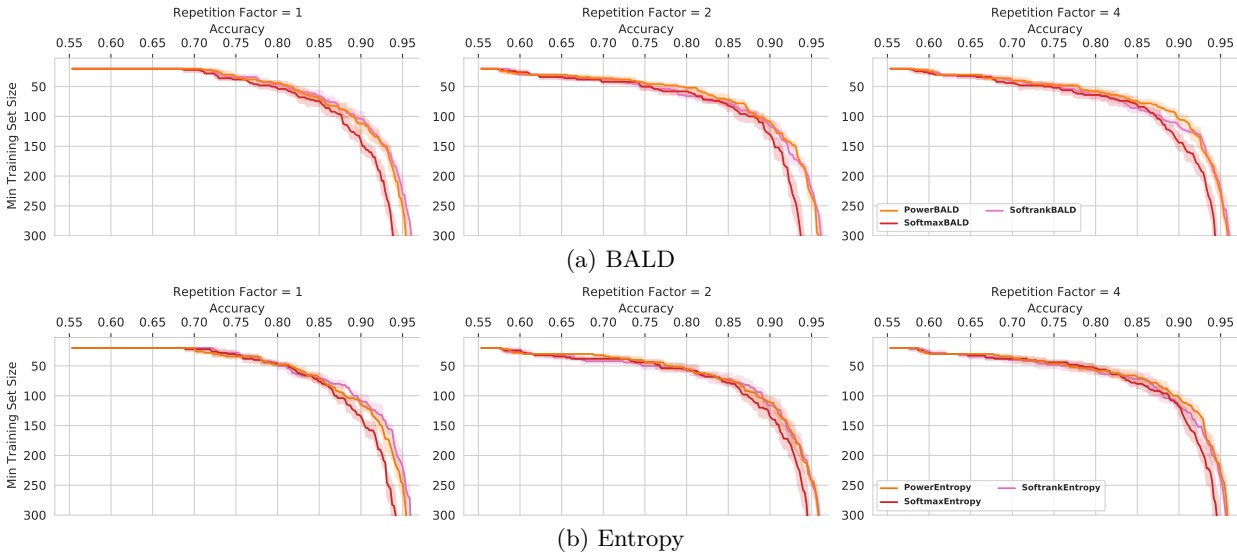

Figure 26: *Repeated-MNIST (5 trials): Performance with all three stochastic strategies.*

**Repeated-MNIST.** In Figure 26, power acquisition performs best overall, followed by soft-rank and then softmax.

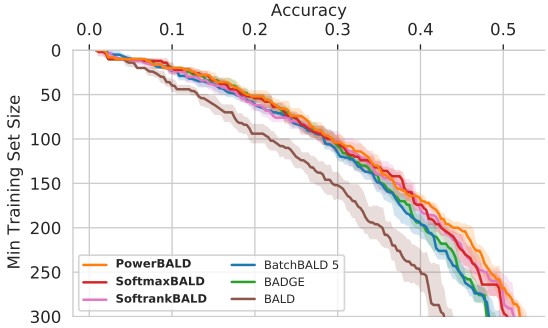

Figure 27: *EMNIST (Balanced) (5 trials): Performance with all three stochastic strategies with BALD.* PowerBALD performs best.

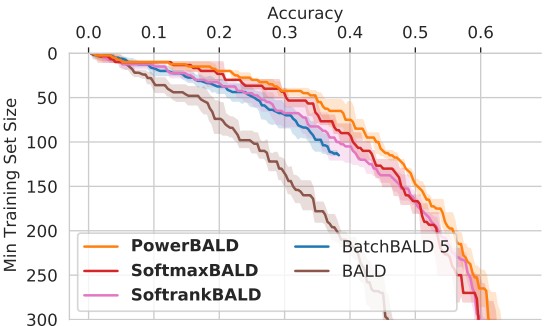

Figure 28: *EMNIST (ByMerge) (5 trials): Performance with all three stochastic strategies with BALD.* PowerBALD performs best.

**EMNIST.** In Figure 27 and 28, we see that PowerBALD performs best, but Softmax- and SoftrankBALD also outperform other methods. BADGE did not run on EMNIST (ByMerge) due to out-of-memory issues and BatchBALD took very long as EMNIST (ByMerge) has more than 800,000 samples.

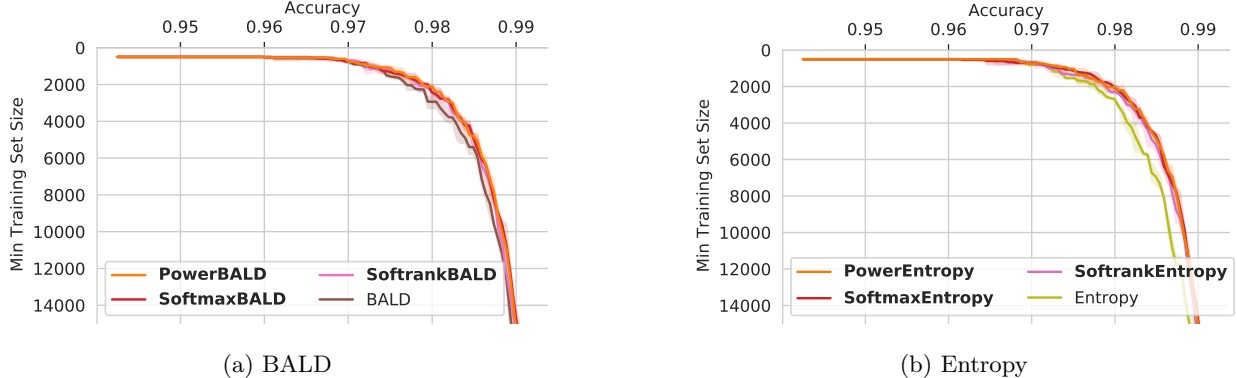

(a) BALD

(b) Entropy

Figure 29: *MIO-TCD (3 trials): Performance with all three stochastic strategies.*

**MIO-TCD.** In Figure 29, we see that all three stochastic acquisition methods perform about equally well.

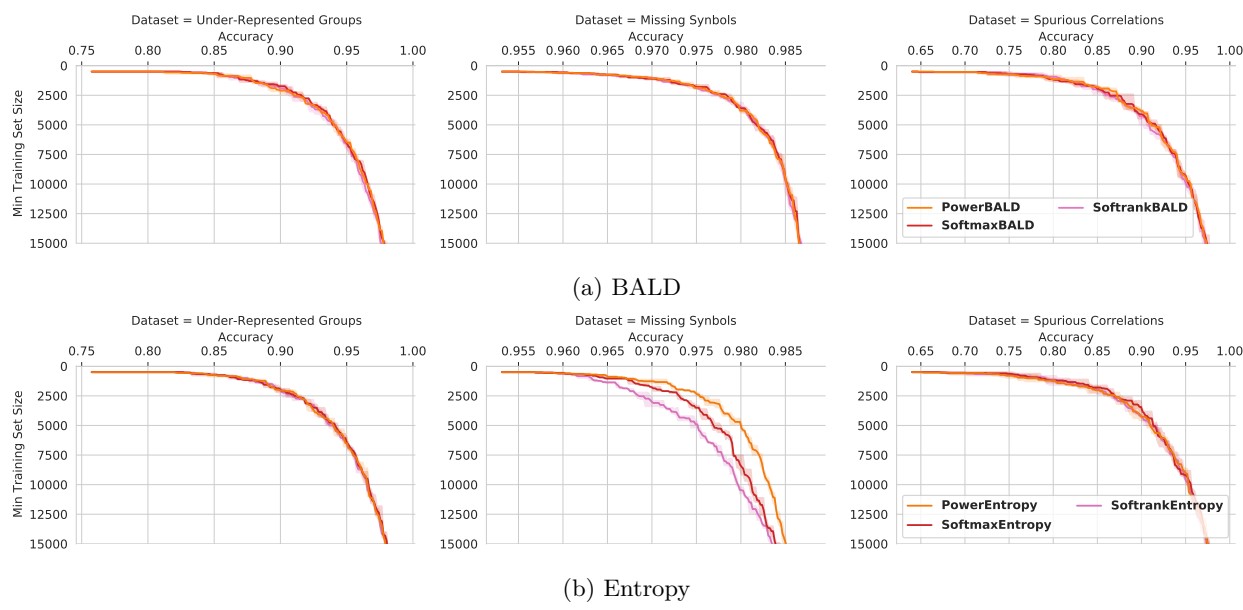

(a) BALD

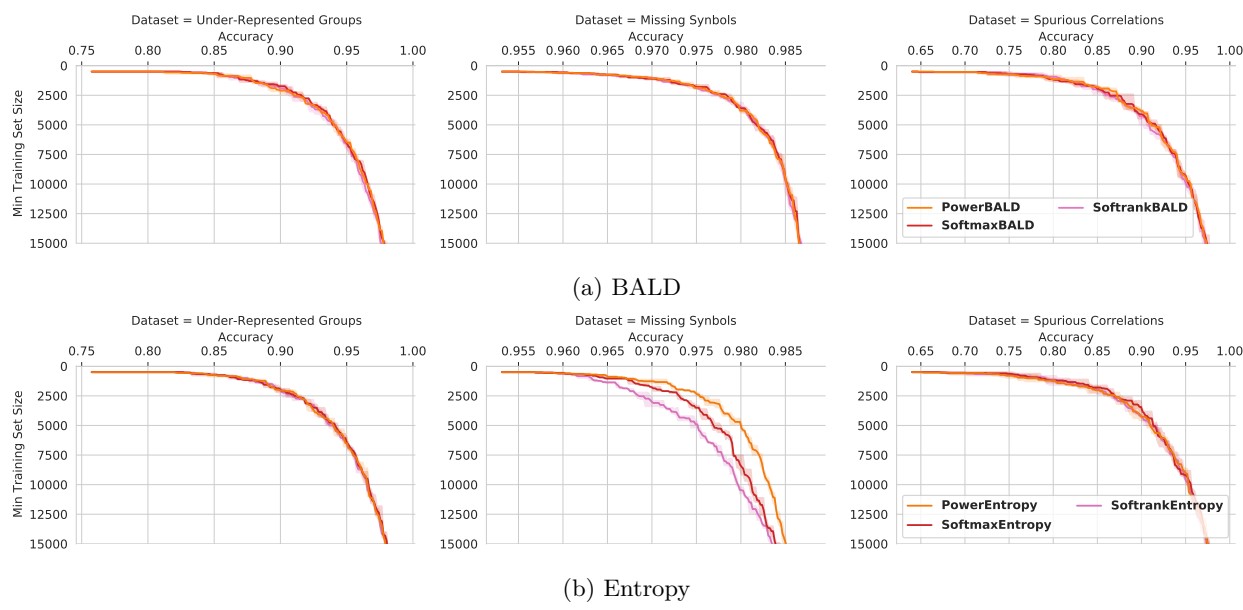

(b) Entropy

Figure 30: *Synbols edge cases (3 trials): Performance with all three stochastic strategies.*

**Synbols.** In Figure 30, power acquisition seems to perform better overall—mainly due to the performance in Synbols Missing Characters.

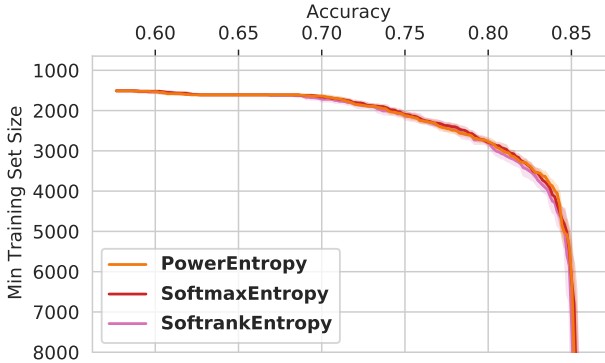

Figure 31: *CLINC-150 (10 trials): Performance with all three stochastic strategies.*

**CLINC-150.** In Figure 31, all three stochastic methods perform similarly.

### D.2 Investigation

To further examine the three stochastic acquisition variants, we plot their score distributions, extracted from the same MNIST toy example, in Figure 32. Power and softmax acquisition distributions are similar for $\beta = 8$ (power, softmax) and $\beta = 4$ (softmax). This might explain why active learning with these $\beta$ shows similar accuracy trajectories.

We find that power and softmax acquisition are quite insensitive to $\beta$ and thus selecting $\beta = 1$ might generally work quite well.

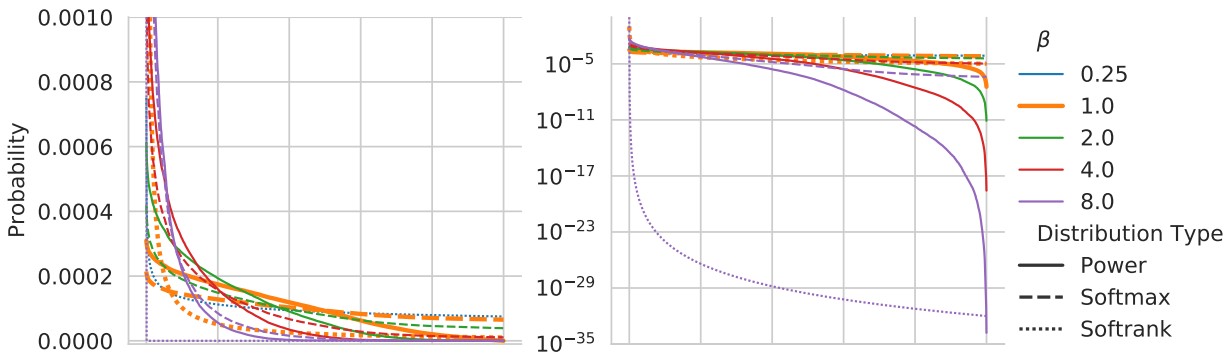

Figure 32: *Score distribution for power and softmax acquisition of BALD scores on MNIST for varying Coldness $\beta$ at $t = 0$.* Linear and log plot over samples sorted by their BALD score. At $\beta = 8$ both softmax and power acquisition have essentially the same distribution for high scoring points (closely followed by the power distribution for $\beta = 4$). This might explain why the coldness ablation shows that these $\beta$ to have very similar AL trajectories on MNIST. Yet, while softmax and power acquisition seem transfer to RMNIST, this is not the case for softrank which is much more sensitive to $\beta$. At the same time, power acquisition avoids low-scoring points more than softmax acquisition.

# E   Effect of changing the aquisition sizes

## E.1   Repeated-MNIST

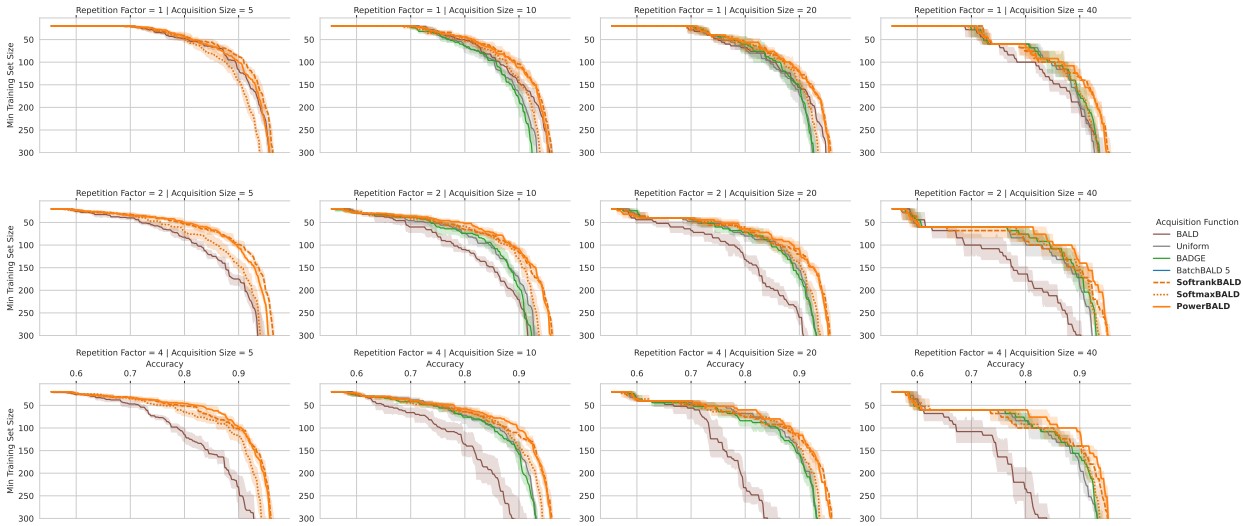

Figure 33: *Repeated-MNIST: Acquisition size ablation grouped by acquisition size.* Stochastic acquisition strategies (except for Softrank at acquisition size 5) always perform better than top-$K$ BALD and also better than BADGE. Softrank is the most sensitive to acquisition size changes because it is independent of the scores.

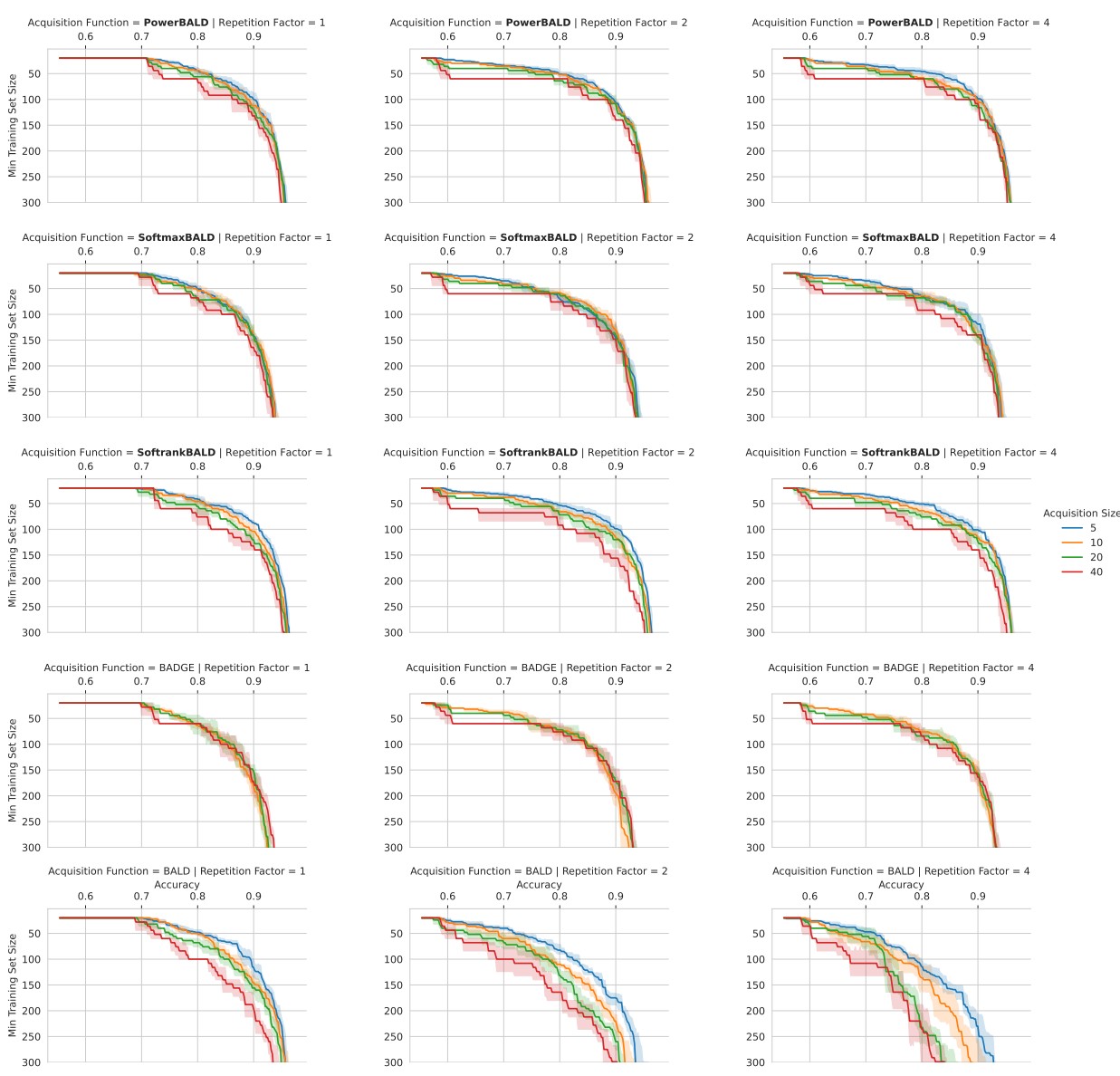

Figure 34: *Repeated-MNIST: Acquisition size ablation grouped by acquisition function.* PowerBALD and SoftmaxBALD are less sensitive to acquisition size changes while SoftrankBALD is more sensitive to acquisition size changes. Overall, all three are much less sensitive to acquisition size changes than top-$K$ BALD.

## E.2 EMNIST

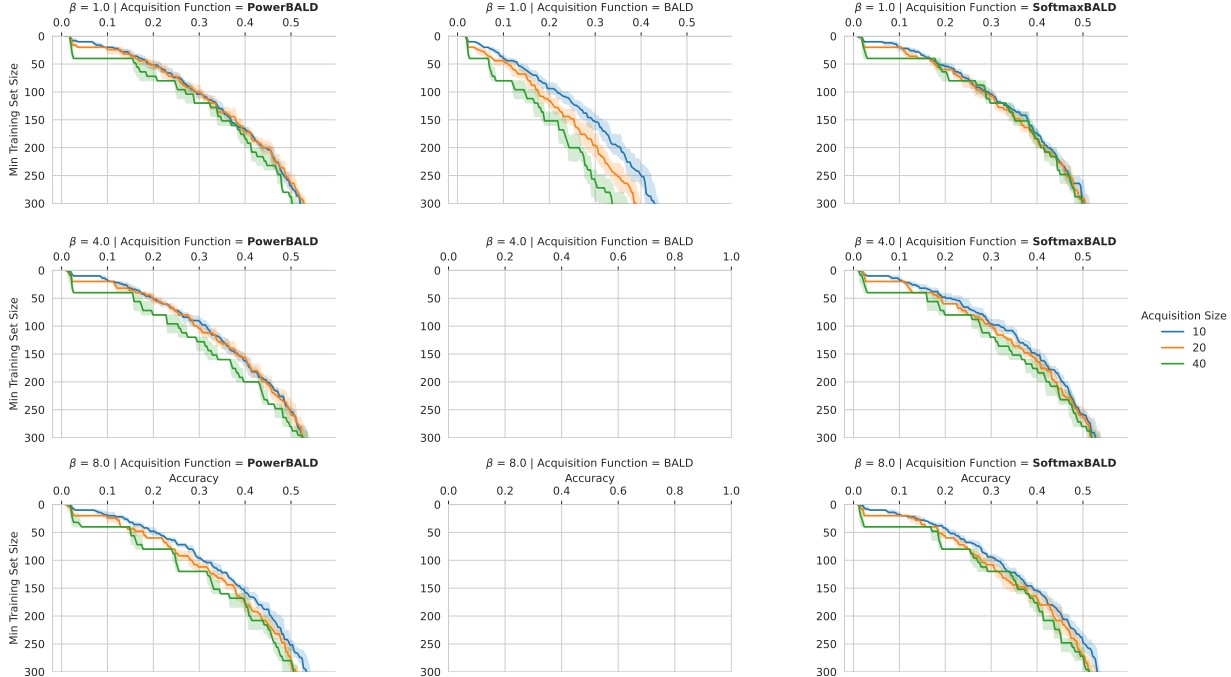

Figure 35: *EMNIST 'Balanced': Acquisition size ablation.* For (top-$K$) BALD, $\beta$ makes no difference, so $\beta = 1$ is used as placeholder. The other two subplots are intentionally left out as they are very similar to this one.

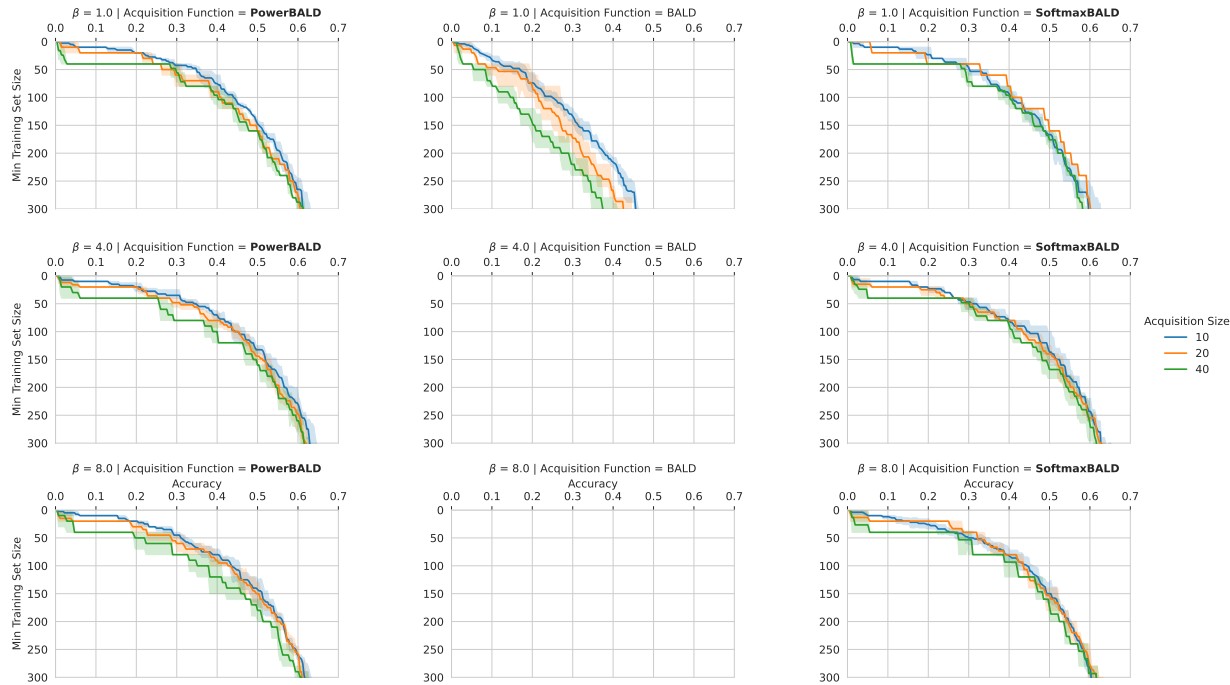

Figure 36: *EMNIST 'ByMerge': Acquisition size ablation.* For (top-$K$) BALD, $\beta$ makes no difference, so $\beta = 1$ is used as placeholder. The other two subplots are intentionally left out as they are very similar to this one.

## E.3  ACS-FW

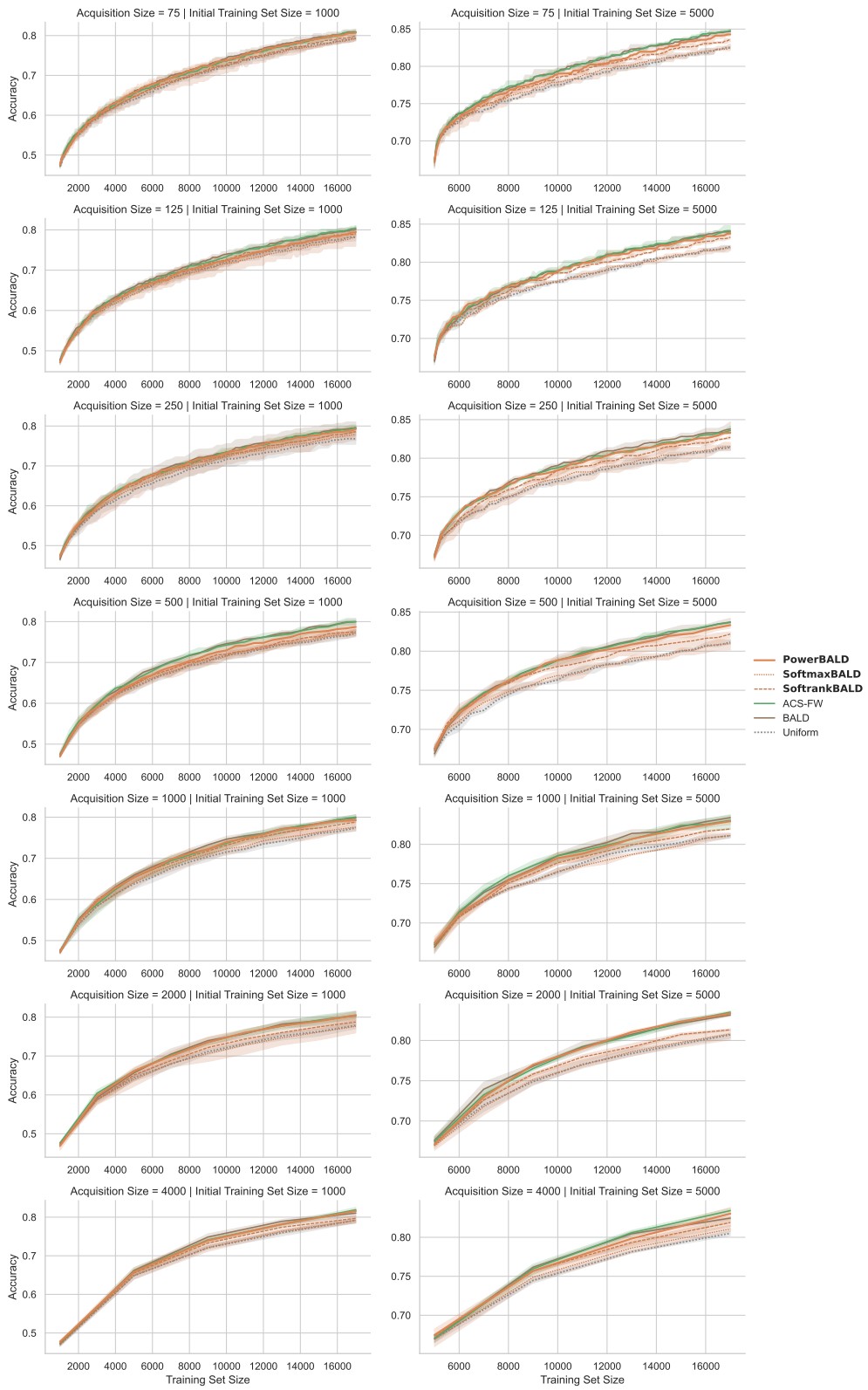

Figure 37: *MFVI Last-Layer Classification on CIFAR-10: Acquisition size ablation.*

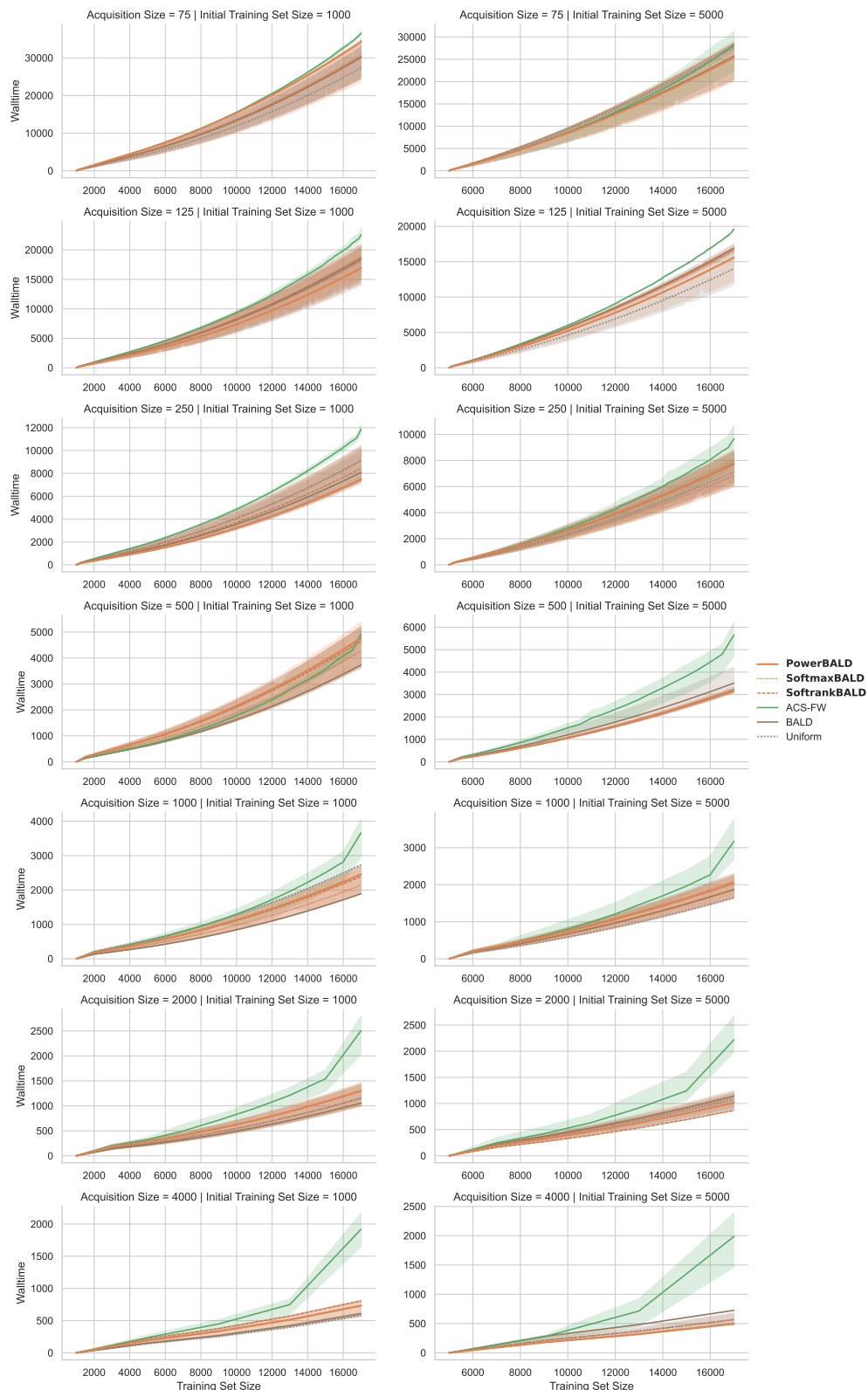

Figure 38: *MFVI Last-Layer Classification on CIFAR-10: Acquisition size ablation (Walltime).*

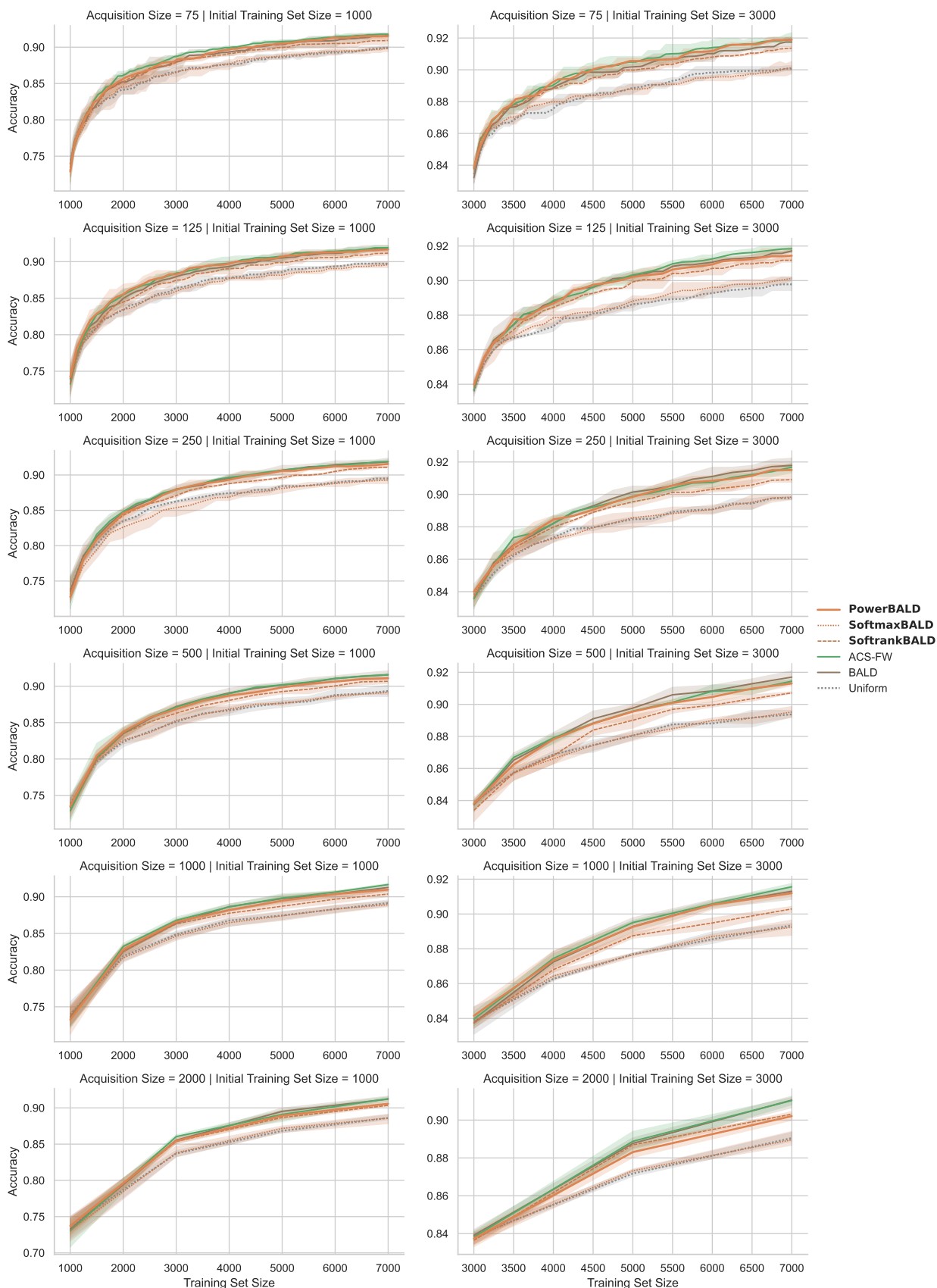

Figure 39: *MFVI Last-Layer Classification on SVHN: Acquisition size ablation.*

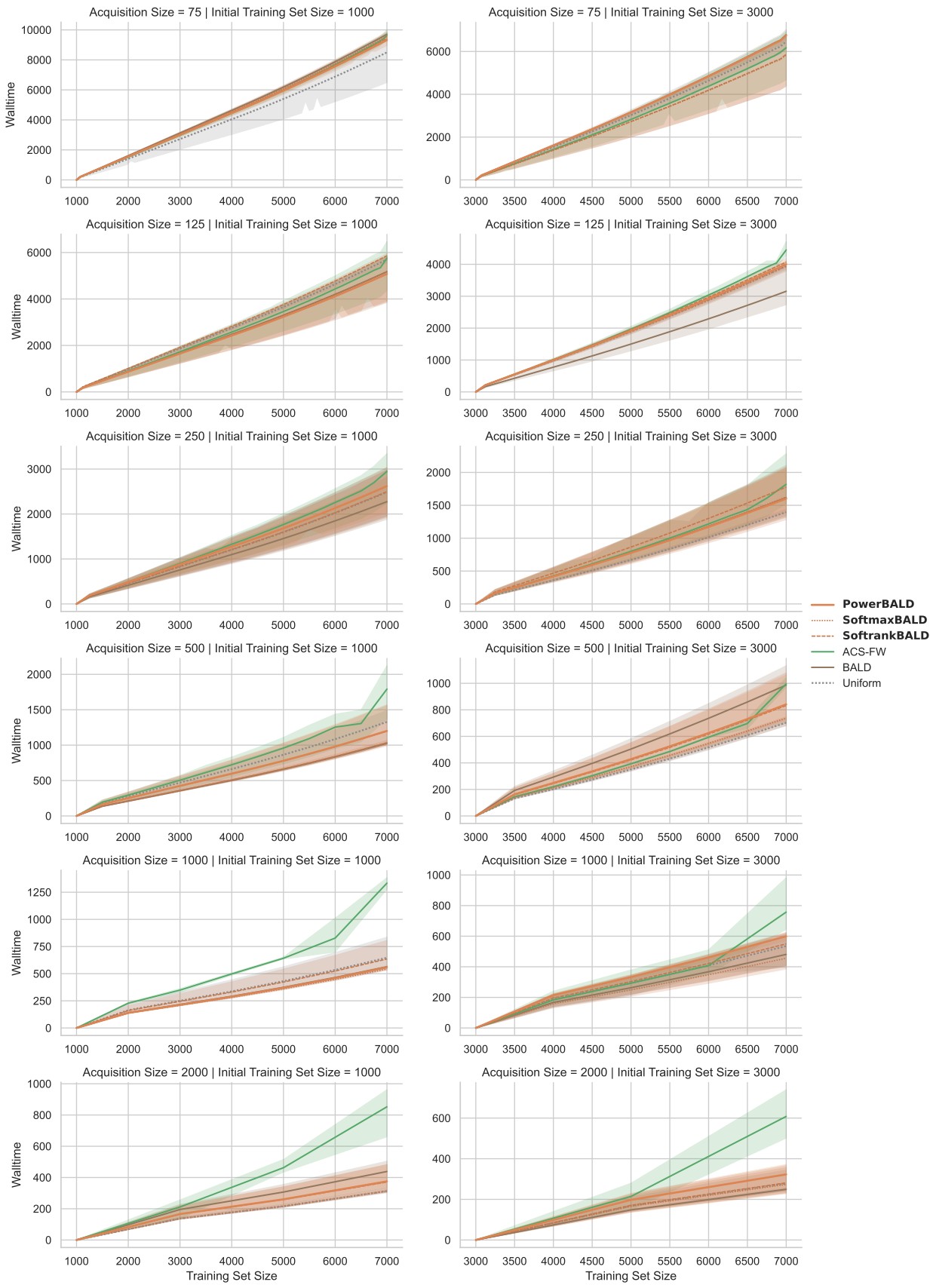

Figure 40: *MFVI Last-Layer Classification on SVHN: Acquisition size ablation (Walltime).*

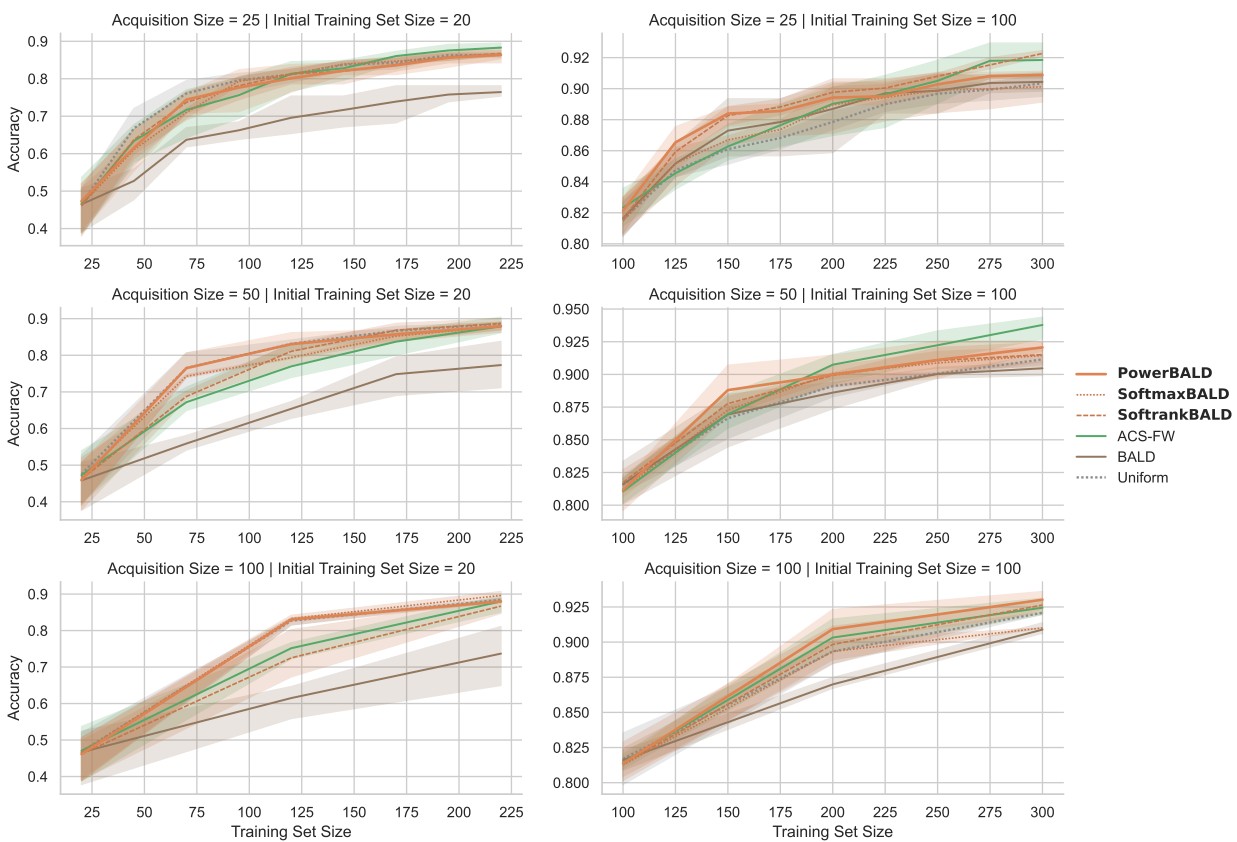

Figure 41: *MFVI Last-Layer Classification on Repeated-MNIST: Acquisition size ablation.*

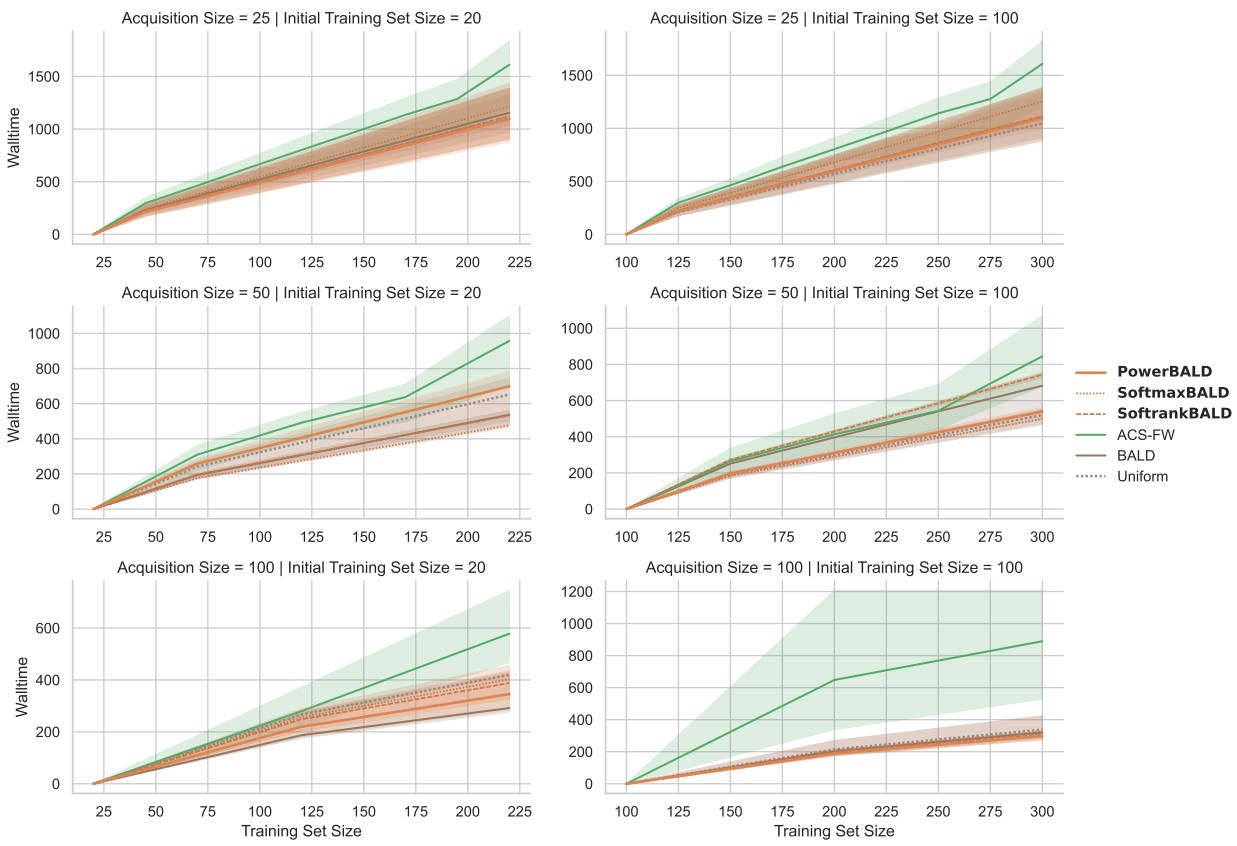

Figure 42: *MFVI Last-Layer Classification on Repeated-MNIST: Acquisition size ablation (Walltime).*

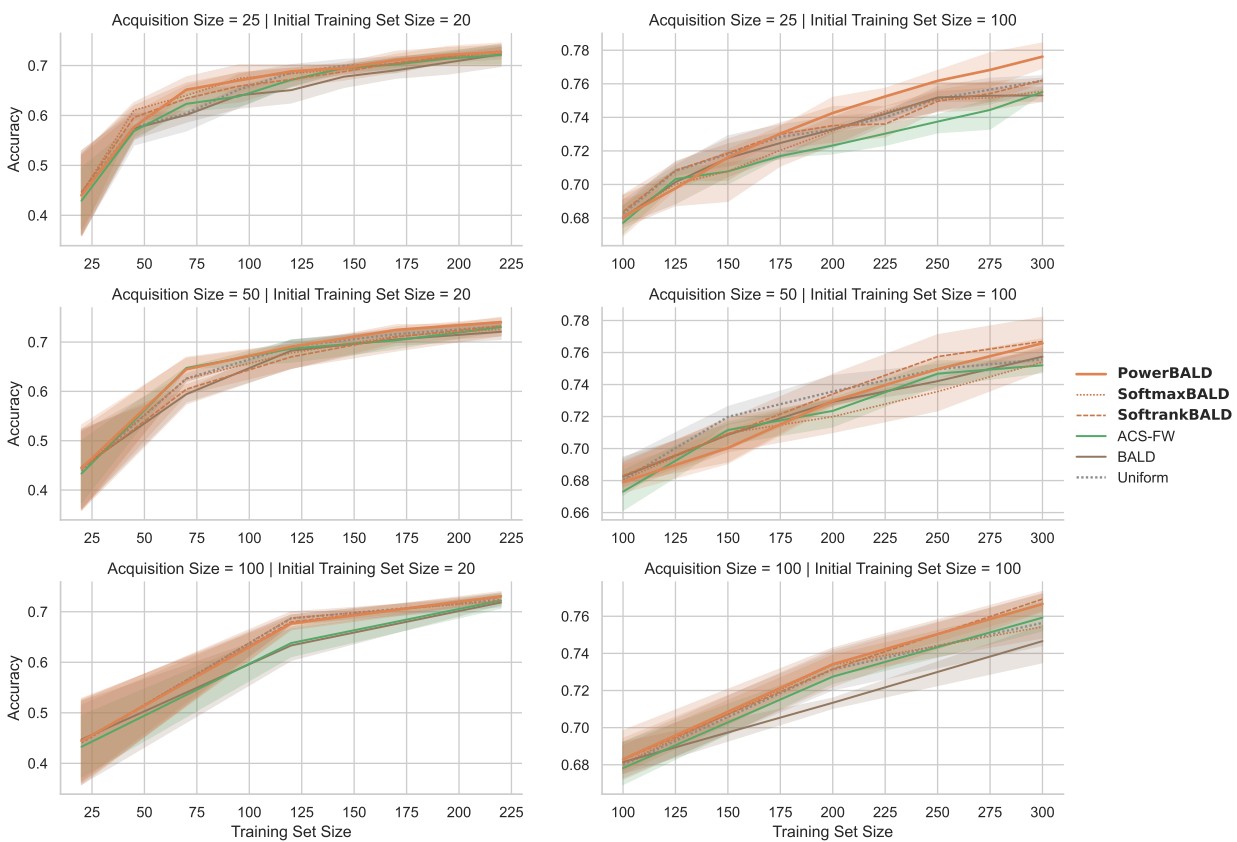

Figure 43: *MFVI Last-Layer Classification on Fashion-MNIST: Acquisition size ablation.*

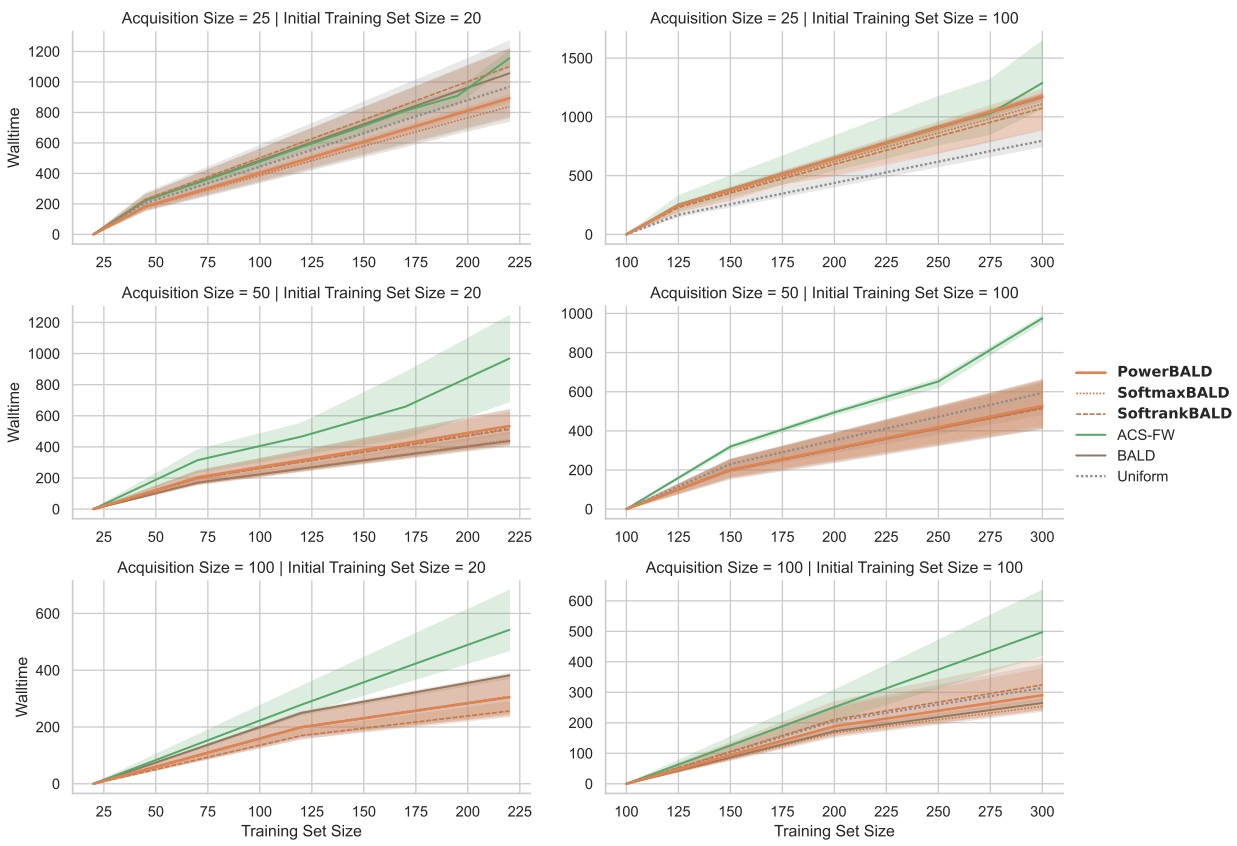

Figure 44: *MFVI Last-Layer Classification on Fashion-MNIST: Acquisition size ablation (Walltime).*

# F    Effect of changing $\beta$

## F.1    Repeated-MNIST

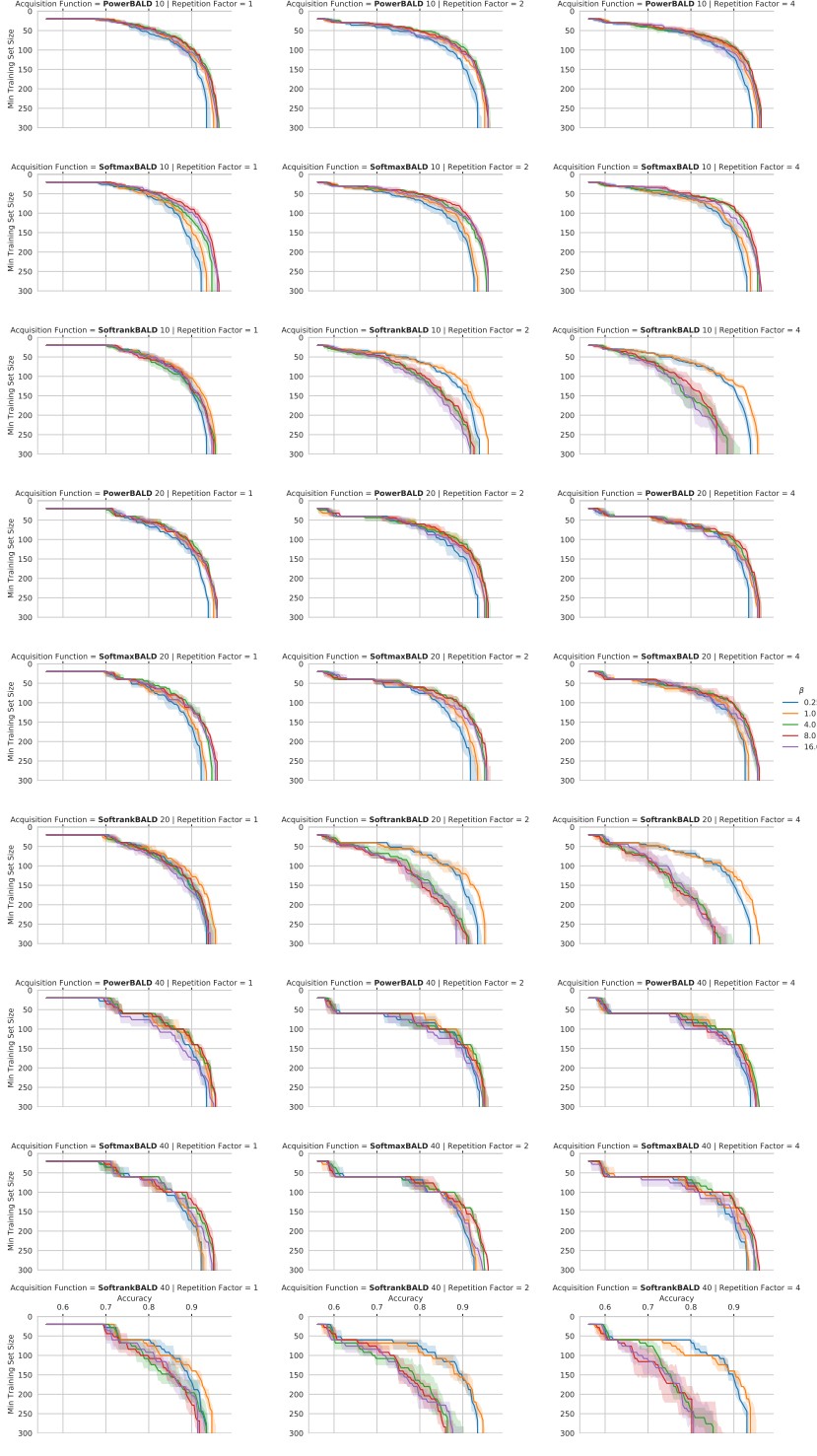

Figure 45: *Repeated-MNIST: $\beta$ ablation for *BALD.*

### F.1.1   MIO-TCD and Synbols

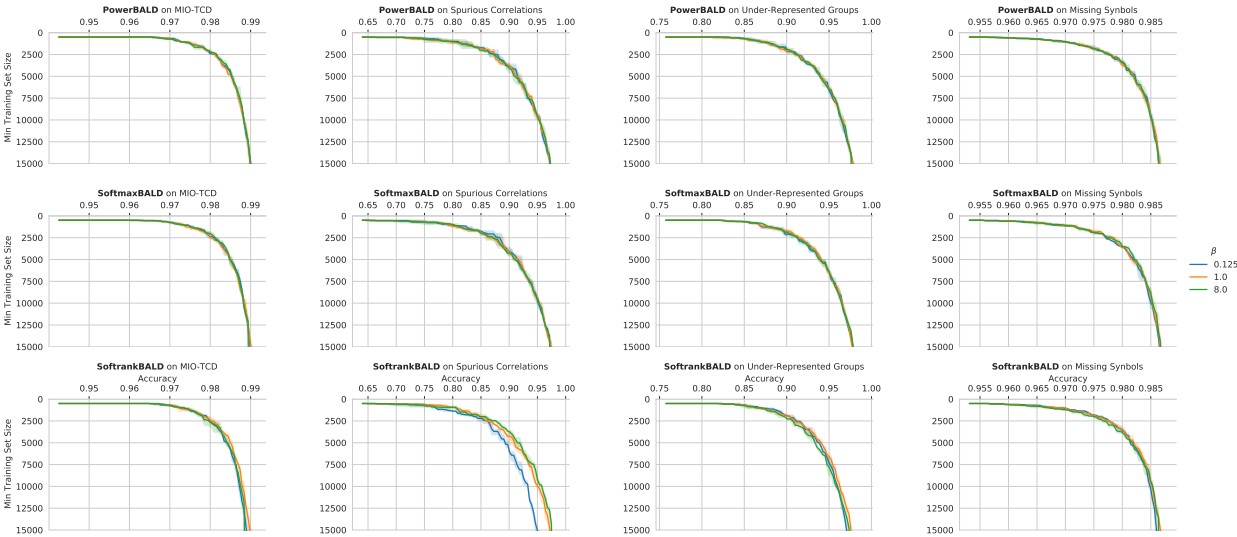

Figure 46: *MIO-TCD and Synbols: β ablation for \*BALD.*

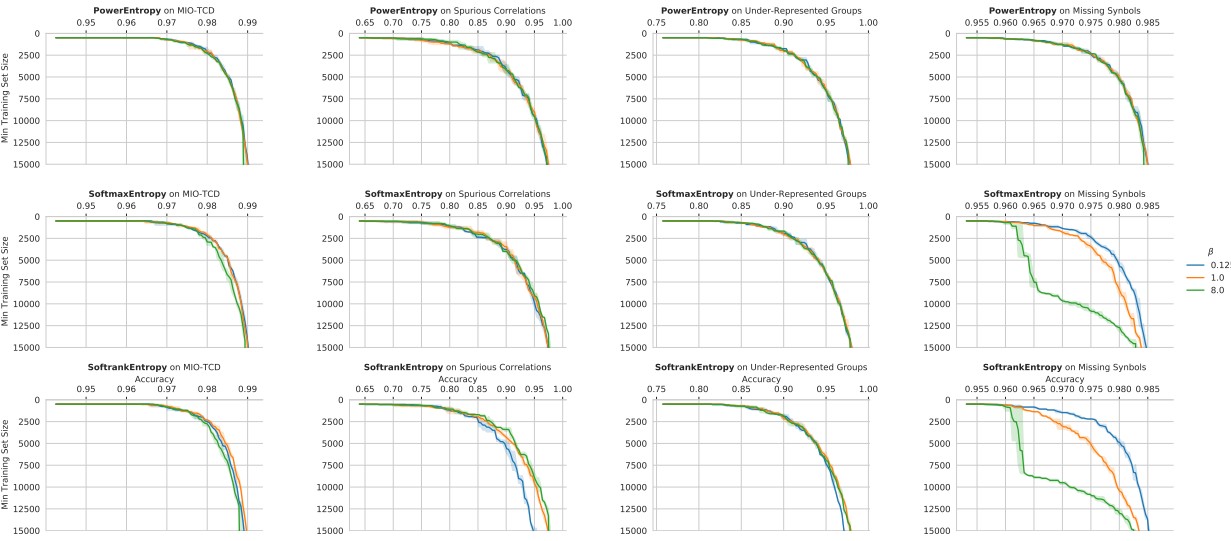

Figure 47: *MIO-TCD and Synbols: β ablation for \*Entropy.*

## F.2 EMNIST

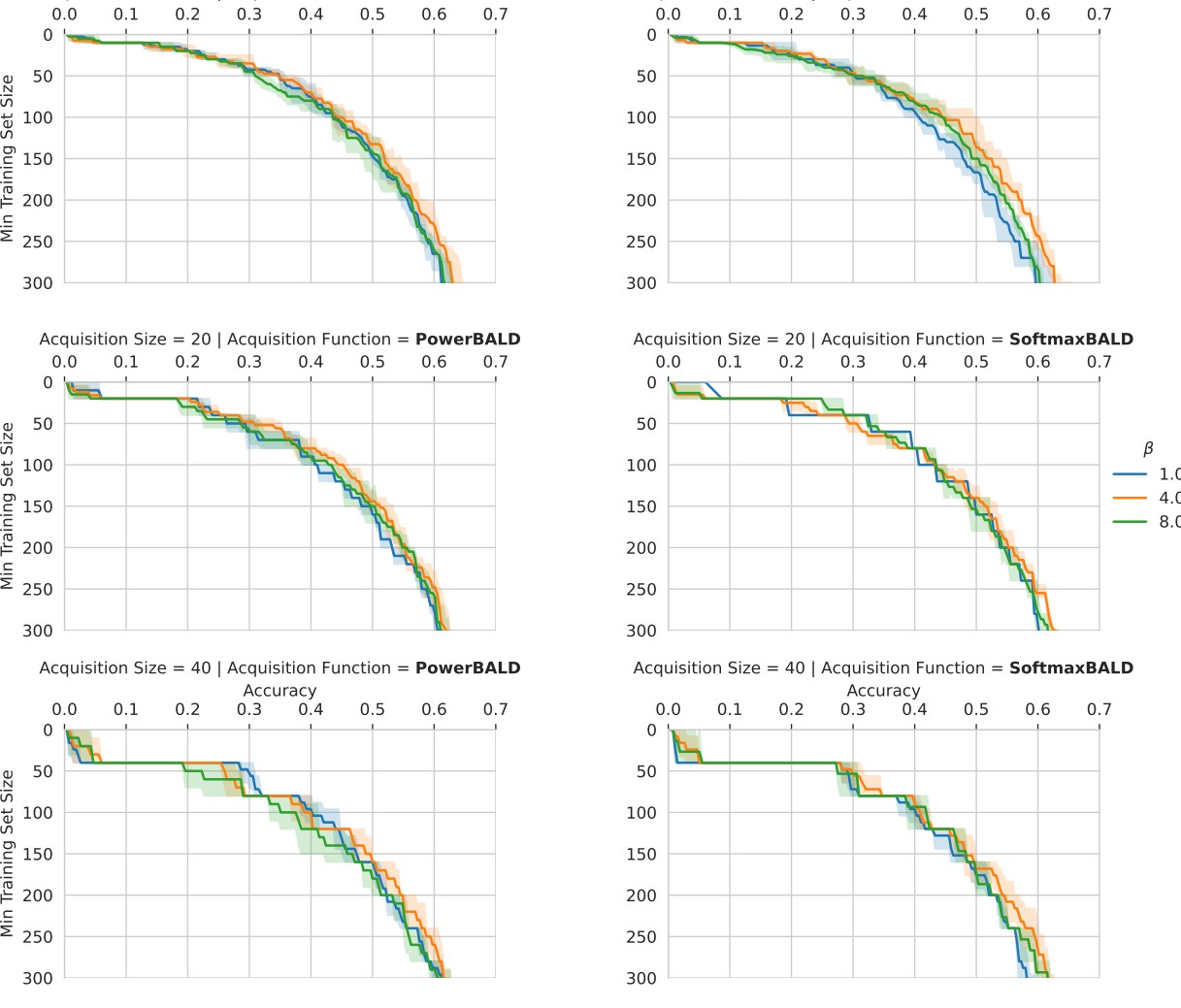

Figure 48: *EMNIST 'Balanced': β ablation for Softmax- and PowerBALD.*

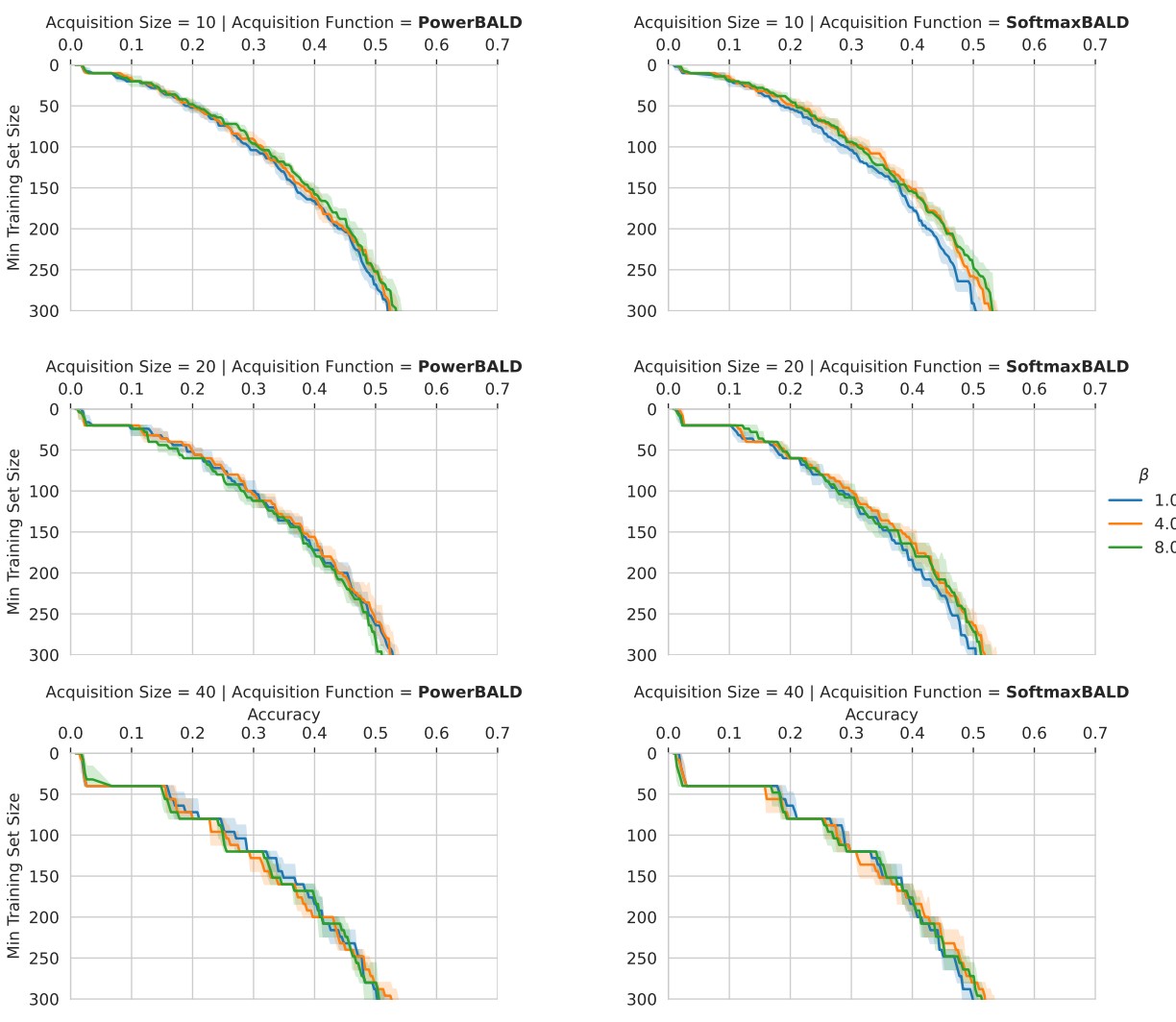

Figure 49: *EMNIST 'ByMerge': β ablation for Softmax- and PowerBALD.*

## F.3 CausalBALD: synthetic dataset

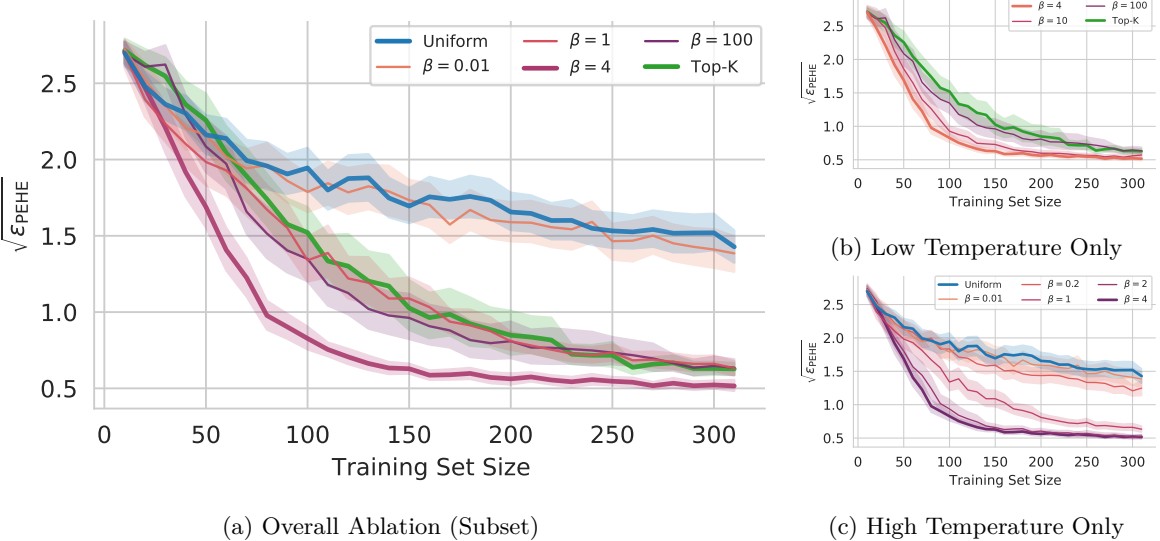

(a) Overall Ablation (Subset)

(b) Low Temperature Only

(c) High Temperature Only

Figure 50: *CausalBALD: Synthetic Dataset.* (a) At a very high temperature ($\beta = 0.1$), PowerBALD behaves very much like random acquisition, and as the temperature decreases the performance of the acquisition function improves (lower $\sqrt{\epsilon_{\text{PEHE}}}$). (b) Eventually, the performance reaches an inflection point ($\beta = 4.0$) and any further decrease in temperature results in the acquisition strategy performing more like top-$K$. We see that under the optimal temperature, power acquisition significantly outperforms both random acquisition and top-$K$ over a wide range of temperature settings.

We provide further $\beta$ ablations for CausalBALD on the entirely synthetic dataset which is used by Jesson et al. (2021). This demonstrates the ways in which $\beta$ interpolates between uniform and top-$K$ acquisition.

## F.4 CLINC-150

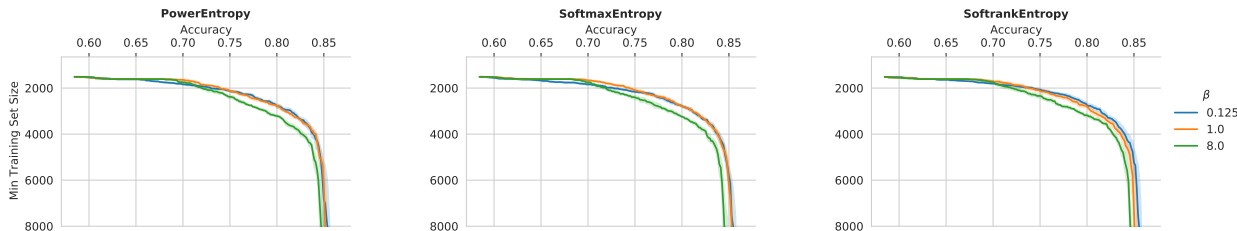

Figure 51: Performance CLINC-150: $\beta$ ablation for *Entropy.

## G    Simple Implementation of Stochastic Batch Acquisition

```python
def get_random_samples(scores_N: torch.Tensor, *, aquisition_batch_size: int):
    N = len(scores_N)
    aquisition_batch_size = min(aquisition_batch_size, N)

    indices = np.random.choice(N, size=aquisition_batch_size, replace=False)
    return indices.tolist()

def get_softmax_samples(scores_N: torch.Tensor, *, beta: float, aquisition_batch_size: int):
    # As beta -> 0, we obtain random sampling.
    if beta == 0.0:
        return get_random_samples(scores_N, aquisition_batch_size=aquisition_batch_size)

    N = len(scores_N)
    noised_scores_N = scores_N + scipy.stats.gumbel_r.rvs(loc=0, scale=1 / beta, size=N, random_state=None)

    return torch.topk(noised_scores_N, k=aquisition_batch_size)

def get_power_samples(scores_N: torch.Tensor, *, beta: float, aquisition_batch_size: int):
    return get_softmax_samples(torch.log(scores_N), beta=beta, aquisition_batch_size=aquisition_batch_size)

def get_softrank_samples(scores_N: torch.Tensor, *, beta: float, aquisition_batch_size: int):
    N = len(scores_N)

    sorted_indices_N = torch.argsort(scores_N, descending=True)
    ranks_N = torch.argsort(sorted_indices_N) + 1

    return get_power_samples(1 / ranks_N, beta=beta, aquisition_batch_size=aquisition_batch_size)
```

Listing 1: *Code for stochastic batch acquisition.* Colab here.

## H  Hypoexponential Distribution Evaluation

```python
import numpy as np
from scipy.stats import expon, gumbel_r
import matplotlib.pyplot as plt

# Set the rate parameter
# Change here to get different scales of exponentials
lam = 1/2

# Generate 10 samples from the exponential distribution
samples = expon.rvs(scale=1/lam, size=5000)

# Now sample from the samples using the previous samples as lambda of exponential distributions
new_samples = expon.rvs(scale=1/samples, size=(4000,5000))

what_are_you = new_samples.mean(axis=1)

# Fit a Gumbel distribution to the data
params = gumbel_r.fit(what_are_you)

x = np.linspace(np.min(what_are_you), np.max(what_are_you), 1000)
gumbel_cdf = gumbel_r.cdf(x, *params)

# Plot both what_are_you and the Gumbel distribution in the same plot

plt.plot(np.sort(what_are_you), np.linspace(0, 1, len(what_are_you), endpoint=False), label="Simulated Hypoexponential")
plt.plot(x, gumbel_cdf, label="Gumble Fit")
plt.legend()
plt.show()
```

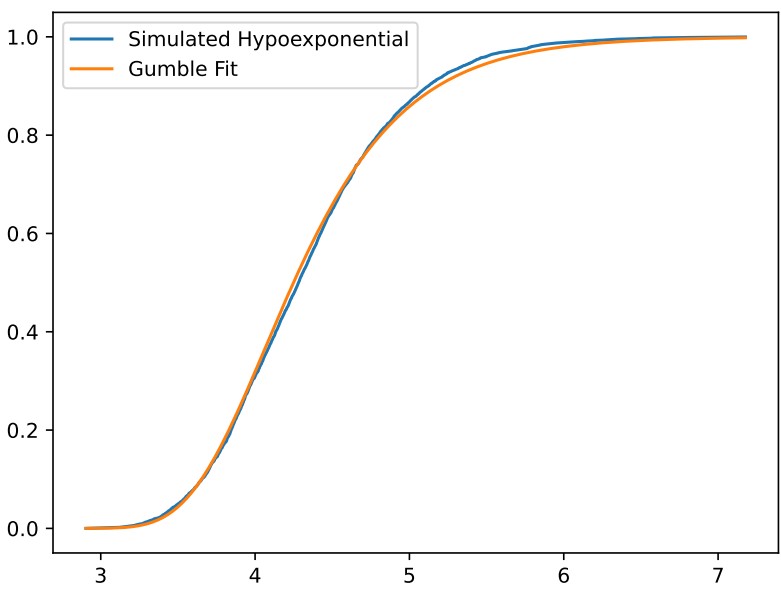

Listing 2: *Code for fitting a Gumbel distribution to the mean of samples from a hypoexponential distribution.* Colab here.

