# OpenReview forum: "Stochastic Batch Acquisition: A Simple Baseline for Deep Active Learning"
_TMLR — Accepted by TMLR_

### Review · Reviewer_NuWJ · 2023-02-09

**Summary Of Contributions:**

The authors consider the task of batch acquisition in active learning, i.e., acquiring multiple labels at a time, rather than single ones as is done in many popular approaches. Methods such as BatchBALD (Kirsch et al., 2019) or BADGE (Ash et al., 2020) are designed to model this task properly, while the simplest heuristic would be to take the top-K points at each labeling round as proposed by the acquisition function. The authors discuss and demonstrate the flaw in this naive heuristic (its assumption on the stability of the ranking between labeling rounds) and instead propose a simple sampling-based approach that performs as well as/better than methods designed for batch acquisition while remaining at the computation cost of top-K acquisition. The approach is tested extensively in a variety of experimental settings and ablations.

**Audience:**

Yes

**Claims And Evidence:**

Yes

**Requested Changes:**

See weaknesses.


**Strengths And Weaknesses:**

## Strengths
The approach provides a simple and computationally cheap solution to the task of batch acquisition. It is well-motivated and evaluated on a large array of different data sets, where the approach can provide consistent results. The paper is well written.

As the authors acknowledge in the related work, similar stochastic approaches appear repeatedly within the active learning literature often hidden in experimental sections and appendencies as approximate heuristics that seem to help without being evaluated on their own. The contribution is therefore minor in the sense of coming up with the proposed idea. But given that, it is even more valuable as being a proper study that finally evaluates these approaches with surprisingly strong results.

Given these results presented in this work, the challenge to the community of providing a model that can improve upon this heuristic is indeed justified.


## Weaknesses
- The reference section is a mess. A lot of references simply have a year without further information on the venue, while others do. Provided venues vary a lot, e.g., NIPS, NeurIPS, Advances in Neural Information Processing Systems. Please reformat all references into a consistent style.

### Typos
- Fig 3: Plot sizes and font sizes vary a lot.
- p8 last line: the ranks of the top-**1%** scoring...

---

> ### Author Response · Authors · 2023-05-03
> **Thank you!**
>
> Dear Reviewer NuWJ,
>
> Thank you for taking the time to review our paper and providing your insightful comments and feedback.
>
> We apologize for the inconsistency in the reference section and for the typos in Fig 3 and on page 8. We have addressed these issues by reformatting the reference section for consistency and correcting the typos and formatting in Fig 3 and on page 8.
>
> We appreciate your positive comments on the strengths of our proposed method and the value of our study in evaluating stochastic approaches in active learning. We agree that our contribution may be minor in terms of the originality of the proposed idea, but we believe that our study provides valuable insights into the effectiveness of stochastic batch acquisition methods.
>
> Thank you for highlighting the need for future research to improve upon our proposed method. We hope that our study will encourage further investigation into this area of research.
>
> Once again, we appreciate your feedback and have updated the paper accordingly. Please let us know if you have any further feedback for us.
>
> Thank you and best wishes,\
> the Authors

---

### Review · Reviewer_TQqT · 2023-02-20

**Summary Of Contributions:**

This paper proposes a simple way to extend any single-datapoint acquisition function for active learning to batch settings, where many labels are collected at once in a particular round before the model is re-trained.  The method is to simply sample from a distribution that upweights the datapoints with the highest acquisition values. This naturally encourages greater diversity than simply choosing the datapoints with K highest acquisition values, yet is not more computationally intensive.

I find the proposed method highly practical and the empirical experiments on many image datasets and some text datasets convincing. Everything looks correctly done as far as I can tell.

The one thing I am not sure about is whether this or similar methods have been previously proposed, given how straightforward they are.

**Audience:**

Yes

**Broader Impact Concerns:**

I don't have any broader impact concerns about this work.

**Claims And Evidence:**

Yes

**Requested Changes:**

If you want to present a more convincing case, consider including other recently-published batch active learning methods in your experimental comparisons. That said, I am already convinced that the proposed method is a good idea by the current experiments, especially given the method’s simplicity.

Adding more active learning methods would help convince me that your method truly matches the state-of-the-art. For instance, methods which use stochastic acquisition with explicit encouragement of diversity such as:  choosing top acquisition values sampled from different clusters of the data, or using the acquisition values and data features to define a determinantal point process distribution.

Batch Active Learning Using Determinantal Point Processes
https://arxiv.org/abs/1906.07975

Related Work should mention other related ideas like “kriging believer”. That is another very simple approach to extend single datapoint acquisition values to batch settings, but does still involve a lot of model retraining to compute the batch scores.


**Strengths And Weaknesses:**

Strengths:

How to properly extend acquisition functions to batch settings is an important problem in making active learning more practical, especially as model-training times grow larger for today’s large neural networks.

Many of the methods previously published in this area are highly complex and computationally intensive.
Papers like this one which show that a straightforward and cheap-to-implement/compute baseline outperforms large numbers of complicated publications serve as important reality-checks for the field. These are usually the papers with methods that end up actually being used in applications (eg. in industry/science).

Overall the method is clearly presented and the writing is easy to understand.

The experiments cover many different datasets.

Weaknesses:

The experiments do not compare against many other batch baseline methods. There have been many recent batch active learning methods published that could also be compared against. That said, I do find the existing experiments sufficient since they cover so many different datasets.

---

> ### Author Response · Authors · 2023-05-03
> **Thank you!**
>
> Dear Reviewer TQqT,
>
> Thank you for your valuable feedback on our paper. We appreciate your positive comments on the practicality of our proposed method and the clarity of our presentation.
>
> Regarding your suggested changes, we agree that it may be premature to claim that our method matches all state-of-the-art methods in batch active learning. We apologize for any confusion this may have caused and have updated our claims in the paper to be more precise and appropriate.
>
> We also agree that including other recently-published batch active learning methods in the experimental comparisons would provide a more comprehensive evaluation of our proposed method. Unfortunately, due to changes in our team's circumstances, we no longer have access to the same compute resources, which prevents us from running equivalent experiments for all datasets. However, we have made extensive additions to the paper and aim to add more experimental results to the camera-ready version.
>
> We have updated the related literature section to include the k-DPP paper you suggested, and we appreciate your valuable input on this matter. Regarding the kriging believer, we have looked into it and found it to be similar to GP acquisition, which can be related to current active learning methods (see https://arxiv.org/abs/2203.09410). Still, it seems only tangential to this paper. If you could provide further clarification on the relevance of the kriging believer to our work, we would be grateful.
>
> In conclusion, we hope that the changes we have made and will make in the paper will address your concerns and improve the quality of our work. Again, thank you for your feedback and time in reviewing our paper.
>
> Thank you, and best wishes,
> the Authors

---

### Review · Reviewer_uz6d · 2023-04-01

**Summary Of Contributions:**

This paper shows that complex batch acquisition strategies commonly used in active learning provide comparable predictive performance to the simple baseline of stochastic sampling from a softmax (or other distribution) based on the scores of the pool set, despite the latter strategy providing a vastly reduced computational complexity. This result calls into question the conventional wisdom in the field – that sophisticated batch acquisition strategies are worth their additional computational cost. The paper also discusses 3 different methods for constructing the sampling distribution, each making different assumptions about the scores. The paper shows that the power distribution performs slightly better than the softmax and soft-rank distributions for commonly used active learning score functions.

**Audience:**

Yes

**Broader Impact Concerns:**

No concerns.

**Claims And Evidence:**

No

**Requested Changes:**

I have grouped my requested changes into "Major" and "Minor", which are the changes which are critical for securing my recommendation and those that would simply strengthen the paper, respectively.

## Major changes

1. Please clarify the following. While some of these are minor issues in isolation, they add up to make the paper significantly less clear than it could be.
    1. What does figure 1 have to do with the statement "... and it was assumed deep learning models hardly change after adding a single new point to the training set."? Is the large change in rank correlation at the initial acquisition steps being referenced here?
    2. Some discussion should accompany equation 8 to highlight why this is a naive strategy. Figure 1 from the BatchBALD paper does a great job at this. A figure isn't necessary, but some discussion would be very helpful.
    3. How does the $\beta$ parameter of the softmax correspond to the "expected rate at which the scores changed as more data is acquired"?
    4. After equation 9, it isn't obvious how taking the top-k samples corresponds to sampling from the soft-rank distribution.
    5. Regarding power acquisition, what is meant by "ideally" in "This assumption holds ideally for other scoring functions ..."?
    6. Regarding power acquisition, how does the notion of time (e.g., "future log scores"/"current log scores") relate to equation 11?
    7. Similarly, in section 6, the assumptions regarding rank correlations across acquisitions were really not made clear to me.

2. Please add at least a few experiments comparing to Pinsler et al., as discussed above. (Pinsler et al. should also be discussed in the related work).

3. Please provide citations (or other demonstrations/evidence) for the following statements:
    1. In section 2, regarding entropy scoring, "... performs worse for data with high observation noise ...".
    2. In section 2, regarding acquisition functions, "... and it was assumed deep learning models hardly change after adding a single new point to the training set." Who assumed this?
    3. In section 3, "... which is frequently used for modelling extrema.".
    4. In section 5, "... BatchBALD, which was SotA for small batch sizes.".
    5. Similarly, in the caption for Figure 6, BatchBALD is mentioned as being SotA without a reference.

4. In figures 2, 3a and 3b, why does PowerBALD use an acquisition size of 10? This seems like it might make the comparison between it and BatchBALD unfair. Can you provide results for PowerBALD with an acquisition size of 5? And/or an ablation for the acquisition size for PowerBALD like was done for BADGE in figure 14? In general, I also think that the choices of hyperparameters should be better motivated, and any sweeps done should be described.

5. Similarly, why was the acquisition size for figures 3c and 3d raised to 100? My concern is that with a larger acquisition size, the impact of low diversity in the acquired examples can be mitigated. Would it be possible to see these results for an acquisition size of, say, 10?

6. It should be mentioned that for the CLINIC results, whose figure is somewhat hidden in the appendix, PowerEntropy does perform slightly worse than BADGE.

7. This might be a big misunderstanding on my part, but in Figure 4, it is not clear to me that the "Top" and "Bot" lines should be so similar. In fact, I think this contradicts assumption (3) – aren't the bottom 1% or 10% supposed to be less informative, so shouldn't the change in correlation in these two cases be closer to, if not even worse than for 100%? Could you clarify this? This currently leaves me somewhat sceptical of the results of these further investigations. Perhaps providing the additional experiment (just on MNIST, for example) that K can indeed be increased would provide additional evidence here.

8. An important missing sensitivity study is to use different BNN methods. Currently, the paper only uses MC Dropout for the BNN, so it is unclear whether these results generalize to other BNN approximations. In particular, I think that the linearised Laplace approximation (e.g., see Daxberger et al. (2021)) would be a great addition. Results for a single dataset would suffice.

Erik Daxberger, Agustinus Kristiadi, Alexander Immer, Runa Eschenhagen, Matthias Bauer, Philipp Hennig:
Laplace Redux - Effortless Bayesian Deep Learning. NeurIPS 2021: 20089-20103

## Minor changes

1. After equation 2, I would suggest saying, "... the index set of $\mathcal{D}^\text{train}$ **initially** containing ...".

2. The footnote on page 2 should be removed.

3. In section 2, provide the expansion of the acronym for BALD as it helps provide intuition behind the method.

4. In section 2, provide brief descriptions of BatchBALD and BADGE since these are the most important baselines in the experiments.

5. Regarding the soft-rank acquisition, to be clear, there is only invariance insofar as the ranking us unchanged when the scores change.

6. Since BADGE and BatchBald are (understandably) not included in every figure, it would be helpful to always include "Uniform", so that the improvements over Bald from the stochastic aquisiton have a consistent point of comparison.

7. In the summary for section 5, the papers listed are not exactly "new", two being from 2021.

8. It would be great to see the $\beta$ ablation for the EMNIST dataset too.


**Strengths And Weaknesses:**

## Strengths

1. Challenging the conventional wisdom in batch active learning is both important and interesting. This paper has the potential to act as a stepping stone for the development of better theory and methods for batch active learning.

2. The paper advocates for a simple but effective baseline which is underutilized in the literature and likely in practice too.

3. The paper provides an interesting discussion of 3 different methods for constructing a distribution from which to sample from the pool set given scores. The power and soft-rank distributions are novel, as far as I am aware.

## Weaknesses

1. **Clarity:** the paper is unclear in several places; see the detailed list under "Requested Changes". The main issue in this regard is the connection between sampling from the acquisition distributions and the idea that scores at time $t+1$ are the scores at time $t$ with some perturbation applied. This idea is mentioned a few times in the paper but is not clearly described or obvious to me.

2. **Evidence:** not all claims are well supported by evidence. I believe that most of these issues can be addressed by adding citations. However, one issue I am concerned about is that the paper only uses BatchBALD and BADGE as examples of batch acquisition strategies. The problem is that these are *extremely* computationally expensive (either in memory or compute time). However, there exist well-performing batch acquisition strategies that scale much more gracefully. In particular, ACS-FW (with random projections) from Pinsler et al. (2019) scales linearly with pool-set size. (The authors also mention in the related work that methods exist for improving the scaling of core-set-based methods). Without including comparisons to some batch acquisition methods with better scaling, and thereby providing a more comprehensive picture of such methods, I do not believe that the paper's (interesting and potentially important) question about whether or not batch acquisition methods are "pulling their weight" can be taken seriously.

3. **Positioning:** the positioning of the paper is, in my opinion, somewhat strange and misleading. In particular, the idea of sampling from a distribution based on active learning scores in order to improve diversity in batch acquisition might not be well-known (and could definitely benefit from some advocacy) but has certainly been used before. It is a straightforward implementation of Boltzmann exploration to the active learning setting. Yet the paper presents this as a novel method, frequently referring to it as "our method", and the second claimed contribution of the paper is the demonstration that the sampling strategy improves on naive top-K acquisition. The specific implementations using the power and soft-rank distributions are novel as far as I know, but this paper is not clear enough in disentangling what is novel and what is not. (To be clear, I am not criticising this paper for lack of novelty. I am aware that this is not a criterion for acceptance in TMLR. Furthermore, I think there is plenty of novel/interesting content in the paper. However, the paper should be clear about its contributions and the sources of novelty.)




Robert Pinsler, Jonathan Gordon, Eric T. Nalisnick, José Miguel Hernández-Lobato:
Bayesian Batch Active Learning as Sparse Subset Approximation. NeurIPS 2019: 6356-6367

---

> ### Author Response · Authors · 2023-05-03
> **Thank you!**
>
>
> Dear Reviewer uz6d,
>
> Thank you very much for the detailed feedback and useful suggestions that will significantly improve our paper. We appreciate the time and effort you have invested in reviewing our work.
>
> **TL;DR:** We have addressed many of the major and minor requested changes. Due to changes in our team's circumstances, we have not yet been able to run all the additional experiments in detail, but we are working on them. We have made significant improvements in the paper's clarity, and we have started running experiments using the code provided by Pinsler et al. on Repeated-MNIST. We hope to deliver results soon and update the figures to include the uniform baseline when available.
>
> We hope we have addressed the major requests within the paper. In particular, we want to highlight that:
>
> * We have added a different figure that hopefully conveys something similar to the figure in the BatchBALD paper (based on the BALD score distribution grouped by class which also highlights the redundancy issue).
> * We have looked at Figure 4 and the rank correlations experiments in detail. Your concern was very helpful. Indeed, the bottom ranks frequently swap their order because their scores are very close to 0, and the estimator noise heavily influences their rankings. We have rounded all scores to two decimal scores (ranging from 0..~1.2 nats), which reduces this issue for the bottom scorers and resolves your concern.
>     * Additionally, we have examined the AUROC of the score separation for different quantiles, which is equivalent to the probability that points from the top-K batch (1%, 10%, 50%) at $t$ have scores larger than other points at $t+n$ (which would be beneficial for top-K). This entirely ignores ranks within the top-K group and within the outside group. We observe the expected behavior which validates the rank correlation results further. A sanity check on the bottom-K batches behaves as you would expect, too.
>     * (Now) Figure 6 provides another sanity check on MNIST that shows the results for various K at both $t=0$ and $t=100$ in line with what you requested.
> * We have motivated the use of the Gumbel distribution in more detail. We also hope we have been able to clarify the motivation better overall.
> * We have expanded the investigation section to more concretely discuss the acquisition asymptotics of BALD and entropy under some strong but sensible assumptions and point out some theoretical challenges due to the sampling-based estimator in BatchBALD.
>
> Overall, we have tried to clarify the contributions and claims and ensured we only refer to BatchBALD and BADGE. We hope this addresses your concerns about the evidence we provide.
>
> Sadly, due to other deadlines and conference travel, we have been unable to finish the additional requested experiments at this time. However, we have started running experiments using the code provided by Pinsleret al. on Repeated-MNIST and hope to deliver results soon.
>
> We also want to provide results using a method different from MC Dropout. Pinsler et al. compute scores using linearization, too, which might offer an attractive solution to provide another baseline and BNN approach.
>
> Regarding your questions about PowerBALD and the acquisition sizes used, we can rerun experiments for this. However, we already use BatchBALD and BADGE at their strongest in our comparisons (according to the ablations we have run):
> 1. At acquisition batch size 1, BatchBALD and top-K BALD are equivalent to BALD  (BALD-1), and in particular top-K BALD converges to BALD-1, while BatchBALD-5 performs pretty much exactly like BALD-1. BALD-1 performs the best (this also matches the results in the BatchBALD paper. BatchBALD-10 takes much longer to run and already performs worse.
> 2. The appendix contains an ablation for acquisition batch sizes for the stochastic acquisition functions in Figure 28. We will add another figure to the camera-ready to make it easier to compare across different batch acquisition sizes (instead of different β).
> 3. Drawing from our experience with BADGE, its performance tends to be less effective at a batch acquisition size of 10. We opted for a batch size of 100, as it balanced addressing your concerns and managing the computational costs associated with the experiments.
>
> In conclusion, we hope that the significant changes address your concerns. We are incredibly grateful for your detailed comments and suggestions that have improved the quality of our work and will do so further for the camera-ready. Thank you once again for your feedback and for your time in reviewing our paper.
>
> Thank you, and best wishes,\
> the Authors

---

> > ### Author Response · Authors · 2023-05-08
> > **First results for ACW-FS**
> >
> > Dear Reviewer uz6d,
> >
> > We are pleased to share the preliminary results of additional experiments comparing ACW-FS, PowerBALD, and BALD on RepeatedMNISTx3 (https://pasteboard.co/KMZkNxixJXfG.png). These experiments used three seeds and were conducted in the codebase of Pinsler et al. (2019), which we updated to include PowerBALD. The results indicate that PowerBALD performs well in a setting similar to the experiments in our codebase when compared to ACW-FS. We aim to provide more comprehensive results soon and hope that this information offers reassurance for the future camera-ready version of our paper. Please note that we are facing some resource constraints due to the approaching NeurIPS deadline.
> >
> > We eagerly anticipate your feedback.
> >
> > Sincerely,\
> > The Authors

---

> > ### Comment · Reviewer_uz6d · 2023-05-09
> > **Great improvements, but still some work to be done**
> >
> > Thanks for engaging with my review! I think that the paper is looking much better as a result. However, there are still some areas which I think could be improved.
> >
> > I still think there is room for improvement in the positioning of the paper. In particular, I think the contributions and novelty could be clearer. That is, make it clear that the idea of doing Boltzman exploration in active learning is not new but under-explored. And that the paper's contributions are a detailed exploration of this idea–with some surprising results–and two novel implementations using the power and soft-rank distributions. Additionally, change mentions of "Our stochastic method" (e.g., in conclusion) to be more specific (e.g., "Boltzaman exploration with our proposed power distribution implementation"). I feel strongly about this point.
> >
> > Regarding experimental results. I understand that it is difficult to get all of the additional experimental results due to your circumstances. Furthermore, I am convinced from your preliminary results that you are acting in good faith and will endeavor to have the key results (a complete comparison with ACW-FS, an ablation for a different BNN type, and the figure for comparing different acquisition sizes) done for the camera ready. Thus, the experimental results are no longer a major concern for me.
> >
> > Regarding this comment, "Drawing from our experience with BADGE, its performance tends to be less effective at a batch acquisition size of 10. We opted for a batch size of 100, as it balanced addressing your concerns and managing the computational costs associated with the experiments." -- I do understand that you were trying to represent BADGE well. However, I still think it is worth seeing the results for a batch acquisition size of 10 due to my concern that the impact of low diversity in the acquired examples can be mitigated with a larger acquisition size. Do you disagree that this is a potential concern?
> >
> > A few minor, and easy to fix, issues that are worth addressing in the paper (apologies if I missed these; please let me know):
> > 1. A mention of ACS-FW in the related work.
> > 2. A discussion of how $\beta$ represents the expected rate at which the scores change.
> > 3. A discussion of why/how the Gumbel is used to model extrema.
> > 4. A clarification about the exact nature of the invariance to the actual scores when using soft-rank. (I.e., the ranking of the raw scores is still important.)

---

> > > ### Author Response · Authors · 2023-05-15
> > > **Thank you!**
> > >
> > > Dear Reviewer uz6d,
> > >
> > > We appreciate your supportive feedback! Our apologies for missing references to "our" in the previous version. We have updated the paper to correct these oversights.
> > >
> > > We have added to the introduction:
> > >
> > > > In sequential decisison making, stochastic multi-armed bandits and reinforcement learning, variants of this are also known a Boltzmann exploration \citep{cesa2017boltzmann}.
> > >
> > > Additionally, we have clarified our contributions:
> > >
> > > > examine a family of three computationally cheap stochastic batch acquisition strategies (softmax, power and soft-rank)---the latter two which have not been explored and compared to in detail before;
> > >
> > > We agree that larger acquisition batch sizes could mitigate the issue of diversity. We will run an experiment to compare stochastic acquisition methods to BatchBALD using the same batch acquisition size instead of a larger one for stochastic batch acquisition as you suggested.
> > >
> > > In response to your points:
> > >
> > > Re 1. We have included Pinsler et al. in the related work section and, more significantly, we have incorporated it into the background section, where we also discuss BADGE and BatchBALD, in anticipation of the additional experimental results.
> > >
> > > Re 2 & 3. We have clarified $\beta$ and removed the mention of the expected rate of score change. e expanded the initial discussion about Gumbel noise to explain why it is a meaningful method for modeling the expected score increase, considering not-yet-selected additional acquisition points. We hope we have sufficiently emphasized that our aim is to provide motivation, not a formal explanation. We found another more recent related work that has formal statements for Softmax acquisition in a non-deep-learning context that we now include in our related work section. However, this work does not offer an intuitive motivation.
> > >
> > > Specifically, in our motivational model, we assume that the acquisition scores of not-yet-selected points follow an exponential distribution as a rough approximation. After selecting additional samples, we expect this rate to change, affecting the overall expected increase in acquisition scores. The sum of exponential random variables with different rates follows a hypoexponential distribution. We have yet to find a result about the tails of this distribution or the maximum distribution, and we apologize for our limited knowledge in this specific area of statistics.
> > >
> > > However, we have conducted a simple empirical evaluation that suggests the maximum distribution follows a Gumbel distribution as well. We will include a link to a Colab notebook with the camera-ready version. To maintain anonymity, we provide a simple script for visualizing the fit here:
> > >
> > > ```python
> > > import numpy as np
> > > from scipy.stats import expon, gumbel_r
> > > import matplotlib.pyplot as plt
> > >
> > > # Set the rate parameter
> > > # Change here to get different scales of exponentials
> > > lam = 1/2
> > >
> > > # Generate rate samples from an exponential distribution
> > > samples = expon.rvs(scale=1/lam, size=5000)
> > >
> > > # Now sample from the samples using the previous rate samples as rate of exponential distributions
> > > new_samples = expon.rvs(scale=1/samples, size=(4000,5000))
> > >
> > > what_are_you = new_samples.mean(axis=1)
> > >
> > > # Fit a Gumbel distribution to the data
> > > params = gumbel_r.fit(what_are_you)
> > >
> > > x = np.linspace(np.min(what_are_you), np.max(what_are_you), 1000)
> > > gumbel_cdf = gumbel_r.cdf(x, *params)
> > >
> > > # Plot both what_are_you and the Gumbel distribution in the same plot
> > >
> > > plt.plot(np.sort(what_are_you), np.linspace(0, 1, len(what_are_you), endpoint=False), label="Simulated Hypoexponential")
> > > plt.plot(x, gumbel_cdf, label="Gumble Fit")
> > > plt.legend()
> > > plt.show()
> > > ```
> > >
> > > Re 4. We have made it clear in the summary that "However, even the soft-rank acquisition works well in practice, suggesting that the choice of score perturbation is not critical for its effectiveness, **provided that the ranks are not significantly affected.**"
> > >
> > > In summary for the camera-ready:
> > >
> > > - a complete comparison with ACW-FS, an ablation for a different BNN type, and the figure for comparing different acquisition sizes;
> > > - $\beta$-ablation plot for EMNIST;
> > > - acquisition batch size ablation plot using PowerBALD acquisition batch size 5;
> > > - addition of uniform baselines in figures that do not include BatchBALD or BADGE.
> > >
> > > Apologies for the delay in our reply. We hope you find our further changes encouraging. Please let us know of any feedback or corrections, and for engaging extensively with our submission and helping us improve it.
> > >
> > > Best wishes,\
> > > the Authors
> > >
> > > PS: To see all the changes, here is an online PDF diff: [https://draftable.com/compare/LWbDRDTvqesb](https://draftable.com/compare/LWbDRDTvqesb)

---

> > > > ### Comment · Reviewer_uz6d · 2023-05-17
> > > > **Looking good!**
> > > >
> > > > Thanks for all the clarifications, changes, and willingness to keep improving the paper! I look forward to seeing the final version with all the new results. At this point, I am happy that the camera-ready version will meet the requirements for publication in TMLR, and I recommend acceptance conditioned on the additional results being included.

---

> > > > > ### Author Response · Authors · 2023-05-22
> > > > > **Thank you!**
> > > > >
> > > > > Dear Reviewer uz6d,
> > > > >
> > > > > Thank you for your constructive feedback and recommendation for our paper's acceptance. Your insightful comments have greatly helped improve the quality of our work.
> > > > >
> > > > > We will incorporate all the additional results in the final version of our manuscript, as promised. Your trust in our work is greatly appreciated.
> > > > >
> > > > > Again, thank you for your valuable input and the time you spent reviewing our paper.
> > > > >
> > > > > Best wishes,\
> > > > > The Authors

---

### Decision · Action_Editors · 2023-06-05

**Recommendation:** Accept with minor revision

**Comment:**

The reviewers found the contribution of this paper to be useful: a simple strategy can work as a convincingly effective baseline for batch mode active learning. The AE concurred and recommended acceptance.

In addition to incorporating the reviewers' comments in the final version (if not yet), the AE also suggests the following revisions in the final version:

(Main suggestions)
- In the comparison plots, using the same K for all algorithms except for BatchBALD (and clarify the computational challenges & anticipated performance decline of running BachBALD with the same K).

- Clarify the choices of K in all experiments for all algorithms

(Minor comments)
- Is it possible to clarify the motivation of the Gumbel perturbation on the acquisition scores in pages 5-6 more? If I understand correctly, the last paragraph of page 5 says that, for choosing the k-th point in the batch, ideally we would like to model the information gain of this k-th point (relative to the batches of examples queried so far and the k-1 examples already selected in this batch) assuming that the remaining K-k points are selected optimally. The paper then mentioned the maximum of a set of exponential random variables; however I had difficulty in understanding what are the iid exponential random variables in the context of batch active learning. It might be better to remove the discussion if this connection is not so precise..

- could you clarify why in Eq. (17), log M is greater than the mutual information expression? M depends on the number of samples, where as the mutual information expression does not?

- could you provide a mathematical expression of the AUROC evaluated in Figure 7?

- In Figure 9(b), why is it called a near-duplicate? Is it because it is very similar to Figure 9(a)?

- typo in B.2: Gumbel -> Gumble


**Audience:**

This paper will be of good interest to practitioners of deep / batch-mode active learning.

**Claims And Evidence:**

The paper conducts evaluations of several active learning strategies across an extensive list of experimental settings.

---

> ### Author Response · Authors · 2023-09-18
> **Camera-Ready Revision**
>
> Dear AE,
>
> We appreciate your patience and are pleased to inform you that the camera-ready version has been uploaded.
>
> For easy comparison, an interactive online diff is available at: [Interactive Diff](https://draftable.com/compare/UzuQuEFrPwOP).
>
> **Changes and Updates:**
>
> - **Comparison Plot:** Enhanced Fig. 2 and added a plot with acquisition batch size 5 in Fig. 11 (Appendix §C.2).
> - **Choices of \( K \):** Clarifications made throughout the manuscript.
> - **Gumbel Motivation:** Relocated to §6, per your recommendation.
> - **Empirical BatchBALD Bound:** Clarified as indicated.
> - **AUROC:** Added formula and clarified semantics.
> - **"Duplicates" in Fig. 9:** Formatting corrected and clarification provided.
>
> **Additional Material:**
>
> - **MFVI and ACS-FW:** Added hooks and TL;DR to main text; results presented in §C.7; setup detailed in §C1.7.
> - **$\beta$-Ablation Plot for EMNIST:** Plots included in Appendix §F.2 for both EMNIST subsets (ByMerge and Balanced).
>
> We extend our heartfelt thanks to you, the AE, and all reviewers for the insightful and iterated feedback that has significantly and truly improved the quality of the paper.
>
> Best regards,\
> the Authors